# Mislocalization of pathogenic RBM20 variants in dilated cardiomyopathy is caused by loss-of-interaction with Transportin-3

Julia Kornienko [1,2,3], Marta Rodríguez-Martínez[1], Kai Fenzl [1,3], Florian Hinze[4,5,6], Daniel Schraivogel[1], Markus Grosch [1,3,7], Brigit Tunaj[1], Dominik Lindenhofer [1], Laura Schraft [1], Moritz Kueblbeck[1], Eric Smith [8], Chad Mao [9], Emily Brown[10], Anjali Owens[11], Ardan M. Saguner[12], Benjamin Meder [13], Victoria Parikh [14], Michael Gotthardt [4,5,6] & Lars M. Steinmetz[1,3,7,15] ✉

Severe forms of dilated cardiomyopathy (DCM) are associated with point mutations in the alternative splicing regulator RBM20 that are frequently located in the arginine/serine-rich domain (RS-domain). Such mutations can cause defective splicing and cytoplasmic mislocalization, which leads to the formation of detrimental cytoplasmic granules. Successful development of personalized therapies requires identifying the direct mechanisms of pathogenic RBM20 variants. Here, we decipher the molecular mechanism of RBM20 mislocalization and its specific role in DCM pathogenesis. We demonstrate that mislocalized RBM20 RS-domain variants retain their splice regulatory activity, which reveals that aberrant cellular localization is the main driver of their pathological phenotype. A genome-wide CRISPR knockout screen combined with image-enabled cell sorting identified Transportin-3 (TNPO3) as the main nuclear importer of RBM20. We show that the direct RBM20-TNPO3 interaction involves the RS-domain, and is disrupted by pathogenic variants. Relocalization of pathogenic RBM20 variants to the nucleus restores alternative splicing and dissolves cytoplasmic granules in cell culture and animal models. These findings provide proof-of-principle for developing therapeutic strategies to restore RBM20's nuclear localization in RBM20-DCM patients.

Correct protein localization is fundamentally based on the recognition of a targeting signal within the nascent protein by a targeting factor for the destination organelle[1]. Gene variants impairing these targeting signals often result in severe diseases[2]. For example, cytoplasmic mislocalization of p53 facilitates cancer progression[3], and cytoplasmic mislocalization of TDP-43 is associated with Amyotrophic Lateral Sclerosis[4]. Details about the transport mechanisms involved are required for developing targeted therapies and are still elusive for many

[1]Genome Biology Unit, European Molecular Biology Laboratory (EMBL), Heidelberg, Germany. [2]Faculty of Biosciences, Heidelberg University, Heidelberg, Germany. [3]DZHK (German Centre for Cardiovascular Research), partner site Heidelberg/Mannheim, Heidelberg, Germany. [4]Max Delbrück Center for Molecular Medicine in the Helmholtz Association, Berlin, Germany. [5]German Center for Cardiovascular Research (DZHK), Partner Site Berlin, Berlin, Germany. [6]Charité-Universitätsmedizin Berlin, Berlin, Germany. [7]Department of Genetics, Stanford University School of Medicine, Stanford, CA, USA. [8]University of Michigan, Ann Arbor, MI, USA. [9]Children's Healthcare of Atlanta & Emory University, Atlanta, GA, USA. [10]Johns Hopkins Hospital, Baltimore, MD, USA. [11]University of Pennsylvania, Philadelphia, PA, USA. [12]Department of Cardiology, University Heart Center Zurich, University Hospital Zurich, Zurich, Switzerland. [13]Cardiogenetics Center Heidelberg, Department of Cardiology, Angiology and Pulmology, University Hospital Heidelberg, Heidelberg, Germany. [14]Stanford Center for Inherited Cardiovascular Disease, Division of Cardiovascular Medicine, Department of Medicine, Stanford University School of Medicine, Stanford, CA, USA. [15]Stanford Genome Technology Center, Palo Alto, CA, USA. ✉e-mail: Lars.Steinmetz@stanford.edu

cases. Our recent image-enabled cell sorting (ICS)[5] technology allows for high-throughput isolation of cells with mislocalized proteins from a heterogeneous population, and can be combined with functional genomics, transcriptomics, and other analyses. Here, we apply this combinatorial approach for the first time to decipher the mechanism of dilated cardiomyopathy (DCM) caused by mislocalized RBM20.

DCM is a heart condition characterized by enlargement of the cardiac left ventricle and systolic dysfunction. It is a highly prevalent disease affecting 1 in 250–500 individuals and eventually leading to heart failure or sudden cardiac death[6,7]. Besides heart failure therapy, targeted approaches are largely lacking in clinics[8], and the only available curative treatment is heart transplantation. About half of DCM cases are familial with primarily autosomal dominant inheritance[9]. Variants of several genes have been classified with high confidence as DCM-causing in humans[10,11], including those in the RNA-binding motif protein 20 (*RBM20*). *RBM20* variants can result in a particularly severe form of the disease often causing arrhythmia and progressive heart failure, and account for about 3% of familial DCM cases[12–15]. Current guidelines for RBM20 patients suggest evaluation for primary prophylactic placement of implantable cardioverter defibrillators based on individual predicted risk[14–16]. RBM20 is predominantly expressed in the heart, and is involved in the regulation of tissue-specific alternative splicing[17]. Among its targets are genes involved in sarcomere structure (e.g., *TTN*), mitochondrial function (e.g., *IMMT*), calcium handling (e.g., *RYR2*), and ion channels (e.g., *CACNA1C*)[18–20]. The majority of DCM-causing *RBM20* variants are heterozygous missense mutations, many of which cluster in a conserved stretch encoding for six amino acids PRSRSP (amino acid position 633–638) in the protein's arginine/serine (RS)-rich domain[12,18,21,22].

Heterozygous mutations in *RBM20* result in haploinsufficiency with respect to transcriptional splicing[18,19,22,23], where alternative splicing of RBM20's targets is proportional to the amount of wild type (WT) versus mutated RBM20 expressed[24]. We previously demonstrated that a compound upregulating *RBM20* expression alleviates the disease phenotype in induced pluripotent stem cell-derived cardiomyocytes (iPSC-CMs)[22]. However, recent studies suggested that some *RBM20* variants display gain-of-function effects related to mislocalization of the mutated protein outside the nucleus. In a porcine DCM model, as well as in patient-derived iPSC-CMs harboring the R636S mutation, RBM20 mislocalized to the cytoplasm and formed potentially detrimental RNP granules[24]. In a murine DCM model, the S637A mutation (S635A in humans) caused similar mislocalization, lower survival and higher levels of fibrosis compared to an *Rbm20* knockout (KO)[25]. Unlike a full KO in vivo, *Rbm20*-mutant mice showed changes in global expression of genes involved in cardiac function[25,26]. In human iPSC-CMs, RBM20-R636S protein preferably bound to the 3' UTR of transcripts in the cytoplasm, and co-localized with the P-body marker DDX6[27]. Altogether, aberrant localization of RBM20 was shown for mutations of all residues in the PRSRSP stretch[24–31]. Overall, it is unclear whether these mutations affect RBM20's intrinsic role as a mediator of spliceosome activity, or whether the splicing haploinsufficiency results from protein mislocalization alone. This remains uncertain because the mechanism driving RBM20 cellular localization is unknown.

In this study, we show that pathogenic RS-domain variants do not disrupt the splice-regulatory activity, and that the splicing defect is mainly due to mislocalization of RBM20. We uncover the molecular basis of RBM20's nuclear transport, and demonstrate how RS-domain mutations disrupt this process. Our findings have implications for the development of therapeutic strategies targeted at improving the nuclear import of mislocalized RBM20.

## Results

### Splice-regulatory activity of RBM20 variants is proportional to their nuclear localization

We analyzed the localization of homozygous RBM20-P633L and -R634Q variants that we engineered before[22] in human iPSC-CMs by immunofluorescence (IF) staining followed by confocal microscopy (Fig. 1a, b) or ICS (Fig. 1c). The R634Q variant resulted in severe cytoplasmic mislocalization and granule formation of RBM20, as shown for other described RBM20 mutations[24–29]. In contrast, the P633L mutation resulted in only partial mislocalization of RBM20 to the cytoplasm. The partially correct localization of P633L correlated with a less severe splice phenotype as determined by the analysis of *TTN* isoform expression (Fig. 1d), and global splicing activity in comparison to R634Q[22]. These differences were independent of *RBM20* expression levels (Supplementary Fig. 1a). Similar localization patterns were observed in HeLa cells stably expressing eGFP-tagged WT or mutated RBM20 (Supplementary Fig. 1b–f). *IMMT* splicing measured by qPCR correlated with mislocalization of RBM20 variants in HeLa cells (Supplementary Fig. 1g). These data demonstrate that RBM20-P633L causes milder mislocalization and mis-splicing compared to other RS-domain variants.

To verify the clinical relevance of this finding, we collected data from patients with pathogenic or likely pathogenic (P/LP) variants in the RSRSP stretch of *RBM20* who were identified by cascade family screening. We compared patients with the P633L variant to the rest of the cohort (Fig. 1e–g). In patients with the P633L mutation, ventricular remodeling, a characteristic of RBM20-DCM[14], was less severe. Left ventricular ejection fraction (LVEF) for patients with P633L was normal to mildly decreased and was in the top 50% of other P/LP variant patients at the time of diagnosis (Fig. 1e, LVEF range for healthy individuals is 53–73%[32]). This preserved function was not explained by younger age compared to other cases (Fig. 1f). Internal Left Ventricular Diastolic Dimension (LVIDd) normalized to body surface area (BSA) was normal or borderline normal (< 3 cm/m²) for P633L patients (Fig. 1g, LVIDd/BSA range for healthy individuals is 2.4–3.2 cm/m²[32]). These data offer clinical corroboration of a milder effect of the P633L variant on the underlying mechanism of RBM20-DCM as compared to other pathogenic variants in the RS-rich domain.

The mixed phenotype (nuclear and cytoplasmic) of homozygous RBM20-P633L iPSC-CMs allowed us to differentiate between the consequences of nuclear and cytoplasmic RBM20 localization in the same genetic background. To that end, we compared gene expression, alternative splicing, and protein interactor changes between differentially localized RBM20-P633L, RBM20-WT, and RBM20-R634Q.

Using ICS, we sorted iPSC-CMs with differentially localized RBM20 from homozygous P633L mutation background based on correlation with a nuclear staining (Fig. 1h and Supplementary Fig. 2a). This was followed by RNA sequencing of the sorted populations. We identified 1415 differentially expressed genes in P633L-cytoplasmic (P633L-cyt) compared to WT (Fig. 1i and Supplementary Data 1). In contrast, there were only 50 differentially expressed genes between P633L-nuclear (P633L-nuc) and WT (Fig. 1i and Supplementary Data 1). Downregulated genes in both P633L-cyt and R634Q impacted cardiac-related processes (Supplementary Fig. 2b–e and Supplementary Data 2). Similarly, gene expression of the core RBM20 targets[13] was either unchanged or downregulated in P633L-cyt, unlike P633L-nuc (Supplementary Fig. 2f). This indicates that mislocalization of RBM20 to the cytoplasm may down-regulate genes important for cardiomyocyte function, as previously suggested[25–27]. Overall, nuclear-localized RBM20-P633L caused minor changes in gene expression compared to WT (50 instead of 1415 for cytoplasmic RBM20-P633L).

We tested splicing changes between the sorted populations of iPSC-CMs. The list of RBM20 targets employed in this study consists of 45 different genes[13] (further referred to as "core RBM20 targets"). We used these genes to estimate the splicing activity of RBM20, as their splicing was consistently affected across *RBM20* perturbations (e.g., KO, S635A) and across model species (rat, human, mouse)[15,18,19]. For all exons of the core RBM20 targets[13], we assessed the percentage of spliced-in (PSI) values (Supplementary Data 3). PSI is defined as the

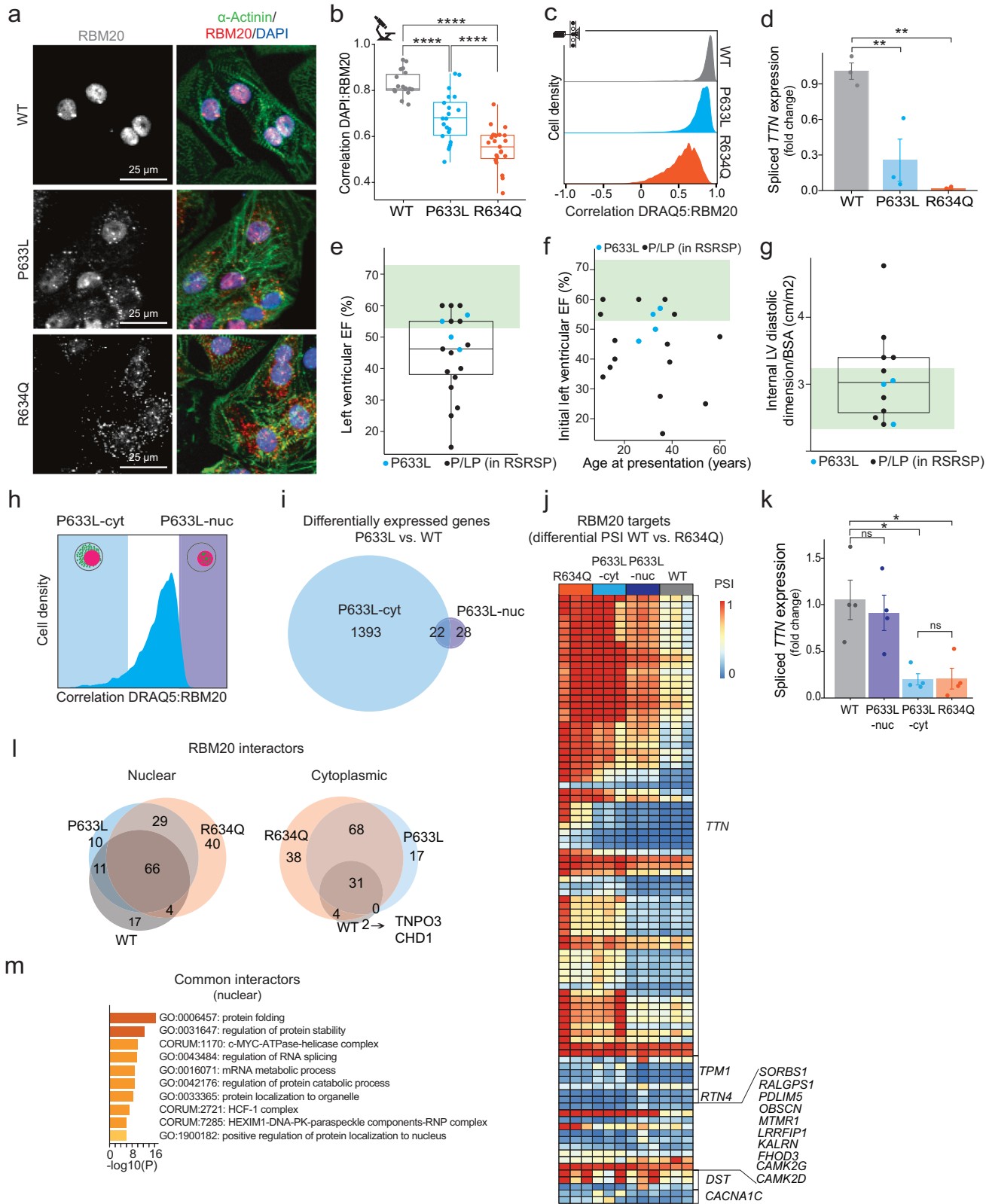

proportion of all sequenced reads that include the given exon to total reads for this exon (spliced-in and -out, see "Methods"). In cells with R634Q and P633L-cyt, the majority of alternatively spliced exons were spliced-in (Fig. 1j). In contrast, in WT and P633L-nuc cells, they were predominantly spliced-out (67% AS events restored in P633L-nuc from those mis-spliced in P633L-cyt). We confirmed the finding by qPCR analysis of *TTN* splicing (Fig. 1k). Altogether, our results demonstrate

that nuclear relocalization of P633L at least partially rescues alternative splicing of the core RBM20 targets.

To analyze global splicing changes in pairwise comparisons to RBM20-WT iPSC-CMs, we employed rMATS[33]. We observed fewer differential splicing events in P633L-nuc (234) compared to P633L-cyt (522) or R634Q (307, Supplementary Fig. 2g–i, Supplementary Data 4, see "Methods"). We compared our results to ref. 22, and provide lists of

**Fig. 1 | Splice-regulatory activity of RBM20 variants is proportional to their nuclear localization. a** Immunofluorescence (IF) staining for RBM20 and alpha-actinin in iPSC-CMs, $n = 3$. **b** DAPI:RBM20 colocalization based on data from (**a**). Each dot represents a Pearson coefficient for at least five cells, $n = 3$. **c** ICS-based analysis of DRAQ5:RBM20 correlation in iPSC-CMs, $n = 3$. **d** qPCR analysis of *TTN* exon 242 splicing-out in iPSC-CMs normalized to *GAPDH* expression displayed as fold change versus the first replicate of RBM20-WT (means with standard errors, $n = 3$). **e** Left ventricular ejection fraction (LVEF) for patients with P633L ($n = 4$) or with other pathogenic or likely pathogenic (P/LP) mutations in the RSRSP stretch ($n = 15$) at diagnosis (norm for healthy individuals: 53 to 73%, green shading). Mean LVEF for P633L vs. P/LP variants: $52 \pm 4.9\%$ vs $43.1 \pm 13.9\%$, $P = 0.059$ (two-sided *T*-test). **f** Initial LVEF as a function of the age at presentation (norm for healthy individuals: 53 to 73%, green shading, R2 = 0.001, $P = 0.89$, linear regression). **g** Internal Left Ventricular Diastolic Dimension (LVIDd) corrected for body size (norm for healthy individuals: 2.4–3.2 cm/m², green shading). Mean LVIDd/BSA for P633L ($n = 3$) vs. P/LP variants ($n = 9$): $3.2 \pm 0.74\%$ vs $2.8 \pm 0.26\%$, $P = 0.28$ (two-sided *T*-test). **h** ICS-sorting strategy for RBM20-P633L, $n = 3$. **i** Numbers of differentially expressed genes ($|\log 2FC| > 1$ and $P < 0.1$, Wald test with Benjamini–Hochberg correction, DeSeq2[58]) in pairwise comparisons to RBM20-WT, $n = 3$. **j** PSI values for alternative splicing events in the core RBM20 targets different between WT and R634Q ($|\text{delta PSI}| > 0.1$, $P$ value $< 0.05$, two-sided *T*-test). **k** qPCR analysis of *TTN* exon 242 splicing-out in sorted iPSC-CMs, same representation as (**d**), two biological replicates, with two technical replicates each. **l** Enriched RBM20 interactors (FDR < 0.05 (Limma), and $|\log 2FC| > 0.5$ vs. no-bait control) in HeLa, $n = 3$. **m** Pathway enrichment analysis for common interactors between RBM20-WT, -P633L, and -R634Q in the nuclear fraction, Metascape[59]. Boxplots (**b**, **e**, **g**) display quartiles Q1, Q2 (center), and Q3, with whiskers extending to the furthest data points within 1.5 times interquartile range (IQR). Statistical significance for (**b**, **d**, **k**) was calculated using one-way ANOVA with Tukey's HSD post test (two-tailored): Ns = not significant, *$P < 0.05$, **$P < 0.01$, ***$P < 0.001$, ****$P < 0.0001$. Actual $P$ values are shown in the source data file.

common AS events including those that are not located in the core RBM20 targets in Supplementary Data 5. The latter indicates potentially novel splicing targets that may associate with P633L and R634Q variants.

Next, we investigated whether the protein binding partners of nuclear or cytoplasmic RBM20-P633L match the nuclear WT or the cytoplasmic R634Q variant, respectively. We performed mass-spectrometry analysis of the interactors that co-immunoprecipitated with RBM20-WT, -P633L, and -R634Q in the nuclear or cytoplasmic fraction in HeLa cells (Supplementary Fig. 3a–c; see "Methods", Supplementary Data 6). We found that the majority of interactors are shared between RBM20-WT, -P633L and -R634Q in the nucleus (Fig. 1l). Only 17 proteins mildly lost their ability to bind the two tested RBM20 variants (Supplementary Fig. 3d), four of which were identified in ref. 19 as enriched with WT protein (SRP14, RBM14, RBMX, and RBM15), and only one of them (RBMX) was a component of the spliceosome[34] (Supplementary Data 6). Pathway enrichment analysis of common interactors revealed enrichment of categories related to protein folding, mRNA metabolism, and splicing (Fig. 1m and Supplementary Data 2). These results indicate that RS-domain variants exhibit mainly unaltered interactors if located to the nucleus in HeLa cells.

In agreement with a published study[35], we identified MOV10 and PUM1 as interactors of RBM20 variants in the cytoplasm (Supplementary Fig. 3d and Supplementary Data 6). Proteins involved in spliceosome machinery, mRNA metabolism, and protein folding were enriched for binding to cytoplasmic RBM20 variants compared to WT (Fig. 1l and Supplementary Fig. 3d–f), suggesting their presence in RNP granules. The observed gain of interaction with spliceosome components could indicate that RBM20 in the cytoplasm sequesters other components of the splicing machinery. This may result in disruption of mRNA processing, in addition to the splicing defects observed in *RBM20* KO models[13]. Notably, since RBM20-WT presents predominantly nuclear localization, the amount of WT protein in the cytoplasm is relatively low (Fig. 1a and Supplementary Fig. 1b). This could result in an underestimation of its cytoplasmic interactors. Nevertheless, we found two factors that specifically interacted with RBM20-WT, and not with the mutants in the cytoplasm, namely TNPO3 (Transportin-3, Transportin-SR) and CHD1 (Fig. 1l and Supplementary Fig. 3d). This provides a potential insight into the nuclear import mechanism that we investigate in detail further.

### Restoring nuclear localization of RS-domain RBM20 variants rescues their splicing function

To investigate whether mislocalizing RS-domain variants of RBM20 are functional after relocalization to the nucleus, we applied a splicing reporter assay[18] in HEK293T cells (Fig. 2a). We analyzed multiple variants of *RBM20* with or without the nuclear localization sequence (NLS) of simian virus SV40 (Fig. 2b). Addition of the NLS resulted in

significant restoration of splice-regulatory activity for all tested RS-domain mutations (Fig. 2b). We also observed restored splicing of endogenous *IMMT*, an RBM20 target gene, in cells expressing NLS-tagged P633L and R634Q (Supplementary Fig. 4a–c). As a control, addition of the NLS to the V914A variant, which resides outside the RS domain and does not mislocalize to the cytoplasm[29], had no effect on splicing. These data suggest that RS-domain variants are splice-competent, once their nuclear localization is restored.

We validated these results in iPSC-CMs by lentiviral over-expression of WT-, R634Q-, or NLS-tagged RBM20-R634Q in splice deficient cells carrying the homozygous frameshift mutation (S635FS)[22]. Unlike RBM20-R634Q, overexpressed NLS-R634Q localized to the nucleus, similar to RBM20-WT (Fig. 2c). Genome-wide RNA-Seq analysis showed that the expression of 1751 genes was altered in cells expressing RBM20-R634Q compared to RBM20-WT (Fig. 2d and Supplementary Data 7). The genes were consistent with severe cardiac impairment (Supplementary Fig. 4d and Supplementary Data 2) and similar to the genes altered in expression in cells with cytoplasmic P633L (Supplementary Fig. 2b). A core RBM20 target gene expression was consistently either unchanged or downregulated, but never upregulated (Supplementary Fig. 4e, f). We found relatively few (115) differentially expressed genes between WT and NLS-tagged-R634Q expressing cells compared to R634Q cells (1713, Fig. 2d and Supplementary Data 7). Moreover, splicing of the core RBM20 targets was restored in NLS-R634Q expressing cells to similar levels seen in WT (80% AS events were restored from those mis-spliced in R634Q, Fig. 2e and Supplementary Data 8). Global splicing analysis in pairwise comparisons to RBM20-WT iPSC-CMs done with rMATS[33] identified fewer AS events in general, and exon-skipping events in particular, in NLS-R634Q compared to R634Q (total events 437 compared to 676, and exon skipping events 277 compared to 536, respectively, Supplementary Fig. 4g–i, Supplementary Data 9, see "Methods"). These findings suggest that nuclear relocalization of all tested RS-domain RBM20 variants may rescue splicing of *TTN*, and other core RBM20 targets[13]. However, this conclusion is based on ectopic expression of RBM20 variants, and further investigation is needed to characterize splicing restoration under physiological expression levels in cardiomyocytes. Nonetheless, these findings highlight the importance of identifying factors involved in the nuclear transport of RBM20.

### Genome-wide ICS screens identify TNPO3 as the major nuclear importer of RBM20

To identify factors that regulate subcellular localization of RBM20, we performed a genome-wide CRISPR/Cas9 screen in combination with the ICS technology[5] (Fig. 3a). We transfected HeLa cells expressing Tet::Cas9 and eGFP-RBM20-WT with a guide RNA (gRNA) library targeting 18,408 protein-coding genes with six gRNAs per gene. We collected the top 7% of cells exhibiting the most cytoplasmic (lower

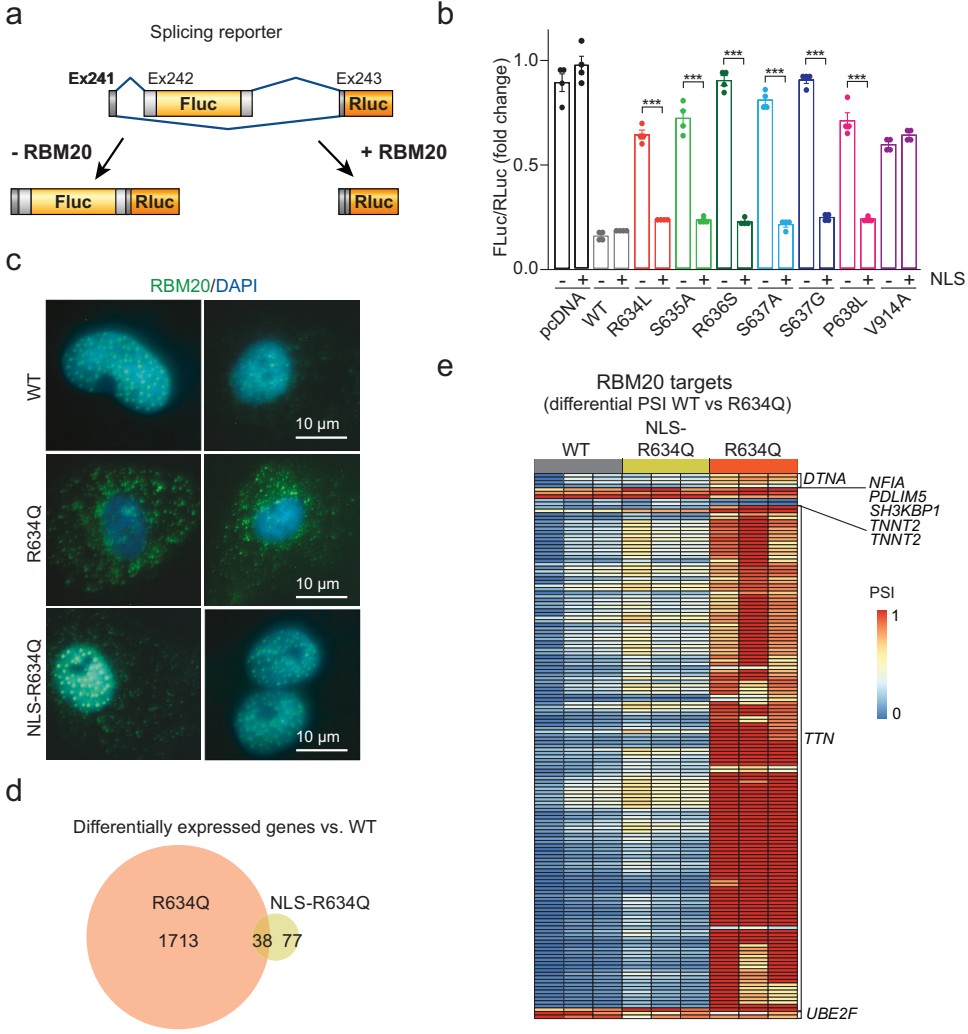

**Fig. 2 | Restoring nuclear localization of RS-domain RBM20 variants rescues their splicing function. a** Illustration of the splicing reporter assay. **b** Ratio of Fluc to Rluc in empty vector control-transfected (pcDNA) or +/− SV40 NLS RBM20 (WT, R634L, S635A, R636S, S637A, S637G, P638L, V914A)-transfected HEK293T cells. Data from one representative experiment is shown, a total of four experiments were analyzed. Each bar shows a mean of four technical replicates with standard errors indicated. **c** iPSC-CMs with a frameshift (S635FS) mutation transduced with eGFP-WT, eGFP-R634Q, or eGFP-NLS-R634Q, representative images, *n* = 3. **d** Numbers of significantly (| log2FC | > 1 and *P* < 0.1, Wald test with

Benjamini−Hochberg correction, DeSeq2[58]) differentially expressed genes in pair-wise comparisons of S635FS iPSC-CMs expressing eGFP-WT vs eGFP-R634Q and eGFP-WT vs eGFP-NLS-R634Q. Three biological replicates were analyzed. **e** PSI values for differential splicing events in the core RBM20 targets between eGFP-WT and eGFP-R634Q cells (|delta PSI| > 0.1, *P* value < 0.05, two-tailored *t*-test). Three independent iPSC-CM differentiations were used as biological replicates. Ns = not significant, *P < 0.05, **P < 0.01, ***P < 0.001, ****P < 0.0001, one-way ANOVA with Bonferroni post test, two-sided. Actual *P* values are shown in the source data file.

fraction) or the most nuclear (higher fraction) eGFP-RBM20 signal, as well as the unsorted input sample (input) (Supplementary Fig. 5a–d). We identified only one gene enriched in the lower fraction (*TNPO3*, positive regulator of RBM20 import), while 56 genes were enriched in the upper fraction (negative regulators) (FDR < 0.01) (Fig. 3b, c and Supplementary Data 10). Since RBM20-WT localization is exclusively nuclear, we assumed that negative regulators (gene knockouts (KOs) that induce a stronger nuclear translocation of RBM20) increased DRAQ5:RBM20 correlation by other means (e.g. changes in cellular or nuclear morphology[5]) and are not relevant for the scope of this study. Therefore, we discarded all negative regulators from further validation. In addition to *TNPO3*, we selected other hits with less significant FDR scores (*CLDN14, GALE, ADAMTS16, SLC29A2, CEBPB, UBQLNL, TRIM33, PMM2, TRIM24, IPPK, XPO6*, Supplementary Fig. 5e) and based on prior knowledge about their relevance for RBM20 function[18,36,37] (*TTN, AKT2, SPRK1, CLK1, LMNA*). We tested these potential positive hits by constructing single-gene KOs in HeLa cells and assessing their impact on RBM20-WT localization by ICS and fluorescence

microscopy. KOs of *GALE, CEBPB, TRIM33, TRIM24* mildly impaired nuclear RBM20 localization as measured by ICS (Fig. 3d and Supplementary Fig. 5f), but could not be validated by fluorescence microscopy (Fig. 3e and Supplementary Fig. 5g). However, KO of *TNPO3* resulted in a substantial shift in RBM20 localization and accumulation of the WT protein in the cytoplasm as determined by both ICS and microscopy analysis (Fig. 3d, e and Supplementary Fig. 5f, g). Importantly, TNPO3 was also one of the only two proteins we identified by mass spectrometry as interacting specifically with RBM20-WT and losing this interaction with mutants (Fig. 1l). We did not detect CHD1 as a positive regulator in the ICS screen.

To test whether any additional factors could retain RBM20 variants in the cytoplasm, we performed an ICS knockout screen in the RBM20-R634Q Tet::Cas9 HeLa reporter line (Supplementary Fig. 6a). This screen identified 151 genes, for which a single-gene KO increased nuclear localization of RBM20-R634Q (FDR <0.01). These are potential negative regulators of RBM20 nuclear import, e.g. sequestering mutant RBM20 in the cytoplasm (Supplementary

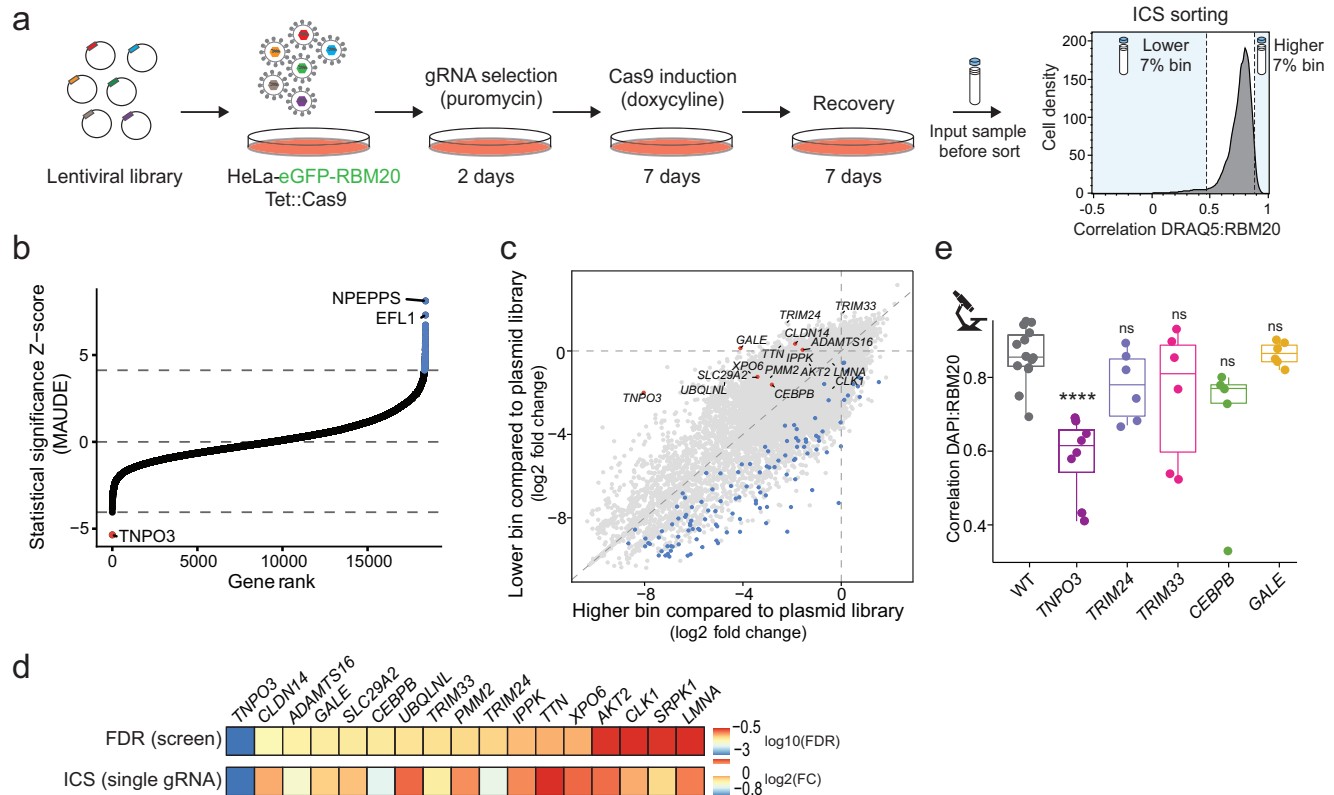

**Fig. 3 | Genome-wide ICS screen identifies TNPO3 as the main nuclear transporter of RBM20. a** Schematic outline of the ICS screen. Six genome-wide libraries were applied to HeLa cells expressing eGFP-RBM20-WT and Tet::Cas9, with 100 cells per gRNA coverage. Cells were sorted based on the correlation between RBM20 and DRAQ5 into 7% higher and 7% lower fractions at final coverage of 500 cells per gRNA per sorted bin. Unsorted input samples were collected too. **b** Reads were combined in silico to one dataset, and hits were called using MAUDE[69]. Genes are ranked by their statistical significance. The horizontal dashed lines indicate an FDR of 1%. Positive/negative regulators with FDR < 1% are marked in red and blue, respectively. **c** Scatter plot of fold changes visualizing gRNA abundance changes in higher (*x* axis) and lower (*y* axis) sorted bins compared with the plasmid library. Red and blue dots indicate statistically significant positive and negative regulators, respectively (FDR < 5% according to MAUDE). Labeled are positive regulators selected for future analyses. **d** The impact of single knockouts of the selected hits

(one gRNA per gene picked based on the strongest Z-score from the pooled screen) on RBM20 localization tested with ICS. The top row in the heatmap shows the log10(FDR) value for each candidate from the screen. The phenotype in the second row represents the standardized difference in RBM20 localization between the knockout (KO) and control cell populations (log2 of the ratio between cell fraction with Pearson coefficient (DRAQ5:RBM20) > 0.7 in the KO divided by cell fraction with Pearson coefficient (DRAQ5:RBM20) >0.7 in the control). **e** DAPI:RBM20 correlation quantified based on fluorescence microscopy analysis shown in Supplementary Fig. 5g, for the single KOs indicated. Each dot represents a Pearson correlation coefficient R for at least five cells, *n* = 3. Boxplots display quartiles Q1, Q2 (center), and Q3, with whiskers extending to the furthest data point within 1.5 times the IQR. Ns not significant, \*\*\**P* < 0.001, one-way ANOVA with Tukey's HSD post test (two-sided). Actual *P* values are shown in the source data file.

Fig. 6b, c and Supplementary Data 11). We then tested the effect of single-gene KOs to validate and further characterize the top candidates, including *DDI2, PPP5C, WTAP, EXOSC8,* and *SRSF3* (Supplementary Fig. 6d, e). Unlike KO of *TNPO3* that resulted in strong mislocalization of RBM20-WT to the cytoplasm (Fig. 3d, e and Supplementary Fig. 5f, g), perturbation of genes from the RBM20-R634Q screen led to only partial nuclear relocalization (Supplementary Fig. 6d, e). Importantly, expression of all but two (*CLDN14* and *FOXJ1)* tested hits from both screens was detected in iPSC-CMs by RNA sequencing, confirming the relevance of identified candidates (Supplementary Data 12).

Altogether, these data strongly suggest that, although other factors (like PPP5C, WTAP, and others identified in this study, Supplementary Data 11) might potentially have an impact on retaining mutant RBM20 in the cytoplasm, TNPO3 is the main and essential nuclear transporter of RBM20.

**Mislocalization of RS-domain RBM20 variants is caused by loss of interaction with TNPO3**

To assess the role of TNPO3 in the nuclear transport of RBM20 in iPSC-CMs, we performed siRNA knock-down (KD) and CRISPR/Cas9-based

KO of *TNPO3* (Fig. 4a, b and Supplementary Fig. 7a, b, d–f). We found that *TNPO3* KD significantly decreased nuclear localization of both RBM20-WT and -P633L. This was accompanied by a decrease of *TTN* and *IMMT* alternative splicing upon *TNPO3* KD (Fig. 4c and Supplementary Fig. 7c). The degree of RBM20-WT mislocalization correlated with TNPO3 levels upon *TNPO3* KO (Supplementary Figure 7g). Cells with decreased TNPO3 levels showed a significant shift in RBM20 localization to the cytoplasm compared to non-transfected cells (Supplementary Fig. 7h). This confirms that TNPO3 is essential for RBM20 nuclear import in iPSC-CMs.

We hypothesized that the disruption of the direct interaction between RBM20 and TNPO3 upon RS-domain mutations may be the main cause of RBM20 mislocalization in DCM. To assess the role of TNPO3 in the nuclear transport of RBM20 variants, we performed siRNA knock-down (KD) of *TNPO3* in HeLa cells expressing WT-, P633L-, R634Q-, or R634Q-S635E-S637E-RBM20 (RSS) variants (Supplementary Fig. 8a, b). In the RSS mutant, three out of six residues in the PRSRSP stretch are substituted, and it displayed the most severe defect in RBM20 nuclear localization (Fig. 4d, e). We found that *TNPO3* KD significantly decreased nuclear localization of WT, P633L, and even RBM20-R634Q, as measured by ICS (Fig. 4d) and confocal

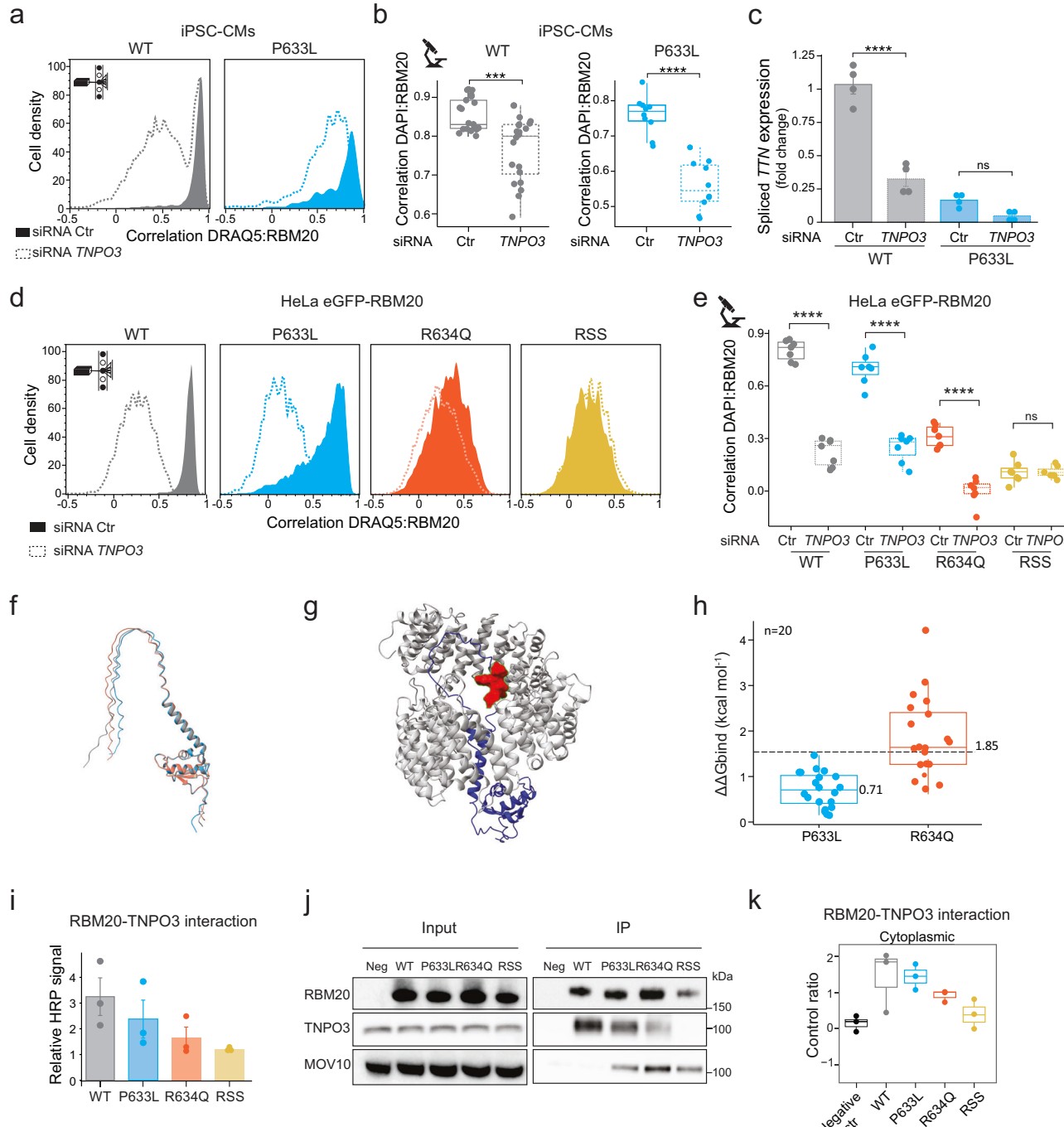

**Fig. 4 | Mislocalization of RS-domain RBM20 mutants is caused by loss of interaction with TNPO3. a** ICS-measured DRAQ5:RBM20 correlation in WT or P633L iPSC-CMs transfected with control (Ctr) or *TNPO3* siRNA. **b** DAPI:RBM20 correlation (based on Supplementary Fig. 7a). Each dot represents a Pearson coefficient for at least five cells, *n* = 3. **c** qPCR analysis of *TTN* exon 242 splicing-out in WT or P633L iPSC-CMs with Ctr or *TNPO3* siRNA normalized to *GAPDH* (mean fold change versus the RBM20-WT with control siRNA, with standard errors, two biological replicates with two technical replicates each). **d** ICS-measured DRAQ5:RBM20 correlation for HeLa expressing eGFP-WT-, -P633L-, -R634Q-, or -RSS-RBM20, with Ctr or *TNPO3* siRNA. **e** DAPI:RBM20 correlation (based on Supplementary Fig. 8a). Each dot represents a Pearson coefficient for at least five cells, *n* = 3. **f** Superimposed AlphaFold2 models of the RRM-RS domain of RBM20 (amino acid 511–673) as WT (gray), P633L (light blue), or R634Q (salmon). **g** Representative AlphaFold2 model of RBM20's WT RRM-RS domain (blue, amino acid 511–673) in complex with TNPO3 (gray, full-length, amino acid 1-923). The PRSRSP stretch (amino acid 633–638) in RBM20 is

highlighted as red spheres. **h** Predicted changes in binding affinity between TNPO3 and RBM20 upon RS-domain mutations. *N* = 20 AlphaFold models of the wild-type RBM20-TNPO3 complex were used to calculate the affinity change with a point mutation (see "Methods"). **i** Quantification of TNPO3 co-immunoprecipitating with RBM20 based on western blot analysis, representative image is displayed in (**j**) (*n* = 3, means with standard errors). **j** Western blot analysis of RBM20, TNPO3, and MOV10 in the cytoplasmic fraction of HeLa, and their co-immunoprecipitation with eGFP-RBM20 (Neg = negative no-bait control). **k** Quantification of TNPO3 peptides identified by mass spectrometry in the cytoplasmic fractions of co-immunoprecipitants with indicated cells, normalized to Neg, *n* = 3. Boxplots (**b**, **e**, **h**, **k**) display quartiles Q1, Q2 (center), and Q3, with whiskers extending to the furthest data point within 1.5 times the IQR. Ns = not significant, *P < 0.05, **P < 0.01, ***P < 0.001, ****P < 0.0001, Student's *t*-test for (**b**), and one-way ANOVA with Tukey's HSD post test for (**c**), (**e**), all two-tailored. Actual *P* values are shown in the source data file.

microscopy (Fig. 4e and Supplementary Fig. 8a). RSS localization did not change, as it was already fully cytoplasmic (Fig. 4d, e, Supplementary Fig. 8a). These data suggest that TNPO3 is responsible for localizing RBM20 variants to the nucleus, and its effectiveness is mutation-dependent.

To gain a deeper understanding of the RBM20-TNPO3 interaction, we used AlphaFold2[38,39] to predict the complex. We observed no structural rearrangements within the intrinsically disordered RS domain in the AlphaFold2 models of the RRM-RS (amino acid 511–673) domain from RBM20-WT, -P633L, or -R634Q proteins (Fig. 4f). The predicted complex of TNPO3 and RBM20's RRM-RS domain indicated that the PRSRSP region is classified as the interaction interface in all obtained models of TNPO3 with the isolated RRM-RS domain of RBM20 (Fig. 4g). This was also the case for the best-ranked prediction of TNPO3 in complex with full-length RBM20 (Supplementary Fig. 9a). All obtained predictions for RBM20's RS-domain had low pLLDT scores, which indicated the intrinsically disordered nature of this region. The obtained Predicted Aligned Error (PAE) plots of TNPO3 in complex with the RRM-RS domain of RBM20 suggest that both proteins are presumably incorrectly located relative to each other and are not ideal for isolated interpretation of a single model (Supplementary Fig. 9a). However, it has been shown that the RS-domain of another alternative splicing factor/splicing factor 2 (ASF/SF2) is the major contributor for the interaction with TNPO3 and that it can function as a transferable TNPO3-dependent NLS on its own[40–42]. We employed MutaBind2[43] to calculate the changes in the binding affinity induced by P633L and R634Q substitutions. We used the top 20 AlphaFold models of the wild-type RRM-RS domain in complex with TNPO3 to account for the uncertainty in the correct placement of the RS domain inside the binding pocket (Fig. 4h and Supplementary Fig. 9b). The predicted changes in the binding affinity induced by P633L and R634Q were both positive in comparison to the WT sequence of RBM20, which indicates a destabilization of the interaction and a decrease in the binding affinity with TNPO3. Mutabind2 generally classifies a single mutation as deleterious if the ΔΔG value is ≥1.5 kcal/mol. The R634Q mutation was classified as such in 13 out of 20 models (Fig. 4h). However, the P633L mutation did not impact the ΔΔG value strongly enough in any model, indicating that it should not fully interrupt the interaction with TNPO3.

To validate the structural predictions, we measured the level of TNPO3 co-immunoprecipitating with WT or mutated RBM20 in HeLa cells (Fig. 4i–k). The stability of the interaction with TNPO3 decreased in line with our predictions and the previously observed severity of RBM20 mislocalization (Supplementary Fig. 1b–d and Fig. 4i). This was observed by western blot (Fig. 4i, j) as well as by mass-spectrometry analyses (Fig. 4k). TNPO3 amount was constant between WT and RBM20-mutant cells, which rules out a potential effect due to differential expression of the transporter (Fig. 4j and Supplementary Fig. 8b, c). Our results indicate that the direct interaction between TNPO3 and RBM20 is essential for its nuclear import. The PRSRSP-mutations affect the stability of this interaction thereby resulting in mislocalization of RBM20 and aberrant splicing of RBM20 target genes.

Finally, we sought to address whether granule formation is a consequence or a cause of protein mislocalization. TNPO3 KD resulted in the accumulation of cytoplasmic granules of RBM20-WT in HeLa cells and iPSC-CMs (Supplementary Fig. 7a, f and Supplementary Fig. 8a), indicating that granule formation is not specific to the RBM20 variants. This finding is in agreement with our obtained AlphaFold models that showed no structural rearrangements within the RRM-RS domain of WT- or mutant RBM20 (Fig. 4f). The present study and previous work[35] have identified MOV10 as one of the main interactors of mutant RBM20 in the cytoplasm (Supplementary Figs. 3d and 8d and Fig. 4j). We analyzed RBM20-WT interaction with MOV10 in the cytoplasm upon TNPO3 KD. We observed a significant gain of interaction between RBM20-WT and MOV10 upon loss of

TNPO3 (Supplementary Fig. 8e, f). Moreover, we observed partial colocalization of MOV10 and RBM20-WT upon TNPO3 KD by confocal microscopy, similar to the mutant variants (Supplementary Fig. 8g). This suggests that loss of interaction with TNPO3 results in the formation of RNP granules detected for both WT and RS-domain mutated RBM20.

## Enhancing RBM20-TNPO3 interaction restores nuclear localization and splicing in vitro and in vivo

We tested whether enhancing the interaction between TNPO3 and RBM20 can rescue the aberrant localization and splicing deficiency caused by RS-domain mutations. We overexpressed TNPO3 in iPSC-CMs with RBM20-WT, -P633L, or -R634Q. While mislocalization of the P633L variant could be fully rescued by overexpressing TNPO3, R634Q mislocalization was only partially rescued (Fig. 5a, b and Supplementary Fig. 10a, b). This could be due to the higher severity of mislocalization seen for this variant (Fig. 1a–c) and lower binding affinity to TNPO3 (Fig. 4h–k). Both variants were able to form two characteristic nuclear foci when relocated to the nucleus, and the degree of their nuclear relocalization correlated with the level of TNPO3 expression (Fig. 5a and Supplementary Fig. 10a, b). Efficient restoration of nuclear localization of RBM20-P633L also resulted in rescue of TTN and IMMT splicing to the levels seen in RBM20-WT cells, while partial relocalization of RBM20-R634Q was associated with a proportional restoration of splicing function (Fig. 5c, d). These results indicate that mislocalization of the mutant variants can be rescued by upregulating the TNPO3–RBM20 interaction.

To test whether increasing TNPO3 levels could improve RBM20 mislocalization in vivo, we delivered Tnpo3 cDNA via AAV9 to mouse hearts bearing homozygous RBM20-P635L (P635L[+/+]) mutations (P633L in humans). We analyzed RBM20 localization and splicing function 4 weeks after AAV9 injection (Fig. 5e). Tnpo3 overexpression in vivo (Fig. 5f) resulted in partial rescue of RBM20 localization (Fig. 5g and Supplementary Fig. 10c) and Ttn splicing (Fig. 5h, i), independently of Rbm20 expression (Supplementary Fig. 10d). These findings reveal a novel therapeutic strategy for RBM20 variants that mislocalize to the cytoplasm.

In conclusion, our results show that increasing TNPO3 expression restores splicing and mislocalization of RS-domain mutant RBM20. Moreover, we provide the first proof-of-principle that this strategy can serve as a promising therapeutic avenue for developing future therapies for RBM20-mediated DCM.

## Discussion

DCM-causing variants in the RS domain of RBM20 have been shown, by our group and others, to result in aberrant RBM20 localization and RNP granule formation in the cytoplasm[24–31]. These mutations are associated with a more severe disease phenotype than an RBM20 KO in vivo[25,26]. Prior to this study, it was unknown why these single point mutations lead to RBM20 protein mislocalization and whether restoration of nuclear localization could restore the splicing activity in vivo. Here, we demonstrate that all tested mislocalizing RBM20 RS-domain variants regain their splicing activity upon addition of the SV40 NLS tag, as previously shown for S635A[44]. Future studies with endogenously expressed RBM20 variants will identify mRNA targets and protein interactors of different relocalized RS-domain variants. In addition, ultrastructural studies of cytoplasmic foci caused by different RS-domain variants could elucidate the functional differences of these foci.

We identify TNPO3 as the main nuclear importer of RBM20 with the genome-wide ICS CRISPR/Cas9 screen and validate its essentiality for nuclear import of endogenous RBM20 in iPSC-CMs. TNPO3 belongs to the β-karyopherin family of nuclear import receptors, which directly interact with their cargo via amino acid sequence recognition. TNPO3 specifically recognizes arginine-serine (RS)

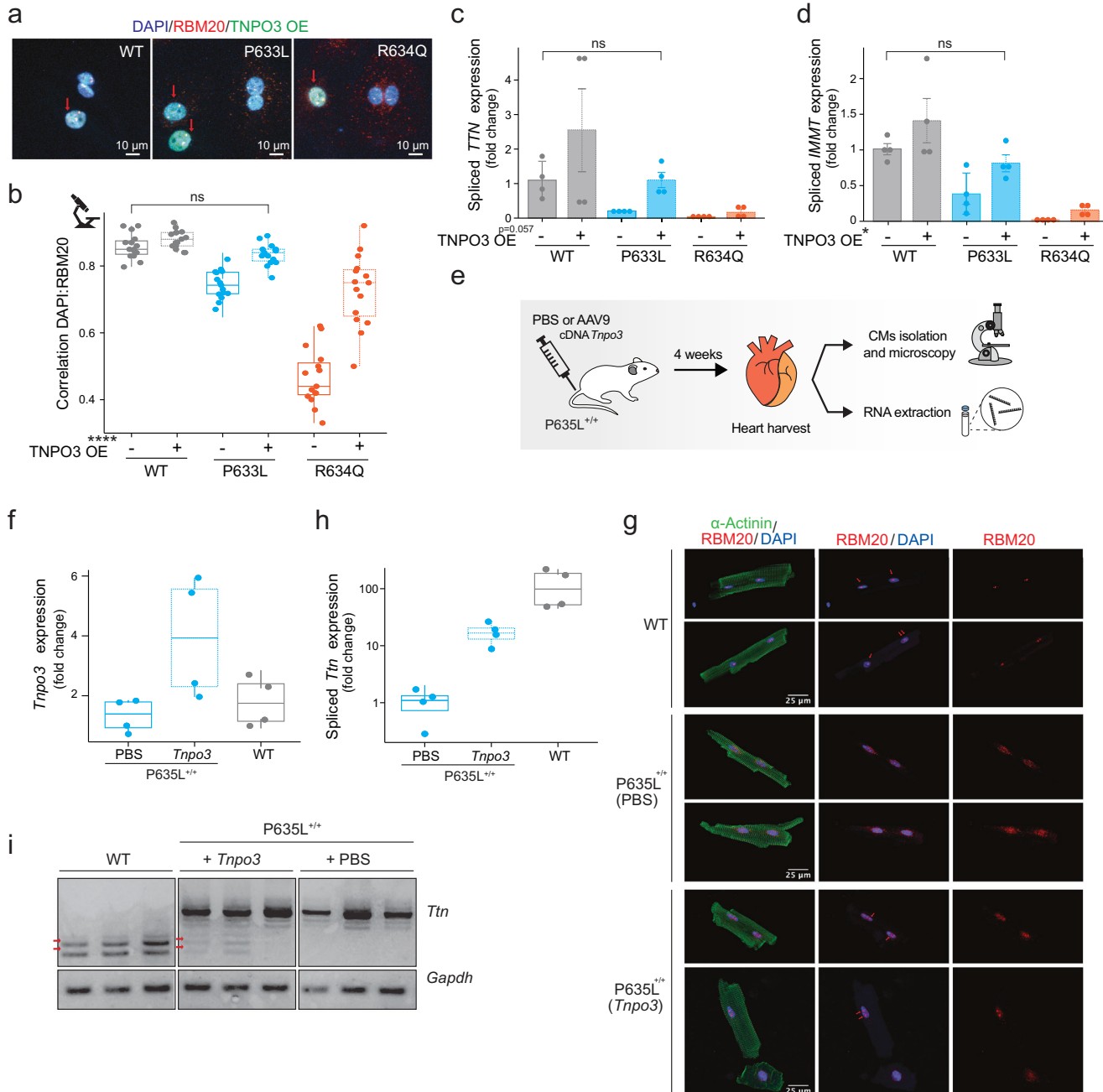

**Fig. 5 | Enhancing RBM20-TNPO3 interaction restores nuclear localization and splicing in vitro and in vivo. a** Representative images of RBM20 localization upon overexpression (OE) of eGFP-TNPO3 in iPSC-CMs. Red arrows point at cells transduced with eGFP-TNPO3, *n* = 3. **b** DAPI:RBM20 colocalization based on the data shown in (**a**). Each dot represents a Pearson coefficient for at least five cells, *n* = 3. TNPO3 overexpression effect's *P* value < 0.0001. **c** qPCR analysis of *TTN* exon 242 splicing (TNPO3 overexpression effect's *P* value = 0.057); and **d** *IMMT* exon 6 splicing (TNPO3 overexpression effect's *P* value = 0.015) in iPSC-CMs. Isoform expression was normalized to *GAPDH* and displayed as fold change versus the WT line (first replicate) without TNPO3 OE (means with standard errors, two biological replicates with two technical replicates for each). **b–d** TNPO3's overexpression effect's *P* values were calculated with Two-way ANOVA with two-tailed Tukey's HSD post test; comparison of WT vs P633L + TNPO3 - two-tailed *t*-test. **e** Scheme of *Tnpo3* OE in vivo. **f** qPCR analysis of

*Tnpo3* expression in *n* = 4 WT mice, *n* = 4 P635L[+/+] mice injected with PBS, or *n* = 4 P635L[+/+] mice injected with *Tnpo3*. Data is normalized to *Gapdh* expression and displayed as fold change versus one of the WT mice with standard errors. **g** Representative images of RBM20 localization in WT mice, P635L[+/+] mice injected with PBS, or P635L[+/+] mice injected with *Tnpo3*. Red arrows point at characteristic nuclear foci formed by RBM20, *n* = 4 mice. **h** qPCR analysis of *Ttn* splicing in *n* = 4 WT mice, *n* = 4 P635L[+/+] mice injected with PBS, or *n* = 4 P635L[+/+] mice injected with *Tnpo3*. Data is normalized to *Gapdh* expression and displayed as fold change versus one of the WT mice with standard errors. **i** RT-PCR of *Ttn* splicing and *Gapdh* expression in *n* = 3 WT mice, *n* = 3 P635L[+/+] mice injected with PBS, and *n* = 3 P635L[+/+] mice injected with *Tnpo3*. Red arrows point at *Ttn* isoforms expressed in WT mice. Each boxplot of this figure (**b**, **f**, **h**) displays quartiles Q1, Q2 (center), and Q3, with whiskers extending to the furthest data point within 1.5 times the IQR. Actual *P* values are shown in the source data file.

repeats that are present in many splicing factors like SRSF1, ASF/ SF2, and others[40–42,45–47]. Here we show that mutations in the RS domain of *RBM20* diminish the interaction with TNPO3. The RBM20 RS-domain variants that abolish the interaction the most showed simultaneously the strongest mislocalization. ICS-based genetic screens in iPSC-derived cardiomyocytes might reveal additional regulators in future studies, e.g. additional factors involved in RBM20 nuclear import or in regulating post-translational modifications of RBM20. It has been observed that the binding of TNPO3 to its cargo can be phosphorylation-dependent[40–42] or independent[46]. In the case of RBM20, serine residues in the RS domain are normally phosphorylated[26,35,37,44], however, phosphomimetic amino acid substiutions do not rescue the localization phenotype[26]. Importantly, both pooled and individual CRISPR KOs of kinases *AKT2*, *CLK1*, and *SPRK1*—previously shown to phosphorylate RBM20[37]—did not impact RBM20 localization (Fig. 3d). This result suggests that either the kinases complement each other, or that the RBM20-TNPO3 interaction is potentially phosphorylation-independent.

We further used in vitro and in vivo models to demonstrate that overexpression of TNPO3 can improve the nuclear import of RBM20 variants and restore splicing deficiency. Importantly, we observed that cytoplasmic granules were reduced after enhanced nuclear transport. Previous studies have shown that elevated expression of nuclear import receptors disperses aggregates formed by mutated RNA-binding proteins linked to neurodegenerative diseases and redirects them to the nucleus[48]. This is in line with our findings and would suggest that upregulating the interaction of a nuclear import receptor that specifically recognizes a mislocalized protein could be used as a therapeutic strategy in other diseases as well. Further investigations will be needed to understand the nature and the effect of RBM20 granule formation. Our data provide the first evidence that cytoplasmic granule formation of RBM20 variants is the result of mislocalization, and not its cause. We show that RBM20-WT, when forced to remain in the cytoplasm, also forms similar granules as the mutant variants. Although the exact nature of RNP granules formed by WT or RS-domain mutated RBM20 in cardiomyocytes needs to be addressed in future studies, our results suggest that RS-domain mutations of *RBM20* do not confer pro-aggregative qualities.

Mutations in *TNPO3* have been linked to impaired myogenesis[47] and myopathies[49,50], which includes one mutation in *TNPO3* that was linked to familial DCM[51]. Detailed structural studies will be needed to further decipher the RBM20-TNPO3 interaction and the direct impact of different mutations on both partners.

Altogether, our data reveal a new therapeutic avenue for DCM patients with disease-causing variants in the RS-rich region of *RBM20*. Since the majority of TNPO3 targets[52] were detectably expressed in iPSC-CMs (Supplementary Data 13), direct overexpression of *TNPO3* may affect their nuclear import resulting in potential side-effects of therapeutic *TNPO3* overexpression. Enhancing RBM20 nuclear import could be achieved by other means, like endogenous tagging of *RBM20* with another NLS to be recognized by other importins. Alternatively, aptamers, bifunctional antibodies, or small molecule drugs that bind allosteric sites in RBM20, could be explored as therapeutic strategy that increases the affinity of mutant RBM20 to TNPO3. Our data demonstrate that these actions can alleviate the known splicing deficiency and abolish cytoplasmic granule formation via enhanced nuclear import.

## Methods

### Ethical statement
Induced pluripotent stem cells used within this study were obtained from the Stanford Cardiovascular Institute Biobank, generated and characterized previously[22]. All experiments with human cells performed within this study conformed to the EMBL Guidelines and were approved by the Bioethics Internal Advisory Committee (BIAC).

All animal care and procedures performed in this study conformed to the EMBL Guidelines for the Use of Animals in Experiments and were reviewed and approved by the Institutional Animal Care and Use Committee (IACUC).

The patient data collection was approved by the independent internal review board (IRB) at Stanford University, Johns Hopkins University, the University of Michigan, Children's Hospital of Atlanta, the University of Pennsylvania, University Hospital Zurich, the University of British Columbia, and the University of Heidelberg. Patient consent was obtained as required by each individual institution.

### Patient data
A kindred with four members (75% males, mean age 42 ± 2.8 years) carrying the P633L variant in *RBM20* was identified through cascade screening. The first identification of this variant of uncertain significance was found in a proband with dilated cardiomyopathy and heart failure in the seventh decade of life. Given ascertainment bias associated with comparing variant carriers identified by family screening to probands who generally present with more severe disease, we compared these four RBM20-P633L relatives to individuals with pathogenic and likely pathogenic variants in the RS domain identified through family screening only (i.e., non-probands, $n = 15$, 42% male, mean age 43.4 ± 0.5 years). To assemble this cohort, we identified genotype-positive family members who were diagnosed after family screening from 9 contributing inherited cardiomyopathy centers (Stanford Center for Inherited Cardiovascular Disease, the University of Heidelberg, the University Hospital Zurich, Johns Hopkins University, Brigham and Women's Hospital at Harvard Medical School, the University of Pennsylvania, the University of Michigan, University of British Columbia, and Children's Hospital Atlanta). Identifying patients with specific rare genetic diagnoses requires collaboration between large referral centers equipped with genetic testing and counseling capability in addition to highly specialized cardiac care. Because of the rarity of RBM20 cardiomyopathy in the population, we collected all identified cases and were unable to respecify sample size. As it is true for many rare disease cohorts, the data may be biased with respect to the socioeconomic status and family screening adherence observed in patients who can pursue care at quaternary referral centers in the United States. Data on age at presentation and initial echocardiogram were collected retrospectively by chart review. The study was approved by the independent internal review board (IRB) at each site, and patient consent was obtained as required by each individual institution.

### Cell culture
**Induced pluripotent stem cells (iPSCs)**. The iPSC lines used in this study were previously generated and characterized in ref. 22. iPSCs were cultured in monolayer in cell culture dishes coated for 1 h at room temperature with Vitronectin (VTN-N) Recombinant Human Protein, Truncated (Gibco, A14700). Cells were cultivated in the E8-Flex medium (Gibco A2858501) and split twice per week using Versene solution (Gibco, 15040066). For splitting, cells were washed once with PBS and incubated with Versene for 5–10 min at room temperature. After that, Versene was aspirated, and cells were resuspended in fresh E8-Flex medium, and re-plated at the desired concentration. Freezing and thawing of cells were done in the presence of RevitaCell (1:100) supplement (Gibco, A2644501). Cryopreservation of cells was done in culture medium supplemented with 1% RevitaCell and 10% DMSO. The EMBL Ethics Committee approved the study protocol for iPSCs.

**Cardiomyocyte differentiation**. For cardiomyocyte differentiation, iPSCs were cultured as monolayer, and Wnt signaling was modulated as previously described[53]. Briefly, iPSCs were plated at low confluency on vitronectin-coated plates to reach 70–80% 4 days post plating. On day 0, the medium was changed to RPMI 1640 (Gibco 21875034) + B27-

insulin supplement (Gibco, A1895601) (RPMI + B27-ins) with the addition of 4 µM CHIR 99021 (LC Laboratories C-6556) in DMSO. On day 1, the medium from the day before was diluted by the addition of an equal volume of RPMI + B27-ins. On day 3, medium was changed to RPMI + B27-ins with the addition of 2 µM Wnt-C59 inhibitor (Tocris 5148) in DMSO. On day 5 and day 7, the medium was changed to RPMI + B27-ins without any additions. On day 9, the medium was changed to RPMI + B27 supplement (Gibco, 17504044) (RPMI + B27). On day 11, the medium was changed to RPMI 1640 with no glucose (Gibco, 11879020) with the addition of 0.1% sodium DL-lactate (L4263-100 ml, Sigma). On day 14, the medium was changed back to RPMI + B27. After day 16, the cells were passaged every 2 weeks on VTN-N coated plates until harvested for downstream analyses. For passaging, cells were washed 1 time with PBS and incubated in TrypLE Select Enzyme (10X) (Gibco A1217701) at 37 °C for 10–15 min. After that, cells were resuspended in four times volume of Passaging medium, composed of RPMI + B27 supplemented with 10% knockout serum replacement (Gibco 10828010) and 1.6 µM Thiazovivin (Stem Cell Technologies, 72252). Cells were pelleted at $350 \times g$ for 5 min, and plated in fresh medium on pre-coated VTN-N plates. Medium was changed on the next day to RPMI + B27, and was then changed twice per week.

**HeLa and HEK293FT culture.** HeLa Tet::Cas9 eGFP-RBM20 cells (engineered using HeLa Tet::Cas9 provided by Paul Blainey (Broad Institute), characterized in[54] and generated in Iain Cheeseman's lab (MIT)) and HEK293FT (Thermo Fisher Scientific) were maintained in DMEM, high glucose (Gibco, 11965084) supplemented with 10% FBS Supreme (Pan Biotech, P30-3031), 1% Sodium Pyruvate (Gibco, 11360070), and 1% Penicillin–Streptomycin (Gibco, 15140122). For splitting, cells were washed 1 time with PBS, and incubated with Trypsin-EDTA (0.25%) (Gibco 25200056) for 5 min at 37 °C. After that, cells were resuspended in the fresh medium, diluted to the desired concentration, and plated. Cryopreservation of cells was done in the culturing medium supplemented with 10% DMSO.

**Lentivirus production**
HEK293FT cells were grown to 80–90% confluency, and transfected using Lipofectamine 3000 reagent (Thermo Fisher Scientific) with two lentiviral packaging plasmids (pMD2.G and psPAX2), and a plasmid carrying the gene of interest mixed at 1:1:1 ratio to obtain in total 2.5 µg of DNA per well of a six-well plate. Six hours later, cells were split in -1:6 ratio (one well of a six-well plate into a 10-cm tissue culture dish), and cultured at 37 °C. Three days later, the supernatant was collected, and the remaining cell debris was filtered through the 0.45-µm filter. The filtered supernatant was then incubated with the addition of 1:3 of its volume of LentiX concentrator (Takara/Clontech) at 4 °C for at least 1 h, to a maximum of overnight, followed by centrifugation at 4 °C, $4000 \times g$, for 45 min. The supernatant was aspirated, and the viral pellets were resuspended in PBS in 1:200 of the initial supernatant volume. The virus was aliquoted and stored at −80 °C until usage.

**Engineering of HeLa reporter cell lines**
TetO-eGFP-GGSG-NLS-Flag-RBM20 plasmid (see full sequence for RBM20-WT in Supplementary Data 14) was cloned via site-directed mutagenesis-based insertion of the SV40 NLS sequence (ccaaaaaa-gaagagaaaggta) into the TetO-eGFP-GGSG-FLAG-RBM20 plasmid (see oligos in Supplementary Table 1) using GeneArt Site Directed Mutagenesis Kit (Thermo Fisher Scientific). The latter was cloned via Gibson assembly of the eGFP-GGSG cDNA, Flag-RBM20 cDNA (GenScript), as well as the fragment of TetO-lenti backbone (gift from Moritz Mall lab, DKFZ, Germany), amplified with the oligos listed in Supplementary Table 1, purified via gel extraction kit (Qiagen), and incubated with Gibson assembly master mix (NEB) at 50 °C for 1 h, followed by

transformation into 5-alpha *E. coli* (NEB), plasmid isolation, and sequence confirmation by Sanger sequencing. The single-point mutations were introduced into *RBM20* cDNA sequence using GeneArt Site Directed Mutagenesis Kit (Thermo Fisher Scientific) and oligos listed in Supplementary Table 1.

For the NLS-rescuing experiment, HeLa Kyoto cells were co-transduced with TetO-eGFP-FLAG-NLS-RBM20-WT, -P633L, or -R634Q, as well as rtTA (Addgene 20342), cultured in the presence of 2 µg/ml of Doxycycline (Sigma) for at least 7 days, single-cell sorted for eGFP fluorescence with FACS, and used for the experiment at least 2 weeks after the sort.

pEFa-eGFP-GGSG-Flag-RBM20 (see full sequence for RBM20-WT in the Supplementary Data 14) plasmids were cloned via Gibson assembly (NEB) of the amplified TetO-backbone (see above), eGFP-GGSG-Flag-RBM20 cDNA from the TetO-plasmids (described above), and pEFa promoter sequence from the Addgene #125592 plasmid (see oligos in Supplementary Table 1).

HeLa Tet::Cas9[54] were transduced with lentivirus delivering pEFa-eGFP–Flag-RBM20-WT, P633L, R634Q, or R634Q-S635E-S637E in six-well plates by adding 20 µl of 100x concentrated virus per well, and single-cell-derived colonies were obtained by FACS. Two weeks after single-cell sorting, the established lines were further analyzed for their purity with FACS and ICS, and the most stable and pure clonal lines were used for downstream applications.

**Cell treatments**
**siRNA transfection.** HeLa Tet::Cas9[54] cells were cultured until they reached a confluency of 20% in medium without Penicillin–Streptomycin, and transfected using Lipofectamine RNAiMAX reagent (Thermo Fisher Scientific) and 20 nM of either nontargeting control siRNA (D-001810-02), or siRNA targeting *TNPO3* (L-019949-01) (Dharmacon, Horizon Discovery Group). Medium was replaced with the normal HeLa culture medium described above 24 h later, and cells were harvested for downstream analyses 72 hours post transfection as described above.

Differentiated iPSC-CMs were cultured at 70–90% confluency in RPMI + B27 medium, and transfected using LipofectamineStem reagent (Thermo Fisher Scientific) and 100 nM of either control siRNA or siRNA targeting *TNPO3*. Medium was exchanged to fresh RPMI + B27 medium 24 h later. Four days post transfection, cells were transfected again in the same way, changing medium 24 h afterwards. After 72 h post the second transfection, cells were harvested for downstream analyses as described above.

***TNPO3* overexpression in iPSC-CMs.** *eGFP-TNPO3* cDNA was purchased from Addgene (167590), and lentivirus was produced as described above. iPSC-CMs were cultured at 70–90% confluency in RPMI + B27 medium, and transduced with 1:1000 volume of the virus. Medium was changed to a fresh RPMI + B27 24 h after transduction, and again, 3 days later. Seven days post transduction, cells were either fixed for microscopy analysis (see below), or FACS sorted in bulk into 1.5 ml reaction tubes for further RNA extraction (see below), to directly compare eGFP-TNPO3 positive and negative cells.

**Overexpression of RBM20 variants in iPSC-CMs.** TetO-eGFP-FLAG-RBM20-WT, -R634Q, or NLS-R634Q were cloned and packaged into lentiviral particles, as described above. iPSC-CMs with a frameshift mutation in *RBM20*'s RS domain[22] were cultivated at 70–90% confluency in RPMI + B27 medium, and co-transduced with 1:1000 volume of the virus delivering TetO-RBM20, as well as rtTA (Addgene 20342). 24 h later, the medium was changed to a fresh RPMI + B27 with the addition of 2 µg/ml of Doxycycline (Sigma). Medium was changed again 3 days later to RPMI + B27 with Doxycycline, and after a total of 7 days post infection, cells were harvested for downstream analyses as described above.

**TNPO3 KO in iPSC-CMs.** For the constitutive Cas9 cell line, a transgene carrying a CAG-Cas9-P2A-dTomato cassette was inserted into the AAVS1 locus in WTC-11 cells (Coriell Institute for Medical Research− GM25256) as described in ref. 55. In short, cells were nucleofected using the 4D-Nucleofector nucleofector (Lonza) using the P3 Primary Cell 4D-Nucleofector® X Kit L (V4XP-3024) and pulsed with program CB-150, grown for 7 days, subsequently sorted for dTomato expression and plated as single cells. Colonies were picked and genotyped to confirm the transgene presence. Cells were functionally tested for Cas9 activity using control gRNAs described in ref. 55, homogenous dTomato signal using flow cytometry and chromosomal integrity using the Infinium CoreExome-24 v1.4. Kit (Illumina).

RBM20-WT iPSCs with constitutive Cas9 were differentiated into iPSC-CMs as described above. After 17 days, they were plated onto the 24-well microscopy plates. On day 18, they were transfected with a mix of three sgRNAs targeting *TNPO3* (500 ng of each, see sequences in Supplementary Table 2), using LipofectamineStem reagent (Thermo Fisher Scientific). On the next day, medium was changed to fresh RPMI + B27. Four days post transfection, cells were transfected again in the same way, changing medium 24 h afterward. After 72 h post the second transfection, cells were fixed with 4% FA and stained as described below. Formaldehyde fixation abolished fluorescent signal from constitutive Cas9-dTomato as indicated in unstained TNPO3 control (Supplementary Fig. 7d).

### Cell staining
For viability staining, cell suspensions in PBS were stained with 50 μg/ml final concentration of DAPI (Thermo Fisher Scientific). For staining of nuclei of live cells, 100 mM of DRAQ5 (Biostatus) was added to the cell suspension in PBS at room temperature. Cells were analyzed with FACS or ICS not earlier than 5 min after the addition of DAPI or DRAQ5.

**Antibody staining for ICS.** For ICS measurement of endogenous RBM20 localization, iPSC-CMs were harvested with TrypLE Select Enzyme (10X) (Gibco A1217701) (see above), resuspended in Passaging medium (see above), washed once with PBS, and fixed with 4% PFA in PBS at room temperature (RT) for 10 min. Then, cells were washed once with PBS, and permeabilized with 0.1% Triton X-100 (Merck) in 1% BSA (Sigma) in PBS for 5 min at RT. Then, cells were incubated with 1:100 dilution of anti-RBM20 (ab233147, Abcam) antibody in 1% BSA in PBS for 1 h at RT, followed by a wash in PBS. Cells were then incubated with 1:500 dilution of AlexaFluor488 goat anti-rabbit antibody (Invitrogen) in 1% BSA for 30 min at 4 °C in the dark. After this, cells were washed once with PBS, and resuspended in PBS containing DAPI and DRAQ5 at the concentrations described above.

**Antibody staining for microscopy.** Cells were cultured either in glass bottom plates, or on coverslips, for microscopy analysis, following cell culture conditions described above. Cells were washed once with PBS, fixed with 4% PFA in PBS for 10 min at RT, washed once with PBS, and permeabilized with 0.5% Triton X-100 (Merck) in PBS for 5 min at RT. Then, the potential nonspecific antibody binding sites were blocked by incubation with 2% BSA (Sigma) in PBS for 1 h at RT. Then, cells were incubated with 1:250 dilution of primary antibodies (anti-RBM20 for human cells−ab233147, Abcam; anti-RBM20 for mouse cell staining−PA5-58068, Invitrogen; anti-sarcomeric alpha-actinin−ab9465, Abcam; anti-MOV10−PLA0195, Sigma; anti-TNPO3 - ab54353, Abcam) in 1% BSA in PBS at 4 °C overnight (or 1 h RT for MOV10 staining). Cells were then washed three times with 2% BSA in PBS at RT, and incubated with secondary AlexaFluor antibodies (Invitrogen, Supplementary Table 3) at 1:500 dilution in 1% BSA in PBS for 1 hour at RT in the dark. Cells were then washed two times

with 2% BSA in PBS at RT in the dark, and incubated in 2 μg/ml Hoechst 33258 (Invitrogen) in PBS for 10 min at RT in the dark for cells cultured on the microscopy plates, or with just PBS for 10 min at RT in the dark for coverslips. Microscopy plates were then washed once with PBS and stored at 4 °C in the dark until imaged. Coverslips were mounted with ProLong Gold antifade reagent with DAPI (Invitrogen) and stored at 4 °C in the dark until imaged.

### FACS and ICS
For all FACS and ICS applications, after being stained, cells were filtered through a 35-μm cell strainer to avoid clumping, and kept on ice until analyzed.

For single-cell FACS sorting, cells were sorted based on the desired fluorophore expression into 96-well plates containing culture medium, one cell per well. For bulk FACS sorting, cells were sorted into 1.5-ml microcentrifuge tubes containing DMEM with 10% FBS, based on the desired fluorophore expression. For all sterile sorts, BD FACSAria™ Fusion was used, using a 100 μm sort nozzle. Viability staining with DAPI was used to sort out dying cells.

For routine checking of transfection/transduction efficacies, as well as for optimizing experimental conditions, BD LSRFortessa™ was used.

Image-enabled cell sorting (ICS) used the BD CellView™ Imaging Technology as previously described[5]. ICS experiments were performed with a 100 μm sort nozzle, with the piezoelectric transducer driven at 34 kHz, automated stream setup by BD FACSChorus™ Software, and a system pressure of 20 psi. All sorts were performed in purity mode.

For correlation-based sorts and measurements, cells were stained with DRAQ5, and for each cell, a Pearson correlation coefficient R was calculated based on the overlap between RBM20 and DRAQ5.

Flow cytometry and ICS data were analyzed using FlowJo_v10.7.1_CL software.

An example of gating strategy used for FACS and ICS-based sorts is added as Supplementary Fig. 11.

### Microscopy
Widefield fluorescence microscopy analysis was performed using Zeiss Cellobserver microscope equipped with an AxioCam camera, using Plan-APOCHROMAT 20x NA0.8 Air DIC2 or LD Plan-NEOFLUAR 40x NA0.6Air Ph2 correction collar 0−1.5 objectives.

Confocal microscopy analysis was performed using Zeiss LSM900 microscope equipped with 405 nm−5 mW, 488 nm−10 mW, 561 nm−10 mW, 640 nm−5 mW lasers, using Objective Plan-Apochromat 40x/0.95 Corr M27 air (FWD = 0.25 mm) objective, and 3 Gallium Arsenide Phosphid-PMT (GaAsP-PMT) for fluorescence detection, standard PMT as transmission detector.

**Colocalization analysis.** For colocalization analysis of RBM20 and DAPI, Fiji (v.2.1.0/1.53c) plugin Coloc2 was used to reflect Pearson correlation coefficient R. For confocal images, Z-stack images were max projected, and fluorescent channels were split. The area covering at least five cells was selected in RBM20 channel, and was used as ROI/mask for quantification of its correlation with DAPI channel.

### RNA extraction
**Live cells.** Cells cultured in tissue culture dishes were washed three times with PBS, lysed in TRIzol (Invitrogen), and transferred to 1.5-ml reaction tubes. RNA extraction and DNAse-I treatment were performed using the Direct-zol RNA Miniprep Plus Kit (Zymo Research), according to the manufacturer's instructions.

**Fixed cells.** Fixed iPSC-CMs (at least 5000 cells per sample) were pelleted by centrifugation at $500 \times g$ for 3 min and resuspended in 16 μl of a 1:16 proteinase K in PKD buffer (Qiagen), incubated at RT for 5 min,

briefly spun down, and incubated at 56 °C for one hour. The solution was then resuspended in 100 μl of TRIzol LS Reagent (Thermo Fisher Scientific). Then, 20 μl of chloroform was added to the TRIzol-sample, and phase separation was achieved at RT by vigorous shaking and centrifugation at 15,000 × g for 5 min. From each sample, 40 μl of the aqueous phase were collected, transported to a new Eppendorf tube, and mixed with 75.5 μl of isopropanol, and 1:150 of glycoblue (Invitrogen). The samples were then left at −80 °C for 24–36 h to dehydrate. RNA was then pelleted by centrifugation at 4 °C at maximum speed for 15 min. The supernatant was removed, and the pellet was washed once with 70% ethanol. The pellet was then air dried, and resuspended in 8 μl of nuclease-free water.

**Tissue.** A piece of left ventricle was homogenized in PBS, spun down, and the pellet was then resuspended in TRIzol (Invitrogen). This was followed by RNA extraction and DNAse-I treatment using the Direct-zol RNA Miniprep Plus Kit (Zymo Research), according to the manufacturer's instructions.

RNA concentration was measured using Qubit High Sensitivity RNA kit (Thermo Fisher Scientific), according to the manufacturer's instructions.

## Quantitative RT-PCR

For cDNA synthesis, the SuperScript IV (Thermo Fisher Scientific) kit was used according to the manufacturer's instruction, with addition of 0.5 mM of each dNTP (NEB), 2.5 μM oligo-dT (Thermo Fisher Scientific), 1.25 μM random hexamer primers (Invitrogen), 5 mM DTT (Thermo Fisher Scientific), 2 u/μl RNAse inhibitor (Invitrogen), 1X SSIV buffer (Thermo Fisher Scientific), and 10 u/μl SSIV RT (Thermo Fisher Scientific), per each reaction. At least 10 ng of total RNA was used per reaction, but not more than 1 μg.

A one-step qPCR reaction (95 °C for 10 min, 40 cycles of [95 °C 15 s, 60 °C 1 min]) was performed using SYBR Green Master Mix (Thermo Fisher Scientific), and primers listed in Supplementary Table 4, using Applied Biosystems StepOnePlus Real-Time PCR System (272006365), and StepOne Software v2.3. Delta-delta cT values were quantified versus *GAPDH* as a housekeeping gene, and versus a control sample for each experiment.

## RNA sequencing

**Library preparation and sequencing.** Prior to library preparation, RNA quality was checked using the 2100 RNA pico Bioanalyzer (Agilent) kit, and 1–10 ng of RNA were used as input.

To prepare RNA-sequencing libraries from ultra-low-input and highly degraded RNA (RIN 1–7) extracted from fixed samples of iPSC-CMs (see above), SMARTer RNA-Seq Kit v3-pico (Takara Bio) was used, according to the manufacturer's instructions. The fragmentation step was omitted. Prepared libraries with unique dual index barcodes for each sample were double-checked on Bioanalyzer (Agilent), and pooled together at equimolar concentrations, with six libraries per pool. Each pool was sequenced individually, with final concentrations of 8–10 ng/ml for each pool. For each pool, a 2.1 pM solution was loaded on the Illumina sequencer NextSeq 500 and sequenced bi-directionally, generating ~500 million paired-end reads, each 75 bases long. Obtained reads were then demultiplexed based on the unique dual barcodes into separate fastq files. After demultiplexing, each sample had between 50,000,000 and 100,000,000 reads in total. Quality control of the sequencing data was done using FASTQC (v0.11.5).

Library preparation for HeLa cells was done based on RNA extracts from live cells (see above).

Barcoded stranded mRNA-seq libraries were prepared from 150 ng of high-quality total RNA samples using the NEBNext Poly(A) mRNA Magnetic Isolation Module and NEBNext Ultra II Directional RNA Library Prep Kit for Illumina (New England Biolabs (NEB), Ipswich,

MA, USA) implemented on the liquid handling robot Beckman i7. Obtained libraries that passed the QC step were pooled in equimolar amounts; 2.1 pM solution of this pool was loaded on the Illumina sequencer NextSeq 500 and sequenced bi-directionally, generating ~500 million paired-end reads, each 75 bases long.

**Data analysis.** Reads were aligned to GRCh38.101 using STAR[56] (v.2.7.5c), and bam files were sorted by coordinate using samtools (v.1.9). Read count files were generated with featureCounts[57] v1.6.4 for each gene. Both raw reads and read count files are deposited at GEO (GSE220833).

For differential gene expression analysis, DeSeq2[58] (v. 1.36.0) was used, and pairwise comparisons between genotypes were performed (expression ~genotype). Adjusted p values were calculated using the Benjamin & Hochberg method. A gene was considered differentially expressed if the log2 of its expression fold change was greater than 0.5, and if the adjusted p value was less than 0.1. Lists of differentially expressed genes are added as Supplementary Data 1 and 7. Pathway enrichment analysis was performed using Metascape[59], results are listed in Supplementary Data 2.

Analysis of global alternative splicing changes compared to RBM20-WT was performed using rMATS turbo[33] (v4.1.1), according to the published manual. We performed pairwise comparisons to corresponding RBM20-WT iPSC-CMs for each experiment, and analyzed AS events based on the junction counts (JC). Events were classified into exon skipping (SE), intron retention (RI), mutually exclusive exons (MXE), alternative 3' splice site (A3SS), and alternative 5' splice site (A5SS). Supplementary Data 4 and 9 provide rMATS outputs for all pairwise comparisons to WT performed, for all types of AS events based on JC. An alternative splicing event was considered significant if the absolute value of inclusion level difference was greater than 0.1, and if the false discovery rate was less than 0.01. These are summarized in Supplementary Figs. 2g–i and Supplementary Fig. 4g–i.

The list of RBM20 target genes was taken from[13]. Briefly, this list consists of 45 genes that were conserved across RBM20 perturbations (KO, S635A) and across model species (rat, human, mouse), summarized in Table 2 of ref. [13] based on several studies[15,18,19]. We refer to this list as "core RBM20 targets".

To compare the percentage of spliced-in values (PSI values) for all exons across samples, without restricting to only pairwise comparisons to RBM20-WT, we used pipeline described in ref. [60]. First, for each exonic part (annotation based on DEXSeq[61]), inclusive and exclusive reads were identified. A read is considered inclusive, if it includes the exon part of interest. A read is considered exclusive, if it spans both the upstream and downstream exons but does not align to the exon of interest. For each exonic part, inclusive and exclusive reads were counted directly from the BAM files based on STAR output as described in ref. [60]. The PSI values are then calculated as the ratio of inclusive reads to the sum of inclusive and exclusive reads per each exonic part. For selecting the most differentially spliced exonic part for the heatmaps (Fig. 1j, Fig. 2e), a Student's *T*-test was used to determine exonic parts of the core RBM20 target genes with *P* value < 0.05, and with absolute value of differences between means of PSI values greater than 0.1, comparing only RBM20-WT to RBM20-R634Q cells. PSI values used for these plots, as well as their exonic part IDs are added to the Source data. Full lists of all PSI values for all exonic parts annotated to corresponding exons (using BEDtools[62] intersect) are added as Supplementary Data 3 and 8.

The package ggplot2 (v.3.4.0) from R Studio (v. 4.2.1) was used to plot all figures based on the Source data file. Heatmaps were plotted using pheatmap package (v. 1.0.12).

## Cell-based Luciferase *TTN* splicing reporter assay

HEK293 cells were seeded on 96-well plates and transfected at 50% confluence with PEI40 at a 1:3 ratio (DNA: PEI40) and a total of 200 ng

of plasmid DNA (1 ng splice reporter TTN-IG Ex241-243 and a 20 × molar excess of the RBM20 expression plasmids (compare[63] for WT, mutations were introduced by site-directed mutagenesis in a two-step cycle PCR approach) or control plasmid pcDNA3.1 (Invitrogen, Cat# V79520).

Plasmids and PEI40 in FBS-free medium were incubated for 15 min before the transfection mixture was added to the cells. Each transfection experiment was repeated ten times and cell viability was measured 60 h post transfection using PrestoBlue (Thermo Fisher Scientific, cat# A13261). Luciferase activity was measured 60 h post transfection using the Dual-Luciferase® Reporter Assay System (Promega) on an Infinite® M200 Pro (TECAN) plate reader. Ratios of firefly to renilla luciferase activity were normalized to the WT RBM20 expressing cells. All data are expressed as the mean of biological replicates ($n = 10$) ± SEM. Group comparisons were analyzed by one-way ANOVA and Bonferroni post test. *P* values were considered statistically significant as follows: $^*P < 0.05$; $^{**}P < 0.01$; $^{***}P < 0.001$.

### CRISPR/Cas9 gRNA library design and cloning
The genome-wide CRISPR/Cas9 gRNA libraries were designed as described in ref. 5. Briefly, the library targets 18,408 protein-coding genes listed in the Consensus Coding Sequence Database[64]. It consists of six independent sub-libraries, each containing one gRNA per gene[5]. Each of these sub-libraries contains the same 118 targeting and 487 nontargeting controls. The library was cloned into the CROPSeq-guide(F + E)-Puro backbone (see Supplementary Data 14), as described in ref. 5. gRNA representation of the genome-wide library at the plasmid stage was checked previously[5], and used as a plasmid stock for the lentivirus generation for this work. These data were also used as a reference for gRNA representation at the plasmid stage for the downstream data processing (see below).

### Cloning of individual gRNAs
For individual KO experiments, gRNAs were synthesized as two short oligos with flanking sequences resembling the Esp3I sticky ends of CROPSeq-guide(F + E)-Puro vector (fwd 5′- CACCG[N20], rev 5′-AAAC[N20-reverse complement]), see sequences in Supplementary Table 2. The oligos (10 mM each) were phosphorylated and annealed using 1 U/µl T4 PNK (NEB), and 1X T4-ligase buffer (NEB) in a thermocycler with the following program: 37 °C 30 min, 65 °C 20 min, 95 °C 5 min, ramp down to 25 °C at 5 °C/min. The phosphorylated and annealed oligos (1 µl) were then ligated with 25 ng of Esp3I-digested (NEB) and gel-extracted (Qiagen) CROPSeq-guide(F + E)-Puro backbone using 1 µl of T4 ligase (NEB), and 1× T4-ligase buffer (NEB) in total volume of 10 µl, for 10 min at RT, inactivated for 10 min at 65 °C followed by transformation into the NEB Stable competent *E. coli* (NEB), according to the manufacturer's instructions. Lentivirus for cell transductions was produced as described above.

### Pooled and individual CRISPR perturbations
For the pooled screening experiment, HeLa Tet::Cas9 pEFa-eGFP-RBM20-WT were plated with a density of 750 000 cells per 15 cm tissue culture dish, and cultured for three days until a confluency of 40% was reached (6,000,000 cells per 15 cm dish). To achieve > 500× gRNA coverage, $60 \times 10^6$ of cells were transduced per each genome-wide library (ten 15-cm dishes per library). Cells were infected with lentivirus delivering the genome-wide library, with 25 µl of 100× concentrated virus per plate, at a low infectivity rate to allow only single qRNA integrations per cell. Twenty-four hours later, cells were trypsinized, resuspended in the culture medium containing 2 µg/µl of Puromycin (Thermo Fisher Scientific), and plated back to the same dishes. The next day, the medium was changed to fresh Puromycin-containing culture medium, to wash away dead cells. Around 20% of cells got Puromycin resistance and were further kept in culture. Three days later, and for the next 7 days, cells were split every 3 days and cultured

in the presence of 2 µg/µl Puromycin (Thermo Fisher Scientific) and 2 µg/ml Doxycycline (Sigma) to activate Cas9. After 7 days of being cultured in the presence of Doxycycline, the culturing medium containing only Puromycin was used until cells were harvested for ICS. At each splitting, cells infected with the same library from all plates were pooled together after trypsinization, and 1,500,000 of cells were plated per each new of seven 15-cm dishes, keeping the coverage at >500× for each individual genome-wide library. Three days prior to harvesting for ICS, 1,500,000 of cells were plated to fifteen 15-cm dishes for each library, and all of them were used for sorting.

For individual CRISPR perturbations, cells were cultured in six-well plates until 40% confluency (150,000 cells per well), and 10 µl of 50× concentrated virus were added per each well. On the next day, cells were trypsinized, resuspended in the medium containing 2 µg/µl Puromycin (Thermo Fisher Scientific), and plated back to the same wells. The next day, the medium was changed to a fresh Puromycin-containing medium. Once cells reached 90–100% confluency, they were split and cultured in medium containing 2 µg/µl Puromycin (Thermo Fisher Scientific) and 2 µg/ml Doxycycline (Sigma) for the first seven days, followed by seven days of only Puromycin-containing medium. Cells were split twice per week. At least 16 days post transduction, cells were harvested for downstream analyses as described above.

### ICS-based CRISPR screens
Cells were prepared as described above using lentiviral transduction. Samples were kept at 4 °C at all times between harvest and genomic DNA isolation after sorting. Sorting was performed as described before[5] with the following modifications. Cells were sorted in batches of 100,000 cells in the collection fractions and the input samples were refreshed regularly by the addition of concentrated cell suspension to a total volume of 1 ml. For the selection of the populations from the eGFP-DRAQ5 correlation parameter, ranged gates were drawn comprising the 7% of cells with the lowest or highest correlation index. From each batch of cells used for sorting, an input sample containing the same number of cells as present in the sorted upper and lower sample was collected. Sorted samples and input samples were collected by centrifugation for 5 min at $500 \times g$ at 4 °C, and pellets were either frozen at −20 °C or stored on ice until gDNA preparation. One million cells were collected per library and pooled into a single tube for gDNA preparation.

### Genomic DNA isolation, library preparation, and sequencing
Genomic DNA was isolated from the sorted cells using NEB Monarch genomic DNA purification kit (New England Biolabs), including the RNase treatment and elution in 50 µl elution buffer. DNA concentration was measured using Qubit High sensitivity dsDNA kit (Thermo Fisher Scientific), according to the manufacturer's instructions.

PCR1 was done with 125–525 ng of gDNA (per reaction, 6 reactions per library), 1.5 µl of 10 µM pU6 fwd, 1.5 µl of 10 µM pLTR-CROP-rev (see sequences in Supplementary Table 5), and 25 µl KAPA HiFi Hotstart Readymix (Roche) in 50 µl total volume. For each gRNA sublibrary, six 50 µl reactions were set up using the total amount of gDNA recovered from sorted cells and the input samples. Cycling conditions for PCR1 were one cycle at 95 °C for 3 min; 24 cycles at [98 °C for 20 s, 67 °C for 15 s, 72 °C for 15 s]; one cycle at 72 °C for 1 min; and cooling to 4 °C. PCR reactions of the same template (same sublibrary) were pooled (six PCR products into one of total volume 300 µl) and the product was purified with 0.8× volume of AMPure XP (Beckman) with two 80% ethanol washes, and elution in 40 µl water. Concentrations were then measured with Qubit High sensitivity dsDNA kit (Thermo Fisher Scientific), according to the manufacturer's instructions.

PCR2 was done with 10 ng PCR1 product (per reaction, 6 reactions per library), 5 µl of 3 µM CROPseq_libQC_i5_s:n staggered primer[65] (different primer for each reaction for one sample), 5 µl of 3 µM

CROPseq_i7:n barcoded primer (same for all reactions for one sample, but unique to every sample), and 25 μl KAPA HiFi Hotstart Readymix (Roche) in 50 μl total volume. Same primers were used as in[5], and are shown in Supplementary Table 5. Cycling conditions for PCR2 were one cycle at 95 °C for 3 min; 8 cycles at [98 °C for 20 s, 67 °C for 15 s, 72 °C for 15 s]; one cycle at 72 °C for 1 min; and cooling to 4 °C. Product was purified as above and eluted in 40 μl H2O. Concentrations were then measured with Qubit High sensitivity dsDNA kit (Thermo Fisher Scientific), and ready libraries were checked using DNA 1000 Bioanalyzer (Agilent) to yield a single product around 300 bp.

For Illumina sequencing, libraries were pooled in equimolar ratio (nine libraries per pool) and sequenced using an Illumina NextSeq 500 (75 bp, single end mode) in high output mode with 8 reads to read out the i7 barcode, and 67 reads on Read1 to read through the stagger sequence and identify the gRNA. PhiX spike-in was used to diversify the libraries.

### CRISPR-screen data analysis

The abundance of gRNA was quantified from the sequencing reads using MAGeCK (v0.5.9) tool with default parameters[66]. To account for differences in sequencing depth, raw gRNA counts were normalized to the median count of the targeting control gRNAs in the corresponding sample. Scaling of the normalized counts was done by multiplication with the median count of targeting controls across all samples. The evaluation of screen quality was done as described in[5] based on the dropout of essential genes in input cell populations compared to the plasmid library (sequenced previously in ref. 5) with reference core- and nonessential gene lists described previously in ref. 67. All precision–recall curves were generated using the R package "ROCR" (v. 1.0-11)[68]. The MAUDE (v. 0.99.4)[69] package for R was used for hit calling, using targeting controls as a reference for MAUDE analysis. False discovery rates for each plot are indicated in the main text.

### Whole-cell extract preparation, cell fractionation and western blotting

For whole-cell extracts, cell pellets were lysed in NP-40 lysis buffer (50 mM Tris-HCl pH 7.5, 150 mM NaCl, 2 mM EDTA, 1% (v/v) NP-40. All protein extraction buffers contain PhosSTOP (Sigma-Aldrich, 04906837001) and Protease Inhibitor Cocktail (Sigma-Aldrich, 05056489001). Cell fractionation was performed as follows. First, cells were resuspended in two pellet volumes of hypotonic buffer (10 mM HEPES pH 7.5, 10 mM KCl, 1.5 mM MgCl2) incubated on ice for 15 min and homogenized with 20 strokes using a loose pestle. Nuclei and insoluble cellular compartments were pelleted at 3900 rpm for 15 min, and the supernatant was collected as soluble cytoplasmic fraction, which was corrected to 10% (v/v) glycerol, 3 mM EDTA, 0.05% (v/v) NP-40 and 150 mM NaCl final concentration. The remaining pellet was resuspended in chromatin digestion buffer (20 mM HEPES pH 7.9, 1.5 mM MgCl2, 10% (v/v) glycerol, 150 mM NaCl, 0.1% (v/v) NP-40 and 125 U benzonase (MerckMillipore, 70746-4) and incubated for 1 h at 4 °C. To ensure extraction of all nuclear soluble and insoluble proteins, as well as of insoluble cytoplasmic components, NaCl concentration was then increased to 500 mM and samples were incubated on ice for 30 min. Prior to centrifugation at 20,000 × $g$ for 20 min at 4 °C, the salt concentration was diluted back to 150 mM NaCl by addition of high salt dilution buffer (20 mM HEPES pH 7.9, 3 mM EDTA, 1.5 mM MgCl2, 10% (v/v) glycerol, 500 mM NaCl and 0.1% (v/v) NP-40) and the supernatant was kept as nuclear and insoluble fraction. 30–100 μg protein/lane was separated on 4–12% or 3–8% NuPage gels (Invitrogen) and transferred to Trans-Blot Turbo Mini 0.2 μm Nitrocellulose (biorad 1704158) using Trans-Blot Turbo Transfer System. Membranes were blocked in 5% (w/v) skimmed milk in PBS-T (PBS, 0.1% (v/v) Tween20) for 1 h at room temperature and incubated with

primary antibody (in 5% (w/v) skimmed milk in PBS-T) overnight at 4 °C (anti-GAPDH (Abcam ab9485 1:1000), anti-MOV10 (Sigma PLA0195, 1:1000), anti-TNPO3 (Invitrogen MA5-37991, 1:1000), anti-RBM20 (Abcam ab233147), anti-H3 (Abcam ab176842, 1:1000)). All used antibodies are listed in Supplementary Table 3. Antibody against GAPDH were used to control loading when necessary. Membranes were washed several times in PBS-T, incubated with HRP-conjugated secondary antibody in 5% (w/v) skimmed milk in PBS-T (Goat anti-rabbit-HRP, Abcam ab97051, 1:10,000) and visualized using SuperSignal West Dura Chemiluminescent Substrate ECL reagent (Thermo Fisher Scientific, 34075) and visualize using Bio-Rad ChemiDoc Touch (Software v. 2.3.0.07).

The limitation of this method is that, together with the nuclear fraction, the insoluble cytoplasmic fraction gets extracted into the same reaction tube, which explains why some cytoplasmic proteins can be found in the nuclear fraction.

### Co-immunoprecipitation

For GFP immunoprecipitations, 1 mg of the whole-cell extracts, cytoplasmic fraction or nuclear and insoluble fractions were incubated with 30 μl of GFP-Trap® Magnetic Particles M-270 (ChromoTek) for at 4 °C for 3 h. Beads were washed 5 times in IP wash buffer (150 mM NaCl, 20 mM Tris-HCl pH 7.5, 1.5 mM MgCl2, 3 mM EDTA, 10% (v/v) glycerol, 0.1% (v/v) NP-40, phosphatase inhibitors (PhosSTOP, Sigma-Aldrich, 04906837001) and protease inhibitor cocktail (Sigma-Aldrich, 05056489001) and eluted in 30 μl of Laemmle buffer with 100 μM DTT.

### Mass spectrometry

**LC-MS/MS analysis.** Samples were subjected to an in-solution tryptic digest using a modified version of the Single-Pot Solid-Phase-enhanced Sample Preparation (SP3) protocol (PMID: 25358341, PMID: 29565595). Eluates were added to Sera-Mag Beads (Thermo Scientific, #4515-2105-050250, 6515-2105-050250) in 10 μl 15% formic acid and 30 μl of ethanol. Binding of proteins was achieved by shaking for 15 min at room temperature. SDS was removed by four subsequent washes with 200 μl of 70% ethanol. Proteins were digested overnight at room temperature with 0.4 μg of sequencing grade modified trypsin (Promega, #V5111) in 40 μl HEPES/NaOH, pH 8.4 in the presence of 1.25 mM TCEP and 5 mM chloroacetamide (Sigma-Aldrich, #C0267). Beads were separated, washed with 10 μl of an aqueous solution of 2% DMSO, and the combined eluates were dried down.

Peptides were reconstituted in 10 μl of H2O and reacted for 1 h at room temperature with 80 μg of TMT6plex (For inputs, Thermo Scientific, #90066) or with 40 μg of TMTpro (For eluates, Thermo Scientific, #A44522) label reagent dissolved in 4 μl of acetonitrile. Excess TMT reagent was quenched by the addition of 4 μl of an aqueous 5% hydroxylamine solution (Sigma, 438227). Peptides were reconstituted in 0.1% formic acid, mixed to achieve a 1:1 ratio across all TMT-channels and purified by a reverse phase clean-up step (OASIS HLB 96-well μElution Plate, Waters #186001828BA).

Pulldowns were analyzed by LC-MS/MS on an Orbitrap Fusion Lumos mass spectrometer (Thermo Scientific) as previously described (PMID:30858367). To this end, peptides were separated using an Ultimate 3000 nano RSLC system (Dionex) equipped with a trapping cartridge (Precolumn C18 PepMap 100, 5 mm, 300 μm i.d., 5 μm, 100 Å) and an analytical column (Acclaim PepMap 100. 75 × 50 cm C18, 3 mm, 100 Å) connected to a nanospray-Flex ion source. The peptides were loaded onto the trap column at 30 μl per min using solvent A (0.1% formic acid) and eluted using a gradient from 2 to 40% Solvent B (0.1% formic acid in acetonitrile) over 2 h at 0.3 μl per min (all solvents were of LC-MS grade). The Orbitrap Fusion Lumos was operated in positive ion mode with a spray voltage of 2.4 kV and capillary temperature of 275 °C. Full scan MS spectra with a mass range of

375–1500 *m/z* were acquired in profile mode using a resolution of 120,000 (maximum fill time of 50 ms or a maximum of 4e5 ions (AGC) and a RF lens setting of 30%. Fragmentation was triggered for 3 s cycle time for peptide-like features with charge states of 2–7 on the MS scan (data-dependent acquisition). Precursors were isolated using the quadrupole with a window of 0.7 *m/z* and fragmented with a normalized collision energy of 38. Fragment mass spectra were acquired in profile mode and a resolution of 30,000 in profile mode. Maximum fill time was set to 64 ms or an AGC target of 1e5 ions). The dynamic exclusion was set to 45 s.

For inputs: Peptides were subjected to an off-line fractionation under high pH conditions (PMID: 25358341). The resulting 12 fractions were then analyzed on a QExactive plus.

Peptides were separated on an UltiMate 3000 RSLC nano LC system (Dionex) fitted with a trapping cartridge (µ-Precolumn C18 PepMap 100, 5 µm, 300 µm i.d. × 5 mm, 100 Å) and an analytical column (nanoEase™ M/Z HSS T3 column 75 µm x 250 mm C18, 1.8 µm, 100 Å, Waters). Trapping was carried out with a constant flow of trapping solution (0.05% trifluoroacetic acid in water) at 30 µL/min onto the trapping column for 6 min. Subsequently, peptides were eluted via the analytical column running solvent A (3% DMSO, 0.1% formic acid in water) with a constant flow of 0.3 µL/min, with increasing percentage of solvent B (3% DMSO, 0.1% formic acid in acetonitrile). The outlet of the analytical column was coupled directly to an Orbitrap QExactive™ plus Mass Spectrometer (Thermo Fisher Scientific) using the Nanospray Flex™ ion source in positive ion mode.

The peptides were introduced into the QExactive plus via a Pico-Tip Emitter 360 µm OD × 20 µm ID; 10 µm tip (CoAnn Technologies) and an applied spray voltage of 2.2 kV. The capillary temperature was set at 275 °C. Full mass scan was acquired with mass range 375–1200 *m/z* in profile mode with resolution of 70,000. The filling time was set at a maximum of 100 ms with a limitation of $3 \times 10^6$ ions. Data-dependent acquisition (DDA) was performed with the resolution of the Orbitrap set to 17,500, with a fill time of 50 ms and a limitation of $2 \times 10^5$ ions. A normalized collision energy of 32 was applied. Dynamic exclusion time of 20 s was used. The peptide match algorithm was set to "preferred" and charge exclusion "unassigned", charge states 1, 5–8 were excluded. MS2 data were acquired in profile mode.

**MS data analysis**. Acquired data were analyzed using IsobarQuant (PMID: 26379230) and Mascot V2.4 (Matrix Science) using a reverse UniProt FASTA Homo sapiens database (UP000005640) including common contaminants and the expressed bait sp| P2147_GFPflagRBM20WT | P2147_GFPflagRBM20WT (see the sequence below).

The following modifications were taken into account: Carbamidomethyl (C, fixed), TMT10plex (K, fixed), Acetyl (N-term, variable), Oxidation (M, variable) and TMT10plex (N-term, variable). The mass error tolerance for full scan MS spectra was set to 10 ppm and for MS/MS spectra to 0.02 Da. A maximum of 2 missed cleavages were allowed. A minimum of two unique peptides with a peptide length of at least seven amino acids and a false discovery rate below 0.01 were required on the peptide and protein level (PMID: 25987413).

**Raw data processing**
**For RBM20 co-IP in siRNA control vs. siRNA TNPO3**. The raw output files of IsobarQuant (protein.txt−files) were processed using the R programming language (ISBN 3-900051-07-0). Only proteins that were quantified with at least two unique peptides were considered for the analysis. 159 proteins passed the quality control filters. Raw TMT reporter ion intensities ('signal_sum' columns) were first cleaned for batch effects using limma (PMID: 25605792) and further normalized

using vsn (variance stabilization normalization−PMID: 12169536). Proteins were tested for differential expression using the limma package. The replicate information was added as a factor in the design matrix given as an argument to the 'lmFit' function of limma. Also, imputed values were given a weight of 0.05 in the "lmFit" function. A protein was annotated as a hit with a false discovery rate (fdr) smaller 0.05 and a fold change of at least 100% and as a candidate with an fdr below 0.02 and a fold change of at least 50%.

**For RBM20 co-IP in WT, P633L, R634Q, and RSS**. The raw output files of IsobarQuant (protein.txt−files) were processed using the R programming language (ISBN 3-900051-07-0). Only proteins that were quantified with at least two unique peptides were considered for the analysis. Moreover, only proteins which were identified in two out of three mass spec runs were kept. 771 proteins passed the quality control filters. Data for nuclear and cytoplasmic fractions were analyzed separately. Raw TMT reporter ion intensities ("signal_sum" columns) were first cleaned for batch effects using limma (PMID: 25605792) and further normalized using vsn (variance stabilization normalization−PMID: 12169536). Missing values were imputed with "knn" method using the Msnbase package (PMID: 22113085). Proteins were tested for differential expression using the limma package. The replicate information was added as a factor in the design matrix given as an argument to the 'lmFit' function of limma. Also, imputed values were given a weight of 0.05 in the "lmFit" function. A protein was annotated as a hit with a false discovery rate (fdr) smaller 0.05 and a fold change of at least 100% and as a candidate with an fdr below 0.02 and a fold change of at least 50%, compared to the no-bait control. All proteins, classified as hits or candidates in comparison to the negative no-bait control, were considered to be interactors with RBM20 in a given sample. Common between WT and the mutant variants, as well as unique for mutant variants only, interactors were then analyzed for pathway enrichment with Metascape[59].

**For inputs**. The raw output files of IsobarQuant (protein.txt−files) were processed using the R programming language (ISBN 3-900051-07-0). Only proteins that were quantified with at least two unique peptides were considered for the analysis. 4553 proteins passed the quality control filters. Raw TMT reporter ion intensities ("signal_sum" columns) were first cleaned for batch effects using limma (PMID: 25605792) and further normalized using vsn (variance stabilization normalization−PMID: 12169536). Proteins were tested for differential expression using the limma package. The replicate information was added as a factor in the design matrix given as an argument to the 'lmFit' function of limma. Also, imputed values were given a weight of 0.05 in the "lmFit" function. A protein was annotated as a hit with a false discovery rate (fdr) smaller 0.05 and a fold change of at least 100% and as a candidate with an fdr below 0.02 and a fold change of at least 50%.

**AlphaFold and MutaSeq predictions**
We employed AlphaFold2 (AF2)[38,39] within the JupyterHub on the EMBL Hamburg HYDE cluster. The default settings of AF2 in Multimer mode (v2.2.2) were used with three recycling rounds per model, with enabled amber relaxation and a total of five predictions per model. This resulted in a total of twenty-five predictions per AlphaFold2 run. The predictions were performed with the full-length amino acid sequence of the canonical TNPO3 sequence (amino acid 1-923 of Q9Y5L0, NM_012470.4) and with full-length RBM20 (amino acid 1–1227 of Q5T481, NM_001134363.3) or only with amino acid sequence 511–673 to predict the RRM-RS domain. We used the Mutabind2 server[43] with all 20 predictions of TNPO3 in complex with the isolated wild-type RRM-RS domain of RBM20 to test the effect of two single amino acid substitutions (P633L or R634Q) within RBM20 on the binding affinity with TNPO3.

## AAV9 production

pCMV-Tnpo3 was cloned into the AAV9 packaging backbone (derived from Addgene 137177, gift from the Genetic and Viral core facility, EMBL Rome, Italy) by digesting the backbone with BglII and AgeI (NEB), and amplifying the murine *Tnpo3* cDNA (GenScript) with the oligos listed in Supplementary Table 1, followed by Gibson assembly of the two fragments (NEB), see the final plasmid sequence in Supplementary Data 14.

The serotype 9 rAAV containing pCMV-Tnpo3 cDNA was produced in HEK293T/17 cells using the triple-transfection method with linear PEI (25 kDa) in a Corning Hyperflask. After 72 h, the cells were lysed and DNA was degraded by adding Triton X-100 (final concentration of 1%) and 19 µl Bensonase (25–35 U/µl) for 1 h at 37 °C with 200 rpm shaking[70]. The cell debris/virus mix was removed and the Hyperflask was washed with 200 ml, sterile 1×PBS. The washing solution and the cell suspension were centrifuged at $4000 \times g$ for 20 min. The supernatant was filtered with a 0.45-µm PES filter and then concentrated to a total volume of 30 ml using tangential flow filtration[70]. The concentrated virus was then purified by standard methods with an iodixanol gradient. The 200 µl final volume of virus in PBS with 0.001% pluronic F-68 was aliquoted and the titer ($3.3 \times 10^{13}$) was determined by qPCR using primers within the CMV promoter.

## Mouse handling and treatments

RBM20-WT or -P635L-Hom strains in a C57BL/6 J genetic background were used for experimental procedures. The animals were maintained in individually ventilated plastic cages (Tecniplast) in an air-conditioned (temperature 22 °C ± 2 °C, humidity 50% ± 10%) and light-controlled room (illuminated from 07:00 to 19:00 h). Mice were fed 1318 P autoclavable diet (Altromin, Germany) ad libitum. All animal care and procedures performed in this study conformed to the EMBL Guidelines for the Use of Animals in Experiments and were reviewed and approved by the Institutional Animal Care and Use Committee (IACUC).

Mice were treated with $10^{12}$ VG of AAV9 delivering either *Tnpo3* cDNA, or PBS as a negative control, diluted in 100 µl of PBS, via injection in the tail vein. Animals were humanely sacrificed according to the protocol approved by the IACUC.

## Isolation of primary mouse cardiomyocytes for imaging

Microscopy slides were coated with 10 µl/ml of laminin (Gibco, 23017015) in PBS overnight. For CM isolation, we adapted the protocol described in ref. 71. Briefly, for heart perfusions, the following buffers were used: EDTA buffer (130 mM NaCl, 5 mM KCl, 0.5 mM NaH$_2$PO$_4$, 10 mM HEPES, 10 mM Glucose, 10 mM 2,3 Butanedione monoxime, 10 mM Taurine, 5 mM EDTA), Perfusion buffer (130 mM NaCl, 5 mM KCl, 0.5 mM NaH$_2$PO$_4$, 10 mM HEPES, 10 mM Glucose, 10 mM 2.3 Butanedione monoxime, 10 mM Taurine, 1 mM MgCl$_2$), Collagenase buffer (Perfusion buffer, 1.5 mg/ml Collagenase II (Gibco, 17101015), 1.5 mg/ml Collagenase IV (Gibco, 17104019), 0.15 mg/ml Protease Type XIV (Sigma, P5147) and stop solution (Perfusion buffer, 5% FBS Supreme (Pan Biotech, P30-3031)). The Collagenase buffer was pre-warmed to 37 °C prior usage. Mice were anaesthetized in a CO$_2$ chamber, opened up, and the descending aorta and vena cava were both cut. Then, 7 ml of the EDTA buffer were injected steadily for about one minute into the basis of the right ventricle, after which the ascending aorta was clamped. The heart was removed and transferred to a 60-mm dish containing 10 ml of the EDTA buffer. A syringe was used to push 10 ml of EDTA buffer through the left ventricle steadily for ~2 min. The heart was then transferred to a 60-mm dish containing 10 ml of the Perfusion buffer, and the left ventricle was steadily injected with 10 ml of the Perfusion buffer to flush the remaining EDTA. Next, the heart was transferred to a 60-mm dish containing 10 ml of the Collagenase buffer, and the left ventricle was steadily perfused five to six times with 10 ml of Collagenase buffer. Afterwards, the heart was cut into the desired regions. After a small piece of the left ventricle was saved and snap frozen to be further used for RNA extraction (see above), the rest of the left ventricle was transferred to the dish containing 3 ml Collagenase buffer for CM isolation. The tissue was teared apart into 1 mm × 1 mm pieces and pipetted up and down for about 5 min to dissociate the cells, after which the collagenase reaction was stopped by adding 5 ml of the Stop solution. The cells were filtered through a 100-µm filter and pelleted by gravity for about 20 min, after which the pellet was gently resuspended in Perfusion buffer and the filtering procedure was repeated one more time. The pellet was then resuspended in pre-warmed DMEM medium containing 10% FBS Supreme (Pan Biotech, P30-303), plated onto the pre-coated microscopy slides, and let in the cell culture incubator for three hours. Slides were then washed twice with PBS, followed by fixation and staining protocols described above.

## Statistical analysis

All experiments were performed in at least three biological replicates, unless otherwise specified in the figure legends. Statistical significance was quantified either with Welch two-sample *t*-test for pairwise comparisons, or with ANOVA with Tukey's HSD or Bonferroni post tests (all two-tailored) for multiple comparisons, unless otherwise specified in the figure legends.

## Reporting summary

Further information on research design is available in the Nature Portfolio Reporting Summary linked to this article.

## Data availability

Raw and processed RNA-sequencing and ICS screen data is deposited at GEO under accession code GSE220833. Raw and processed proteomics data is deposited at PRIDE under accession code PXD038790. The top-ranked predictions of each structural model are available in ModelArchive under the following Archive IDs: ma-gosou, ma-8o0d5, ma-w0ijb, ma-9yrs3. To protect patient privacy in compliance with IRB requirements, individual patient data cannot be shared. Please contact the corresponding author for queries regarding aggregate de-identified patient data. Response to these queries can be expected within a month. Requests for raw de-identified data will require IRB approval and data-sharing agreement. Source data are provided with this paper.

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

## Acknowledgements

We thank S. Clauder-Muenster, B. Goldbrich, J. Froehlich for technical support; A. Grozhik and S. Clauder-Muenster for their work on engineering the RBM20 mouse lines; B. Rauscher for providing scripts and guidance for CRISPR-screen data analysis and interpretation; F. Briganti for providing iPSC lines; P. Blainey for providing Tet::Cas9 HeLa cell line; M. Mall, M. Azodi for generating and providing plasmid constructs; D. Moonen, M. Krause for experimental support; T. Seeger for advising on iPSC-CM differentiation; G. Chojnowski for advice and help with structural modeling; EMBL Proteomics Core Facility (PCF) for support, P. Haberkant and F. Stein for performing mass-spectrometry experiments and processing the data; EMBL Genetic and Viral Engineering Facility for services, A. Castillo and J. Sawitzke for engineering and titrating the AAV9 virus; EMBL Genomics Core Facility for next-generation sequencing services, V. Benes, L. Villacorta, F. Jung for advice; EMBL Flow Cytometry Core Facility for support, advice, instrument maintenance, and sorting services; EMBL Advanced Light Microscopy Facility for support, A. Khan for computational help with quantifying in vivo RBM20 granules; EMBL Laboratory Animal Resources (LAR) for maintenance and support, F. Diego Montoya Castillo for performing mouse injections, organ harvesting, and support. BMBF-DZHK 81Z0500401 (to L.M.S.); DFG-SFB 1550/1 (to L.M.S. and B.M.). DFG-SFB 1470/1 (to M.Go.). Leducq Fondation, CASTT (to V.P., B.M., M.Go., and L.M.S.); the German Centre for Cardiovascular Research (DZHK) (to B.M.); the EMBL Interdisciplinary Postdoc (EIPOD) Programme under Marie Curie Cofund Actions MSCA-COFUND-FP research fellowship (grant agreement number 847543, to K.F.); the Human Frontier Science Program Long-Term Fellowship (HFSP-LT0023/2022-L, to D.L.); NIH T32-HL007853 (to E.S.); The Zurich ACM Program is supported by the Georg und Bertha Schwyzer-Winiker Foundation, Baugarten Foundation, USZ Foundation (Dr. Wild Grant), and Swiss Heart Foundation grant no. FF17019 and FF21073 (to A.M.S.); NHLBI K08 (HL143185) and support from the Jon Taylor Babbitt Foundation (to V.N.P.).

## Author contributions

J.K., M.R.M., and L.M.S. designed the study. J.K., M.R.M., F.H., B.T., and L.S. performed the experiments. J.K. analyzed the data. K.F. performed structural analyses and predictions. M.R.M., K.F., D.S., M.Gr., D.L., M.Go., and L.M.S. provided expert guidance and feedback on analyses and results. D.S. provided CRISPR/Cas9 libraries. D.L. and M.K. provided iPSCs with stable Cas9 expression. V.P. collected and interpreted clinical data from E.S., C.M., E.B., A.O., A.M.S., and B.M. J.K., M.R.M., K.F., D.S., and L.M.S. wrote the manuscript, with feedback from all authors.

## Funding

## Competing interests

L.M.S. is a co-founder and shareholder of Sophia Genetics. The authors J.K., M.R.M., K.F., M.Go., and L.M.S. filed an invention disclosure describing TNPO3 and restoring nuclear localization of RBM20 variants discussed in this paper (U.S. Provisional Patent Application No. 63/452,252, status: filed: March 15, 2023). M.Go. is an advisor for River Biomedics. A.M.S. received educational grants through his institution from Abbott, Bayer Healthcare, Biosense Webster, Biotronik, Boston Scientific, BMS/Pfizer, and Medtronic; and speaker/advisory board/consulting fees from Bayer Healthcare, Biotronik, Daiichi-Sankyo, Medtronic, Novartis, Pfizer, and Stride Bio Inc. V.N.P. is a consultant and/or advisor for Lexeo Therapeutics, BioMarin, Inc, Viz.ai and Nuevocor. The remaining authors declare no competing interests.
