## [Peer Review File · Nature Communications]

Mislocalization of pathogenic RBM20 variants in dilated cardiomyopathy is caused by loss-of-interaction with Transportin-3REVIEWER COMMENTS

Reviewer #1 (Remarks to the Author):

In this important study by Kornienko et al., the authors apply their recently developed protein cellular localization and cell sorting strategy called ICS to identify the mechanism by which wild-type RBM20 is detained in nucleus. The author's data is supported by comparative mutation analyses in iPSC-CMs, co-IP mass spec, structural predictions, high confidence microscopy, luciferase reporter assays and splicing analyses. As such, the authors should be commended for their use of novel integrative omics and high throughput screening strategies. This is a highly innovative approach that is potentially transformative for the evaluation of RNA binding protein (RBPs) localization, for both healthy and disease associated mutations. Indeed, the data produced by the authors provides a proof-of-principle for the design of new RBM20 or TNPO3 directed therapeutic strategies that are applicable to a number of diseases. In the context of RBM20 biology, the authors also attempt to answer the outstanding question: is disease driven by the nuclear or cytoplasmic presence of wild-type or mutant RBM20. To disentangle these separate questions, the authors sort for or genetically perturb iPSC-CMs to obtain those with greater RBM20 cytoplasmic or nuclear localization, with distinct RBM20 RS-domain hotspot mutations. Indeed, the authors are able to demonstrate that wild-type RBM20, when redistributed to the cytoplasm, results in the appearance of cytoplasmic aggregate-like bodies and that exclusion of well-described RBM20 regulated exons is rescued when mutant RBM20 is either enriched in the nucleus (ICS), retained using an exogenous NLS or when partially relocated by TNPO3 over-expression. While the data provided are strong, some of the author claims go too far given incomplete or overly restricted data analyses. Further, some of the results are presented in a manner in which it is difficult to quantify the precise impact of specific mutations or perturbations, due a lack of inclusion of appropriate controls, associated statistics, source data and code. However, overall these data have the potential to serve as an excellent resource for the RBM20 cardiomyopathy research community.

Major Concerns

Analyses of mis-splicing with different mutations or altered localization in this manuscript are focused on what appear to be well-described RBM20 target splicing events (referred to as "known"). This is assumed because no supplemental Excel tables are not provided and methods around such filtering is sparse. If this is the case, conclusions regarding splicing impacts with or without mutations will be limited to RBM20 regulated exons in genes such as TTN, RYR2, IMMT and CAMK2D that largely derive from RBM20 KO studies, rather than those unique to KO or specific hotspot mutations. For example, Fenix et Al. (PMC8566601) recently categorized splicing events into those that are dependent on RBM20 protein expression versus mutation. While such events are less-well characterized, these are important to consider in determining the precise impact of mutations versus altered wild-type protein localization. Indeed, the authors describe hundreds of splicing events in the P633L-nuclear localized cells in their Supplementary Fig 2g, in which a substantial fraction are potentially unique to this variant. Further, the number and relative overlap of splicing events observed with NLS-tagged R634Q with R6364Q versus wild-type should be reported, as the statement "splicing of RBM20 target genes was restored in NLS-

R634Q expressing cells to similar levels seen in WT” is not quantified. If the authors wish to make the claim that mutant RBM20 does not associate with unique RNAs in the nucleus or cytoplasm versus wild-type, they should show this through a comparative supervised analysis, considering variants/regions they have RNA-Seq data on in this study and their previous RBM20 variant iPSC-CM study.

Generally, the manuscript is challenging to follow and critically evaluate, due to a lack of detail in the results, methods and supplemental materials? In particular, a lack of supplemental Excel data tables for results (DEGs, alternative splicing events with statistics, CRISPR dropouts, co-IP MS, gene set enrichments, etc.) and quality metrics (sequencing depth, number replicates, etc.), more detailed algorithm descriptions, supplemental code and source data for figures hinder interpretation by the reviewers and others. The authors should expand their current text and add more specific details in results and methods to clarify confusion. This includes missing callouts for supplemental figures (e.g., Supplemental Fig. 1f) and caveats of the described approaches that are related to sensitivity of the assays used.

Ejection fraction and LVIDd for RSRSP should be shown relative to controls and quantified/visualized separately for P633L and controls (Fig. 1e-g).

Excessive claims are reported throughout the manuscript. These include: 1) the relative pathogenic role of mislocalized RBPs versus splicing in disease remains poorly defined (e.g., “...cytoplasmic mislocalization of TDP-43 causes Amyotrophic Lateral Sclerosis”), 2) minor differences discounted as an absence of signal (e.g., “Overall, nuclear-localized P633LRBM20 does not alter gene expression”), 3) partial versus complete rescue of Rbm20 localization by Tmpo3 over-expression on localization (e.g., “...mislocalization of the mutant variants can be rescued by up-regulating the TNPO3-RBM20 interaction.”).

It would seem as if a genome-wide CRISPR screen with ICS for mutant RBM20 versus wild-type would yield improved targets for therapy. Was this avenue tested and if not, what was the rationale?

Minor

For supplementary splicing event files provided with the manuscript and in GEO, please include the genomic coordinates of the associated junctions to enable identification by the reader. Currently the IDs are of the type: ENSG0000000003:001, which appears to point to an isoform or specific exon.

An additional panel in Supplementary Figure 1 indicating the correlation of IMMT splicing and RBM20 cytoplasmic versus the nuclear localization is likely to better illustrate the correspondence between RBM20 mutant splicing impacts and localization.

Can the authors clarify whether they considered known, novel and alternative promoter regulation in their PSI analyses? Supplemental Fig. 2g shows some categories but this is not clear from the methods or text. The description of splicing analyses in the methods section is also vague. Clarity will help readers

not familiar with the concept of junction-based local splicing variation approaches.

It is surprising the authors did not compare their MS co-IP data to that of Maatz et al. 2014, which compared S635A to WT interactions in HEK293 cells. These analyses also show a WT enrichment of RBMX. The authors should consider this data as a reference for their interactions (do they find the same or different associations).

The RSS combined mutant is first described in Supplemental Figure 3a but not defined until much later. The authors should define in the figure legend of the first mention.

The authors should replace “few” with "relatively few" in the sentence “Strikingly, we found only few differentially expressed genes between WT and NLS-tagged-R634Q expressing cells” as 110 genes are differentially regulated.

Reviewer #2 (Remarks to the Author):

The manuscript by Kornienko and colleagues describe an important novel regulatory step in the pathogenesis of RBM20 DCM: an impaired interaction of RBM20 mutated in the RS domain hotspot with the nuclear importer TNPO3. This discovery is achieved through the combination of studies in both immortalized cells and in more sophisticated cellular models (hiPSC-CMs with either patient-derived mutations or transduced with mutant RBM20) and the application of state of the art omics techniques. The application of ICS is particularly clever and is one of the first examples of how this approach can empower medically-relevant discovery. The experiments are generally comprehensive, well executed, and properly analyzed. The study key findings are novel and of strong relevance to the CVD field and beyond. I have however some important reservations about how some experiments are being (over)interpreted leading to strong conclusions on the effect of RS domain mutations that in my view are not fully supported by the data. Addressing these and other secondary comments would strengthen the study and broaden its impact.

Major comments:

Through the experiments described in Figure 2 (and associated Supplementary Figures) the authors conclude that "all tested RS-domain RBM20 mutations do not affect the intrinsic splice regulatory function of the protein". However, this conclusion is based on non-physiological systems: either the non CM HEK293T cells, or CMs infected with lentiviruses encoding RBM20 through a strong TetO promoter and thus leading to an excess of RBM20 expression compared to the endogenous levels. This is apparent from the IF in Fig. 2C, whereby many RBM20 nuclear foci are formed in WT-infected CMs: this is in contrast with the existence of usually only 2 foci in CMs close by to the TTN loci, as observed in several studies (more foci are observed only in polyploid mature CMs, which are rarely seen in hiPSC-CMs cultures). Of note, such RBM20 foci have been shown to be sites of more efficient AS regulation (PMID: 30948719). Thus, the experiments presented here do not rule out that under normal (and limiting)

amounts of RBM20 the RS mutation may affect the formation of RBM20 foci (i.e. through impaired interaction with the TTN pre-mRNA or with other splicing-regulating RNAs/proteins). The conclusion of Figure 2 should therefore be either toned down (with the aforementioned possibility being discussed) or supported by data whereby NLS-carrying RBM20 mutant proteins are expressed in physiological levels. As a matter of fact the author's own data presented in Figure 5 c-d indicate that rescuing mutant RBM20 localization is not sufficient to completely restore the AS of RBM20 targets, which is consistent with a role of RS mutations besides that of impaired nuclear import.

"This concludes that loss of interaction with TNPO3 results in formation of RBM20 RNP granules, regardless of the mutation present" and "We show that WT RBM20, when forced to remain in the cytoplasm, also forms granules of the same nature as the mutant variant". Similar to my previous comment, this conclusion should be toned down or supported by further data. While the authors convincingly show that WT RBM20 in TNPO3 KD cells form some kind of cytoplasmic granules, these are not well characterized in CMs while experiments in HeLa are interesting but not conclusive of the endogenous behaviour. Considering that studies such as refs 24 and 27 have suggested that bona fide RBM20 mutant RNP granules in adult CMs may be pathogenic in their own right this is not just a matter of semantics or curiosity. I encourage the authors to acknowledge that the exact nature of RBM20 RNP granules may still be dependent on RS mutations unless they are willing and able to establish the contrary through more rigorous characterization of TNPO3 loss-of-function CMs.

Secondary comments:

What's the authors' explanation as to why only a subset of P633L RBM20 mutant hiPSC-CMs show marked cytoplasmic mislocalization? Even more importantly, can they be sure that the P633L-nuclear cells truly express the mutant RBM20 at comparable levels as P633-cyt? Barring issues with the hiPSC clonality, which should be excluded if not done so already, a potential explanation for the heterogeneity may be due to different transcriptional activity of the two RBM20 alleles: could the authors test this by probing the mutant/wt allele ratio at the mRNA level in the two cell populations? If this was not similar it may explain the marked differences in gene expression alterations, not allowing to conclude that these are only due to differential (stochastic?) mislocalization.

Data from Figure 4a and b in WT CMs are not incredibly convincing, possibly due to difficulties in homogeneously KD TNPO3 with siRNA. For instance I wonder whether cells in which the correlation of DRAQ5 and RBM20 is unchanged after TNPO3 siRNA (peak overlapping with siRNA control in the bimodal distribution of TNPO3 siRNA cells) have not been transfected/KD or else are cells whereby other mechanisms are at play bypassing the requirement for TNPO3. Given the importance of validating the role of TNPO3 in the endogenous context of CMs I recommend that the author consider performing a more robust and ideally isogenic experiment using a KO or, should this not be compatible with iPSC-CM differentiation, and inducible KD/KO

Minor comments:

Dots indicating individual data points are too small in several figure panels (i.e. 1b, 1d, 1k, ...). If a reasonably sized dot plot is too crowded perhaps a violin plot would be most suited.

Figure 1b: was the correlation between DAPI and RBM20 calculated on 3D reconstructions based on Z-stacks? Either way should it be feasible for the authors to provide super-resolution imaging for RBM20, as done by the other papers cited in refs 24 and 27, it would add insight to the mislocalization phenotype and how it compares with other RS mutations.

Figure 1e: the authors indicate that LVEF in P633L patients was less severe than other P/LP variants - is this supported by statistical analyses?

The proteomic analyses were performed in non-CM cells: this is an understandable experimental choice but also a limitation that ought to be discussed. For instance, HeLa cells lack TTN and hence the nuclear foci observed therein are unlikely to have the same composition of splicing factors and other proteins (besides the fact that CM-specific proteins are not expressed in HeLa)

Figure 3d: what was the rationale for knocking out TTN in HeLa cells? It should not be expressed. Which of the other genes selected ad hoc (AKT2, SPRK1, CLK1, LMNA) are expressed? In general it would be useful if the authors could present data comparing expression in HeLa vs CMs for the genes hits from the screen (including the discarded negative regulators), and comment on any differences

Supplementary Fig. 7a highlights the intrinsically disordered nature of most of RBM20, which results in a highly unreliable structural prediction. outside of the RRM domain: this limitation should be made clearer in the text

AAV9-TNPO3 is presented as "a promising therapeutic avenue", but what are the potential side effects? Are proteins other than RBM20 more efficiently nuclear-imported? What are the known TNPO3 targets expressed in CMs?

Data availability: the authors indicate that "raw data will be available upon request" but I believe that the editorial policies of the journal require deposition to open access servers. Along the same lines I could not examine the raw/processed data, though I see no obvious issues in their analyses based on the methods described

Reviewer #3 (Remarks to the Author):

In this manuscript, Kornienko et al. study the mechanisms underlying the mislocalization and function of pathogenic RBM20 variants. The authors found that nuclear targeting of RBM20 mutant proteins restores their splicing function. Using a recently developed image-enabled cell sorting (ICS) technology of cells expressing RBM20 variants, which are found either exclusively in the cytoplasm (R634Q) or in

both the nucleus and cytoplasm (P633L), the authors uncovered TNPO3 as a main nuclear import receptor for RBM20. Overexpression of TNPO3 in vitro and in vivo partially restored the nuclear localization and splicing defects caused by RBM20 P635L mutation.

Overall the data are clear, and important, and raise interesting questions for future research. However, there are a few points that would improve the manuscript as follows:

Major points

1) In Figure 1i, a number of differentially expressed genes are presented. While the effect of P633L-cyt and P633L-nuc as well as NLS-mediated targeting of the R634Q variant to the nucleus (Figure 2d) on differential gene expression is very convincing and striking, the heatmap in Figure 1j representing the differential splicing events is not that clear. Half of TTN differentially spliced exons do not appear to be different in P633L-cyt and P633L-nuc, also the effect on other splice targets can not be clearly seen. In Supplementary Figure 2g,h a number of mis-splicing events can be detected in P633L-nuc, which somewhat argues against the conclusion that nuclear targeting of RBM20 mutant proteins restores their splicing function. These results are in contrast to the very clear rescue effect on differentially expressed genes and bring forward the question of whether nuclear targeting of RBM20 mainly rescues other steps of post-/transcriptional control.

2) The mass spectrometry analysis in Supplementary Figure 3d is very important. Together with the loss of interaction of RBM20 with TNPO3, there is a large number of interactions gained in the cytoplasm. For example interactions with components of the splicing machinery, e.g. SRSF9, SRSF10, HNRNPH3, HNRNPA3, HNRNPA0, HNRNPC, etc. Could mislocalization of RBM20 in the cytoplasm sequester other components of the splicing machinery and thereby affect additional steps of mRNA processing together with the classical function of RBM20? This could explain, in part, the different splicing patterns observed in pathogenic RBM20 variants and RBM20 KOs reported before.

3) The current work appears to contradict a previous study (PMID: 34732726) showing that the RBM20 R636S mutant has a preference for 3' UTR sequences. How does this study relate to the earlier work? Is this preference affected by its nuclear targeting?

4) The rationale for using iPSC-CMs carrying RBM20 S635FS mutation is not clear; is any truncated RBM20 protein expressed in these cells? The RBM20 localization in RBM20 S635FS cardiomyocytes overexpressing WT and eGFP-NLS-R634Q appears to be quite different to the endogenous localization pattern and the same experiment performed in HeLa cells.

5) The effect of TNPO3 OE on Ttn splicing in vivo (Figure 5i) is relatively minor. Is this due to the low infection efficiency or the low-binding affinity of P635L to the overexpressed Tnp3? How does Ttn splicing correlate with the nuclear localization of RBM20 in cardiomyocytes isolated from Tnp3 overexpressing mice?

Point-by-point response to Reviewers:

Summary of revisions

We thank the reviewers for their helpful suggestions. We have addressed the concerns and believe they further improve the manuscript and the clarity of our main claims. To briefly summarize our main revisions before our point-by-point responses below:

- We provided a more comprehensive description of splicing analysis throughout the manuscript and added requested additional figure panels (Supplementary Fig. 2i, Supplementary Fig. 4 g-i, Supplementary Fig. 1g).
- We toned down several conclusions of our study regarding restoration of splicing activity upon nuclear relocalization of RBM20, and transparently discussed limitations.
- We added all data generated in this study to Supplementary Tables 1-15. All analyses were done using published pipelines and packages which we refer to in results and methods.
- We performed a new genome-wide ICS CRISPR screen with RBM20-R634Q to capture factors that might retain RBM20 variants in the cytoplasm. We added results of this screen as a new Supplementary Figure 6.
- We performed a new *TNPO3* knockout experiment in iPSC-CMs, followed by quantification of the endogenous TNPO3 signal and the localization of endogenous RBM20 from microscopy data. We added results of this experiment as a new Supplementary Figure 7.
- We performed RNA sequencing of our reporter HeLa cells to compare gene expression of hits identified in both screens, and provide analyzed data as Supplementary tables 12 and 13.

We address the reviewer's comments individually below (reviewer's comments in *blue italics*, our responses in black font, and changes of the main text are in *purple font*).

Reviewer 1:

In this important study by Kornienko et al., the authors apply their recently developed protein cellular localization and cell sorting strategy called ICS to identify the mechanism by which wild-type RBM20 is detained in nucleus. The author's data is supported by comparative mutation analyses in iPSC-CMs, co-IP mass spec, structural predictions, high confidence microscopy, luciferase reporter assays and splicing analyses. As such, the authors should be commended for their use of novel integrative omics and high throughput screening strategies. This is a highly innovative approach that is potentially transformative for the evaluation of RNA binding protein (RPBs) localization, for both healthy and disease associated mutations. Indeed, the data produced by the authors provides a proof-of-principle for the design of new RBM20 or TNPO3 directed therapeutic strategies that are applicable to a number of diseases. In the context of RBM20 biology, the authors also attempt to answer the outstanding question: is disease driven by the nuclear or cytoplasmic presence of wild-type or mutant RBM20. To disentangle these separate questions, the authors sort for or genetically perturb iPSC-CMs to obtain those with greater RBM20 cytoplasmic or nuclear localization, with distinct RBM20 RS-domain hotspot mutations. Indeed, the authors are able to demonstrate that wild-type RBM20, when redistributed to the cytoplasm, results in the appearance of cytoplasmic aggregate-like bodies and that exclusion of well-described RBM20 regulated exons is rescued when mutant RBM20 is either enriched in the nucleus (ICS), retained using an exogenous NLS or when partially relocated by TNPO3 over-expression. While the data provided are strong, some of the author claims go too far given incomplete or overly restricted data analyses. Further, some of the results are presented in a manner in which it is difficult to quantify the precise impact of specific mutations or perturbations, due a lack of inclusion of appropriate controls, associated statistics, source data and code. However, overall these data have the potential to serve as an excellent resource for the RBM20 cardiomyopathy research community.

We thank the reviewer for the summary and recognition of our work. We have sought to address all concerns by providing more analyses of our data and comparing it with previously published RBM20 studies. We point to potential limitations of our approaches, and make more careful conclusions. We have now added all the source data, as well as supplementary data

tables with all identified hits from all our experiments. We further revised our main text and methods sections to emphasize precisely the impact of RBM20 mutations and perturbations.

Major Concerns:

1 - Analyses of mis-splicing with different mutations or altered localization in this manuscript are focused on what appear to be well-described RBM20 target splicing events (referred to as “known”). This is assumed because no supplemental Excel tables are not provided and methods around such filtering is sparse. If this is the case, conclusions regarding splicing impacts with or without mutations will be limited to RBM20 regulated exons in genes such as TTN, RYR2, IMMT and CAMK2D that largely derive from RBM20 KO studies, rather than those unique to KO or specific hotspot mutations. For example, Fenix et Al. (PMC8566601) recently categorized splicing events into those that are dependent on RBM20 protein expression versus mutation. While such events are less-well characterized, these are important to consider in determining the precise impact of mutations versus altered wild-type protein localization. Indeed, the authors describe hundreds of splicing events in the P633L-nuclear localized cells in their Supplementary Fig 2g, in which a substantial fraction are potentially unique to this variant. Further, the number and relative overlap of splicing events observed with NLS-tagged R634Q with R6364Q versus wild-type should be reported, as the statement “splicing of RBM20 target genes was restored in NLS-R634Q expressing cells to similar levels seen in WT” is not quantified. If the authors wish to make the claim that mutant RBM20 does not associate with unique RNAs in the nucleus or cytoplasm versus wild-type, they should show this through a comparative supervised analysis, considering variants/regions they have RNA-Seq data on in this study and their previous RBM20 variant iPSC-CM study.

We thank the reviewer this important comment. Below we address all points of this comment one by one:

Analyses of mis-splicing with different mutations or altered localization in this manuscript are focused on what appear to be well-described RBM20 target splicing events (referred to as “known”). This is assumed because no supplemental Excel tables are not provided and methods around such filtering is sparse.

To clarify this, we implemented the following:

- We added a detailed description of what was used as a list of RBM20 target genes (Results: page 5 lines 151-155): “The list of RBM20 targets employed in this study consists of 45 different genes¹³ (further referred to as “core RBM20 targets”). We used these genes to estimate the splicing activity of RBM20, as their splicing was consistently affected across *RBM20* perturbations (e.g. KO, S635A) and across model species (rat, human, mouse)^{15,18,19}.” We now refer to this list as “core RBM20 targets” throughout the manuscript and our response.
- We improved the materials and method section and explained in detail all included filtering steps (Materials and Methods: pages 23-25).
- We provided full lists of all alternative splicing events identified in our experiments for all analyzed data sets (Supplementary Tables 3, 4, 8, 9);

If this is the case, conclusions regarding splicing impacts with or without mutations will be limited to RBM20 regulated exons in genes such as TTN, RYR2, IMMT and CAMK2D that largely derive from RBM20 KO studies, rather than those unique to KO or specific hotspot mutations.

We acknowledge the importance of identifying alternative splicing (AS) events that are novel and unique to a specific mutation in *RBM20* in comparison to *RBM20* knock-out and wildtype cells. This has been done by other studies, including our own (Briganti et al., 2020), but wasn't the primary focus of this study. In this study, our aim was to assess whether RBM20-dependent splice regulatory activity can, in principle, be restored by its nuclear relocalization, regardless of the RS-domain mutation present. Therefore, we restricted our main conclusions to the above-described core RBM20 targets due to two main reasons:

- 1) to only analyze direct RBM20-mediated splicing without potential secondary effects of mis-splicing, mis-localization, or variability in iPSC-CM differentiations that can be different from batch to batch, clone to clone, and lab to lab;
- 2) to unify our analyses across different RS-domain variants.

To illustrate the level of RBM20-independent splicing variability observed in iPSC-CMs, we have now analyzed data from Briganti et al.²², who tested two RBM20-WT cell lines: WT is the unedited iPSC line that underwent the same procedure as edited RBM20 mutant lines; WT-NC is the parental line. Response Fig. 1a illustrates numbers of alternative splicing events

(FDR < 0.01, and |dPSI| > 0.1) detected in each line (P633L, R634Q, S635FS, and WT-NC) versus the used WT line. Analysis was done based on junction counts (JC) using the rMATS³¹ package, same as for our data present on Supplementary Figure 2g and Supplementary Figure 4g. Many differential splicing events are identified in all comparisons, including the comparison of two RBM20-WT iPSC-CMs lines WT and WT-NC. This indicates that high numbers of AS events detected by rMATS may be RBM20-independent. These differential splicing events need to be excluded from the analysis focused on measuring the direct impact of RBM20 relocalization on AS. In fact, restricting the analysis to the list of the core RBM20 targets lowers the number of AS events identified between two RBM20-WT cell lines (Response Fig. 1b). This validates that using the list of the core RBM20 targets is a more robust estimate of RBM20-dependent alternative splicing activity, though it is indeed oblivious to detection of novel AS events specific to particular mutations.

Response Fig. 1. Analysis of alternative splicing from Briganti et al., 2020 with rMATS³¹. (a) Total number of significant (FDR < 0.01 and |dPSI| > 0.1) AS events in pairwise comparisons of indicated genotypes with WT. (b) Number of significant AS events from panel (a) located in the core RBM20 targets classified by Koelemen et al. (c) Number of significant AS events from Briganti et al. overlapping with those identified in the current study, left two right: WT vs P633L (Briganti et al.) and WT vs P633Lcyt (this study), WT vs P633L (Briganti et al.) and WT vs P633Lnuc (this study), WT vs R634Q (Briganti et al.) and WT vs R634Q (this study).

For example, Fenix et Al. (PMC8566601) recently categorized splicing events into those that are dependent on RBM20 protein expression versus mutation. While such events are less-well characterized, these are important to consider in determining the precise impact of mutations versus altered wild-type protein localization. Indeed, the authors describe hundreds of splicing events in the P633L-nuclear localized cells in their Supplementary Fig 2g, in which a substantial fraction are potentially unique to this variant.

Although identifying novel AS events was not the scope of this study, analyses of our RNA-sequencing experiments provide first insights into potentially novel mRNA transcripts associating with P633L- or R634Q-RBM20 variants upon nuclear relocalization. In addition, we now compared overlaps of significant AS events identified in comparison to respective RBM20-WT cells in the current study and in Briganti *et al.* with rMATS³¹. More than a half of overlapping AS events were identified in the core RBM20 targets for comparisons of WTvsP633L (Briganti *et al.*) and WTvsP633Lnuc (this study), WTvsR634Q (Briganti *et al.*) and WTvsR634Q (this study, Response Fig. 1c). Overlapping AS events in genes that have not been classified as the core RBM20 targets could be novel mRNAs specific for each RBM20 variant. Only 17 AS events in non-RBM20 target genes were shared between comparisons of P633L (Briganti *et al.*) with P633Lnuc and P633Lcyt (out of 43 and 101, respectively, summarized in Supplementary Table 5). Of note, Briganti *et al.* and the current study used iPSC-CM lines with the same RBM20 variants, which makes these datasets well suited for identifying potentially novel splicing targets associating specifically with these variants (unlike Fenix *et al.* who analyzed another RBM20 variant).

Based on this analysis, we now added:

- a supplementary table listing overlapping AS events identified in the current study and in Briganti *et al.*²² in Supplementary Table 5
- a discussion about overlaps between splicing events identified in this study and previously published by Briganti *et al.* (Results: page 5 lines 168-172): “We compared our results to Briganti *et al.*²², and provide lists of common AS events including those that are not located in the core RBM20 targets in Supplementary Table 5. The latter indicate potentially novel splicing targets that may associate with P633L and R634Q variants, although require further experimental validation that goes beyond the scope of this study.”
- A mention of the important question to identify *potentially unique RBM20 targets* in the discussion: page 12 lines 405-406: “Future studies with endogenously expressed RBM20 variants will identify mRNA targets and protein interactors of the different relocalized RS-domain variants.”

Further, the number and relative overlap of splicing events observed with NLS-tagged R634Q with R6364Q versus wild-type should be reported, as the statement “splicing of RBM20 target genes was restored in NLS-R634Q expressing cells to similar levels seen in WT” is not quantified.

We have added the requested data, to Supplementary Figure 4 g-h:

Response Figure 2 (new Supplementary Figure 4 g-h). (a) Total number of genome-wide differential splicing events (SE = exon skipping, RI = intron retention, MXE = mutually exclusive exons, A5SS = alternative 5' starting site, A3SS = alternative 3' starting site, FDR < 0.01, inclusion level difference > 0.1, rMATS³¹) compared to the RBM20-WT iPSC-CMs. (b) Total number of differential splicing events (FDR < 0.01, inclusion level difference > 0.1, rMATS³¹) per RBM20-target gene, compared to the RBM20-WT iPSC-CMs. (c) Numbers and relative overlaps of all significant

We discussed these data in Results: page 7 lines 228-232: “Global splicing analysis in pairwise comparisons to RBM20-WT iPSC-CMs with rMATS³¹ identified fewer AS events in general, and exon skipping events in particular, in NLS-R634Q compared to R634Q (total events 437 compared to 676, and exon skipping events 277 compared to 536, respectively, Supplementary Fig. 4 g-i, Supplementary Table 9, see methods).”

In addition, we added relative overlaps for AS events from Supplementary Fig. 2g as a new Supplementary Figure 2i:

Response Figure 3 (new Supplementary Figure 2 i): Numbers and relative overlaps of all significant (FDR < 0.01, inclusion level difference > 0.05, RMATS) AS events from panel (g) in pairwise comparisons to RBM20-WT iPSC-CMs.

If the authors wish to make the claim that mutant RBM20 does not associate with unique RNAs in the nucleus or cytoplasm versus wild-type, they should show this through a comparative supervised analysis, considering variants/regions they have RNA-Seq data on in this study and their previous RBM20 variant iPSC-CM study.

We did not intend to make any claims regarding potential gain-of-function effects of nuclear re-localization of RBM20 variants on alternative splicing. We fully agree with the reviewer that assessing and validating them in a physiological system with endogenous levels of re-localized RBM20 is of high importance and should be addressed in future studies.

2 - Generally, the manuscript is challenging to follow and critically evaluate, due to a lack of detail in the results, methods and supplemental materials? In particular, a lack of supplemental Excel data tables for results (DEGs, alternative splicing events with statistics, CRISPR dropouts, co-IP MS, gene set enrichments, etc.) and quality metrics (sequencing depth, number replicates, etc.), more detailed algorithm descriptions, supplemental code and source data for figures hinder interpretation by the reviewers and others. The authors should expand their current text and add more specific details in results and methods to clarify confusion. This includes missing callouts for supplemental figures (e.g., Supplemental Fig. 1f) and caveats of the described approaches that are related to sensitivity of the assays used.

We apologize for having been sparse. As requested, we have now added the following:

- Data tables for all DEGs (Supplementary Tables 1 and 7);
- Data tables with global AS events identified versus RBM20-WT iPSC-CMs with rMATS (Supplementary Tables 4, 9), and PSI-values for all exonic parts with annotated exons identified with the pipeline described in⁵⁹ (Supplementary Tables 3, 8);
- CRISPR screens normalized gRNA counts and gene statistics (MAUDE⁶⁷) (Supplementary Tables 10, 11);
- Co-IP mass spectrometry hits (Supplementary Table 6);
- All gene set enrichment analyses in one excel table (Supplementary Tables 2), sheets are named with respective figures;

- Source data for all figure panels in the manuscript (Supplementary Table 14, 15);
- A description of sequencing metrics and more details about computational analyses performed, as well as packages used for figures, in the methods section (pages 24-25).
- We further expanded our current text, adding a more detailed description of our results, especially on pages 4-7 of the results section. See quotes in our response to comment 4.

3 - Ejection fraction and LVIDd for RSRSP should be shown relative to controls and quantified/visualized separately for P633L and controls (Fig. 1e-g).

We thank the reviewer for this comment. We have now indicated a range for normal LVEF (53% to 73%) and LVIDd/BSA (2.4-3.2 cm/m²) based on American Society of Echocardiography guidelines³⁰ in Figure 1 e-f (green shading), and added a description to the main text (Results: page 4 lines 123, 126-127) and figure legend. The ranges indicate normal parameters for healthy individuals. Patients with P633L mutation are already displayed in blue, to distinguish them from patients with other pathogenic/likely pathogenic variants in the RS-domain(black). We further indicate statistical parameters as suggested by Reviewer 2. We hope that these edits made it easier to estimate deviations from healthy norms for P633L or RSRSP patients.

Response Figure 4 (revised Figure 1 e-f). (a) Left ventricular ejection fraction (LVEF) for patients with RBM20-P633L or with other pathogenic or likely pathogenic (P/LP) mutations in the RSRSP-stretch at the time of diagnosis (norm for healthy individuals: 53% to 73%, green shading). Mean LVEF for P633L vs. P/LP variants in RSRSP domain: $52 \pm 4.9\%$ vs $43.1 \pm 13.9\%$, $p=0.059$. (b) Initial LVEF as a function of the age at presentation (norm for healthy individuals: 53% to 73%, green shading). There is no significant correlation between LVEF and age at presentation ($R^2=0.001$, $p=0.89$). (c) Internal Left Ventricular Diastolic Dimension (LVIDd) corrected for body size for analyzed patients (norm for healthy individuals: 2.4-3.2 cm/m², green shading). Mean LVIDd/BSA for P633L vs. P/LP variants in RSRSP domain: $3.2 \pm 0.74\%$ vs $2.8 \pm 0.26\%$, $p=0.28$. Though no significant

difference is identified between P633L and other P/LP variants in the RSRSP domain, all P633L patients have a BSA-indexed LVIDd within normal range.

4 - Excessive claims are reported throughout the manuscript. These include: 1) the relative pathogenic role of mislocalized RBPs versus splicing in disease remains poorly defined (e.g., “...cytoplasmic mislocalization of TDP-43 causes Amyotrophic Lateral Sclerosis”), 2) minor differences discounted as an absence of signal (e.g., “Overall, nuclear-localized P633LRBM20 does not alter gene expression”), 3) partial versus complete rescue of Rbm20 localization by Tmpo3 over-expression on localization (e.g., ...mislocalization of the mutant variants can be rescued by up-regulating the TNPO3-RBM20 interaction.”).

We refined these statements and their conclusions:

1) We corrected the sentence related to TDP-43 and p53 in the introduction: “For example, cytoplasmic mislocalization of p53 facilitates cancer progression³, and cytoplasmic mislocalization of TDP-43 is associated with Amyotrophic Lateral Sclerosis⁴.” (page 2 lines 48-50).

2) As discussed in the Reviewer’s 3 comment 1 and demonstrated on Response Fig. 12, the P633L-nuc fraction does not enrich for cells that show an exclusively nuclear localization of RBM20 (as it is the case for the WT protein). This results in an underestimation of the rescue effect, and thereby gene expression changes between P633L-nuc and WT might be even lower than observed. However, since we cannot exclude the possibility that minor gene expression changes may remain, we adjusted the conclusion as follows: “Overall, nuclear-localized RBM20-P633L caused minor changes in gene expression compared to WT (50 instead of 1,415 for cytoplasmic-localized).” - Results, page 5 lines 148-149.

3) We now clearly state that R634Q mislocalization was partially rescued by TNPO3 overexpression: “While mislocalization of P633L variant could be fully rescued by overexpressing TNPO3, R634Q mislocalization was only partially rescued. This could be due to higher severity of mislocalization seen for this variant (Fig. 1 a-c) and lower binding affinity to TNPO3 (Fig. 4 h-k) (Fig. 5a, b and Supplementary Fig. 10a).” (Results: page 11 line 372-376). We added statistics supporting this statement into the figure legend.

In addition, we adjusted other sentences as follows:

- Results, page 5 lines 162-163: “Altogether, our results demonstrate that nuclear relocalization of P633L at least partially rescues alternative splicing of the core RBM20 targets.”
- Results, page 7 lines 224-226: “We found relatively few (115) differentially expressed genes between WT and NLS-tagged-R634Q expressing cells compared to R634Q cells (1713, Fig. 2d, Supplementary Table 7).”
- Results, page 7 lines 232-234: “These findings suggest that nuclear relocalization of all tested RS-domain RBM20 variants may rescue splicing of TTN, and other core RBM20 targets¹³.”

5 - It would seem as if a genome-wide CRISPR screen with ICS for mutant RBM20 versus wild-type would yield improved targets for therapy. Was this avenue tested and if not, what was the rationale?

We agree that such a screen would generate important insights into RBM20 therapy and strengthen the manuscript. We thus performed a pooled genome-wide CRISPR-Cas9 knockout screen in our RBM20-R634Q Tet:Cas9 HeLa reporter line with ICS and quantified the impact of single gRNAs on nuclear relocalization of the RBM20 variant (Response Figure 5; new Supplementary Figure 6a). We summarized results of this experiment on pages 8-9 lines 270-280 in the Results section. We followed the same experimental and computational procedures as for the RBM20-WT screen.

This screen identified 151 genes, for which a knockout increased nuclear localization of RBM20-R634Q (FDR < 0.01). These genes include potential negative regulators of RBM20's nuclear import, sequestering mutant RBM20 in the cytoplasm (Supplementary Fig. 6 b, c). For example, we identified PPP5C, a serine/threonine phosphatase, and WTAP, a mediator of m6A mRNA modification, both among the top five hits of the screen. As a positive regulator of RBM20 nuclear import we identified nucleoporin NUP37 to be the top hit.

We next tested the effect of single gene knockouts to validate and further characterize the top candidates from this screen based on their FDR values and Z-scores (Supplementary Fig. 6 d,e). We tested 21 out of the 151 candidate regulators with FDR < 0.01, including DDI2, PPP5C, WTAP, EXOSC8, and SRSF3. Unlike KO of TNPO3 from the screen with RBM20-WT described in Fig. 3, perturbation of genes from the new screen with RBM20-R634Q led only to partial nuclear re-localization. These data suggest that, although other factors (like those identified in this study) might potentially have some impact on retaining mutant RBM20 in the

cytoplasm, the level of this effect is relatively small. One could speculate that several proteins in the cytoplasm have affinity to RBM20, thereby collectively contributing to cytoplasmic retention of RBM20-R634Q. However, considering that loss of interaction with TNPO3 showed the described strong effects (Fig. 3 d,e, Fig. 4), interaction with TNPO3 is likely to be the major determinant of cytoplasmic mislocalization of RS-domain variants.

These results were added to the new Supplementary Fig. 6 and described in the main text (Results: pages 8-9 lines 270-280). We added gRNA counts and gene statistics as Supplementary Table 11.

Response Figure 5 (new Supplementary Figure 6). Genome-wide ICS screen with RBM20-R634Q to identify factors that retain mutant RBM20 in the cytoplasm. (a) Schematic outline of the ICS-screen. Five genome-wide lentiviral libraries were applied to HeLa cells expressing eGFP-RBM20-R634Q and TetO-Cas9, with the coverage of 100 cells per gRNA. Cells were sorted based on the correlation between RBM20 and DRAQ5 into 4% higher and 10% lower fractions at final coverage of 500 cells per gRNA per sorted bin. Unsorted input samples were collected for sequencing as well. (b) Reads from collected samples were combined *in silico* to one dataset, with an average of 500 cells per gRNA, 5 gRNAs per gene. Hits were called using the software MAUDE⁶⁶. Genes are ranked by their statistical significance. The horizontal dashed lines indicate an FDR of 1%. Positive/negative regulators with FDR <1% are marked in red and blue, respectively. (c) Scatter plot of fold changes visualizing gRNA abundance changes in higher (x axis) and lower (y axis) sorted bins compared with the plasmid library. Red and blue dots indicate statistically significant positive and negative regulators, respectively (FDR < 5% according to MAUDE). Labelled are regulators selected for future analyses. (d) The impact of single knock-outs of the selected hits from the genome-wide screen (one gRNA per gene; we picked the gRNA that showed the strongest Z-score in the pooled genetic screen) on RBM20 localization tested with ICS. The top row in the heatmap shows the log₁₀ of FDR value for each candidate from the screen. The phenotype in the second row represents the standardized difference in RBM20 localization signal between the knock-out (KO) and control (R634Q) cell populations (log₂ of the ratio between cell fraction with R value for DRAQ5:RBM20 correlation >

0.7 in the knockout divided by cell fraction with R value for DRAQ5:RBM20 correlation > in the control). (e) ICS analysis of eGFP- RBM20- R634Q localization based on DRAQ5:RBM20 correlation upon indicated single gene knock-outs.

Minor concerns:

1 - For supplementary splicing event files provided with the manuscript and in GEO, please include the genomic coordinates of the associated junctions to enable identification by the reader. Currently the IDs are of the type: *ENSG00000000003:001*, which appears to point to an isoform or specific exon.

Coordinates for the analyzed junctions are now included in all the supplementary tables listing AS analyses (Supplementary Tables 3, 4, 5, 8, 9).

2 - An additional panel in Supplementary Figure 1 indicating the correlation of *IMMT* splicing and *RBM20* cytoplasmic versus the nuclear localization is likely to better illustrate the correspondence between *RBM20* mutant splicing impacts and localization.

We added the requested plot to Supplementary Fig. 1g, and mentioned the result in the Results section on page 4 lines 113-114: “*IMMT* splicing measured by qPCR correlated with *RBM20* nuclear localization (Supplementary Fig. 1g).”

Response Figure 6 (new Supplementary Figure 1g). Means of *IMMT* splicing fold change (data from panel (f), y axis) and means of Pearson correlation coefficient R for DAPI:RBM20 correlation (data from panel (c), x axis). Linear regression between the two is displayed as a dashed line, and regression standard error is displayed in grey. Standard errors are indicated as black solid lines.

3 - Can the authors clarify whether they considered known, novel and alternative promoter regulation in their PSI analyses? Supplemental Fig. 2g shows some categories but this is not clear from the methods or text. The description of splicing analyses in the methods section is also vague. Clarity will help readers not familiar with the concept of junction-based local splicing variation approaches.

For all the analyses done with rMATS, the pipeline considers AS events with alternative 3' and 5' mRNA ends. Therefore, alternative promoters are considered in plots illustrated in Supplementary Fig. 2g-i, Supplementary Fig. 4g-i, and Response Fig. 1. Heatmaps of Fig. 1j and Fig. 2e only focus on the list of the core RBM20 targets (see the rationale above), without distinguishing the splicing categories. We updated the methods section (page 24 lines 811-814).

4 - It is surprising the authors did not compare their MS co-IP data to that of Maatz et al. 2014, which compared S635A to WT interactions in HEK293 cells. These analyses also show a WT enrichment of RBMX. The authors should consider this data as a reference for their interactions (do they find the same or different associations).

We thank the reviewer for this suggestion. We added an additional column to the Supplementary table 6 stating whether a protein identified as enriched over negative control for each sample in our study was also identified by Maatz et al.¹⁹.

The reason why we did not add a comparison with Maatz et al. in the first version of the manuscript is that Maatz et al. did not perform subcellular fractionation. Therefore, our and their datasets have limited comparability. We agree, nonetheless, that comparing our nuclear fraction of RBM20-WT sample to their RBM20-WT sample is possible, since RBM20-WT displays solely nuclear localization. We now compared the number of overlapping protein hits in the datasets (our nuclear-WT and WT from Maatz et al.) and summarize the findings for the reviewer:

- Of the 98 proteins that we identified in the RBM20-WT immunoprecipitation in the nuclear fraction, 43 were also identified by Maatz et al., and 18 of those were enriched with WT protein over S635A mutant by Maatz et al. Only four of those lost the interaction with mutant variants in the nucleus in our study (Supplementary Fig. 3d) namely – SRP14, RBM14, RBMX, and RBM15. These may hint on the direct impact of RS-domain mutations on protein-protein interactions in the nucleus, excluding the effects of intracellular protein localization.
- Out of 141 proteins binding to R634Q in the cytoplasm, 63 were also identified by Maatz et al., but only 4 of those were classified as enriched with S635A variant by Maatz et al.

Based on these analyses, we added the following sentence to the main text (Results: page 6, lines 178-182): “Only 17 proteins mildly lost their ability to bind the mutated proteins (Supplementary Fig. 3d), four of which were also identified by Maatz *et al.*¹⁹ as enriched with WT protein (SRP14, RBM14, RBMX, and RBM15), and only one of them (RBMX) was a component of the spliceosome³² (Supplementary Table 6).”

5 - The RSS combined mutant is first described in Supplemental Figure 3a but not defined until much later. The authors should define in the figure legend of the first mention.

We added the definition of RSS to the figure legend of Supplementary Fig. 1, where it is first mentioned.

6 - The authors should replace “few” with “relatively few” in the sentence “Strikingly, we found only few differentially expressed genes between WT and NLS-tagged-R634Q expressing cells” as 110 genes are differentially regulated.

We have made the suggested change: “We found relatively few (115) differentially expressed genes between WT and NLS-tagged-R634Q expressing cells (Fig. 2d, Supplementary Table 7).”

Reviewer 2:

The manuscript by Kornienko and colleagues describe an important novel regulatory step in the pathogenesis of RBM20 DCM: an impaired interaction of RBM20 mutated in the RS domain hotspot with the nuclear importer TNPO3. This discovery is achieved through the combination of studies in both immortalized cells and in more sophisticated cellular models (hiPSC-CMs with either patient-derived mutations or transduced with mutant RBM20) and the application of state of the art omics techniques. The application of ICS is particularly clever and is one of the first examples of how this approach can empower medically-relevant discovery. The experiments are generally comprehensive, well executed, and properly analyzed. The study key findings are novel and of strong relevance to the CVD field and beyond. I have however some important reservations about how some experiments are being (over)interpreted leading to strong conclusions on the effect of RS domain mutations that in

my view are not fully supported by the data. Addressing these and other secondary comments would strengthen the study and broaden its impact.

We thank the reviewer for their positive feedback on the relevance of our work. We addressed the reviewer's reservations below.

1 - Through the experiments described in Figure 2 (and associated Supplementary Figures) the authors conclude that "all tested RS-domain RBM20 mutations do not affect the intrinsic splice regulatory function of the protein". However, this conclusion is based on non-physiological systems: either the non CM HEK293T cells, or CMs infected with lentiviruses encoding RBM20 through a strong TetO promoter and thus leading to an excess of RBM20 expression compared to the endogenous levels. This is apparent from the IF in Fig. 2C, whereby many RBM20 nuclear foci are formed in WT-infected CMs: this is in contrast with the existence of usually only 2 foci in CMs close by to the TTN loci, as observed in several studies (more foci are observed only in polyploid mature CMs, which are rarely seen in hiPSC-CMs cultures). Of note, such RBM20 foci have been shown to be sites of more efficient AS regulation (PMID: 30948719). Thus, the experiments presented here do not rule out that under normal (and limiting) amounts of RBM20 the RS mutation may affect the formation of RBM20 foci (i.e. through impaired interaction with the TTN pre-mRNA or with other splicing-regulating RNAs/proteins). The conclusion of Figure 2 should therefore be either toned down (with the aforementioned possibility being discussed) or supported by data whereby NLS-carrying RBM20 mutant proteins are expressed in physiological levels. As a matter of fact the author's own data presented in Figure 5 c-d indicate that rescuing mutant RBM20 localization is not sufficient to completely restore the AS of RBM20 targets, which is consistent with a role of RS mutations besides that of impaired nuclear import.

We are thankful for this comment. As the reviewer mentioned, RBM20 usually forms on average two characteristic foci in the nucleus under physiological expression levels in human iPSC-CMs, which was also observed in this manuscript (Fig. 1a). This was also observed for the RBM20-P633L and -R634Q variants upon TNPO3 overexpression (Fig. 5 a, g, Supplementary Fig. 10). This suggests that these two RS-domain variants do not affect nuclear foci formation when expressed at physiological levels upon TNPO3 overexpression. We now highlight this in text (Results page: 11 lines 373-374: “Both variants were able to form two characteristic nuclear foci when relocated to the nucleus”

In addition, data from Fig. 5 b-d demonstrate that TNPO3 overexpression restores P633L localization, as well as splicing of *TTN* and *IMMT* to a level that is insignificantly different from that detected in WT cells (Fig. 5b: $p = 0.074$, Fig. 5c: $p = 1.000$, Fig. 5d: $p = 0.210$, t-test). In the case of R634Q+TNPO3 overexpression, localization and splicing were only partially restored: splicing differences were reduced upon TNPO3 overexpression but still significantly different to WT cells (Fig. 5b: $p < 0.001$, Fig. 5c: $p = 0.033$, Fig. 5d: $p < 0.001$). We added a sentence mentioning these comparisons in the main text, and p-values for these pairwise comparisons were added to the figure legend: Results section, pages 11-12 lines 378-381: “Efficient restoration of nuclear localization of RBM20-P633L also resulted in rescue of *TTN* and *IMMT* splicing to the levels seen in RBM20-WT cells, while partial relocalization of RBM20-R634Q was associated with a proportional restoration of splicing function (Fig. 5 c, d).”

We agree with the reviewer that our results described in Figure 5 only provide first insights for these two variants and do not fully rule out potential effects of RS-domain mutations on nuclear foci formation and splicing, especially for other variants. In addition, results described in Figure 2 and Supplementary Figure 4 are, indeed, based on RBM20 overexpression and non-cardiomyocyte model systems. We now discuss these limitations and adjusted our conclusions of Figure 2 and Supplementary Figure 4 as follows:

- Results, page 7 lines 232-236: “These findings suggest that nuclear relocalization of all tested RS-domain *RBM20* variants may rescue splicing of *TTN*, and other core *RBM20* targets¹³. However, this conclusion is based on ectopic expression of *RBM20* variants, and further investigations are needed to characterize splicing restoration under physiological expression levels in cardiomyocytes.”
- Discussion, page 12 lines 405-406: “Future studies with endogenously expressed *RBM20* variants will identify mRNA targets and protein interactors of different relocalized RS-domain variants.”

2 - "This concludes that loss of interaction with TNPO3 results in formation of RBM20 RNP granules, regardless of the mutation present" and "We show that WT RBM20, when forced to remain in the cytoplasm, also forms granules of the same nature as the mutant variant". Similar to my previous comment, this conclusion should be toned down or supported by further data.

While the authors convincingly show that WT RBM20 in TNPO3 KD cells form some kind of cytoplasmic granules, these are not well characterized in CMs while experiments in HeLa are interesting but not conclusive of the endogenous behaviour. Considering that studies such as refs 24 and 27 have suggested that bona fide RBM20 mutant RNP granules in adult CMs may be pathogenic in their own right this is not just a matter of semantics or curiosity. I encourage the authors to acknowledge that the exact nature of RBM20 RNP granules may still be dependent on RS mutations unless they are willing and able to establish the contrary through more rigorous characterization of TNPO3 loss-of-function CMs.

We agree with the reviewer, and we adjusted our conclusions as follows:

- Results, page 11 lines 364-365: “This suggests that loss of interaction with TNPO3 results in formation of RNP granules detected for both WT or RS-domain mutated RBM20.”

Discussion, page 13 lines 434-438: “We show that RBM20-WT, when forced to remain in the cytoplasm, also forms similar granules as the mutant variants. Although the exact nature of RNP granules formed by WT or RS-domain mutated RBM20 in cardiomyocytes needs to be addressed in future studies, our results suggest that RS-domain mutations of *RBM20* do not confer pro-aggregative qualities. “

Indeed, the current study was not aimed at dissecting the exact nature and impact of RNP granules formed by different RS-domain variants. Instead, we focused on deciphering the mechanism of RBM20 nuclear import and its mislocalization, and our finding that RBM20-WT forms cytoplasmic granules in iPSC-CMs if the level of TNPO3 is reduced is solely descriptive.

Secondary comments:

I - What's the authors' explanation as to why only a subset of P633L RBM20 mutant hiPSC-CMs show marked cytoplasmic mislocalization? Even more importantly, can they be sure that the P633L-nuclear cells truly express the mutant RBM20 at comparable levels as P633-cyt? Barring issues with the hiPSC clonality, which should be excluded if not done so already, a potential explanation for the heterogeneity may be due to different transcriptional activity of the two RBM20 alleles: could the authors test this by probing the mutant/wt allele ratio at the mRNA level in the two cell populations? If this was not similar it may explain the marked

differences in gene expression alterations, not allowing to conclude that these are only due to differential (stochastic?) mislocalization.

We thank the reviewer for this comment. We used only homozygous P633L iPSC-CMs in our manuscript, so no WT allele is present in these cells according to Sanger sequencing analysis performed on these cells done by Briganti et al., 2020, and repeated by us prior to initiation of these experiments (Response Figure 7). It can be seen that no WT allele is present in these cells.

Response Figure 7. Sanger sequencing of RBM20-P633L iPSC-CMs.

We didn't identify statistically significant differences in *RBM20* or *TNPO3* gene expression levels between P633L-nuc vs P633L-cyt cells or between different RBM20 mutants (Supplementary Fig. 1a, Supplementary Fig. 7b, Supplementary Fig. 8c). Based on our data, we suppose that the observed heterogeneous localization pattern of RBM20-P633L can be explained by only partial loss of its interaction with TNPO3, which we demonstrate in Figure 4. This results in a heterogeneous mislocalization pattern, where, in some cells, RBM20 makes it more to the nucleus than in others, while in some cells less. Of note, most P633L cells indeed have a mixed localization pattern (see Response Fig. 12 for HeLa cells). We now mention this in the main text:

- Results, page 4 lines 131-133: “The mixed phenotype (nuclear and cytoplasmic) of **homozygous RBM20-P633L iPSC-CMs** allowed us to differentiate between the consequences of nuclear and cytoplasmic RBM20 localization in the same genetic background.”
- Page 4 lines 137-139: “Using ICS, we sorted iPSC-CMs with differentially-localized RBM20 from **homozygous P633L mutation background** based on correlation with nuclear staining (Fig. 1h and Supplementary Fig. 2a).”

2 - Data from Figure 4a and b in WT CMs are not incredibly convincing, possibly due to difficulties in homogeneously KD TNPO3 with siRNA. For instance I wonder whether cells in

which the correlation of DRAQ5 and RBM20 is unchanged after TNPO3 siRNA (peak overlapping with siRNA control in the bimodal distribution of TNPO3 siRNA cells) have not been transfected/KD or else are cells whereby other mechanisms are at play bypassing the requirement for TNPO3. Given the importance of validating the role of TNPO3 in the endogenous context of CMs I recommend that the author consider performing a more robust and ideally isogenic experiment using a KO or, should this not be compatible with iPSC-CM differentiation, and inducible KD/KO

Indeed, homogenous knockdown and knockout of *TNPO3* in an isogenic cell line are challenging since *TNPO3* is essential. We found that *TNPO3* targeting sgRNAs were strongly depleted from our genome-wide ICS screen, as expected for an essential gene (Supplementary Table 10). Furthermore, published genome-wide CRISPR screens classify *TNPO3* as pan-essential gene meaning that a loss of *TNPO3* leads to loss of fitness or cell death in multiple normal tissues and cell lineages (DepMap (Dempster et al., *Genome Biol*, 2021; Meyers et al., *Nat Genet*, 2019), <https://depmap.org/portal/gene/TNPO3?tab=dependency>). This excludes the possibility to test another cell model system where *TNPO3* is not essential but RBM20 is expressed.

We therefore performed a *TNPO3* knockout experiment in iPSC-CMs stably expressing Cas9, followed by quantification of endogenous *TNPO3* signal and the localization of endogenous RBM20 from microscopy data. We optimized the knockout conditions in order to cope with the essentiality effects of *TNPO3* knockout but yield sufficient cell numbers with decreased *TNPO3* protein levels (see methods, page 19-20, lines 639-656). First, we directly transfected iPSC-CMs with a mix of three *TNPO3* targeting sgRNAs instead of generating loss-of-function mutations in iPSCs before differentiation. Second, we performed a double-transfection of the sgRNA mix, with a first transfection on day 0, and second transfection on day 4, which overall increased the number of cells showing decreased *TNPO3* protein levels. Third, we performed microscopy analysis early (3 days) after the last transfection - before strong essentiality effects lead to a complete loss of perturbed cells. We then fixed the cells and stained them with anti-RBM20 and anti-*TNPO3* antibodies (Supplementary Table 20). We found one *TNPO3* antibody to provide a specific enough signal as validated by a loss in *TNPO3* signal upon *TNPO3* sgRNA transfection (Response Figure 8 a-d). We observed only minor background signal in unstained cells (Response Fig. 8 a). Cells that were not transfected with *TNPO3* sgRNAs did not show a correlation between RBM20 localization (DAPI:RBM20 signal

correlation) and TNPO3 levels (Response Fig. 8 d). However, when cells were transfected with *TNPO3* sgRNAs, we observed a decrease in TNPO3 levels that correlated with a decrease in DAPI:RBM20 colocalization (Response Fig. 8 c,d). In addition to a decrease in TNPO3 protein levels upon sgRNA transfection, cell density strongly decreased compared to the non-transfection control. This indicates the essential role of TNPO3 expression and explains the sparse presence of cells that show TNPO3 depletion (Response Fig. 8 a-c). Taken together, this data shows that TNPO3 expression levels directly correlate with RBM20 nuclear localization in cardiomyocytes. To further support this statement, we plotted TNPO3 expression and RBM20 localization from our TNPO3 overexpression data in Fig. 5a (Response Figure 9, new Supplementary Figure 10 b). Again, we see a clear correlation between RBM20 nuclear localization and TNPO3 expression, which further suggests that the bimodal distribution seen in Figure 4 a-b is presumably caused by the inefficiency of the KD.

To test whether some cells could exploit a secondary RBM20 import mechanism in response to TNPO3 loss, we further classified cells based on TNPO3 levels (Response Figure 8d) as “KO” (average pixel intensity of TNPO3 staining < 1750 - below the lowest level observed in non-transfected cells). Comparison of classified TNPO3 KO cells and non-transfected cells showed that all KO cells display mis-localization of RBM20 to the cytoplasm (Response Fig. 8e). This data strongly suggests no additional bypassing mechanism of RBM20 nuclear import.

We added the results of this experiment as a new Supplementary Figure 7, and discussed them in the manuscript as follows:

Results, page 9 lines 291-299: “To assess the role of TNPO3 in the nuclear transport of RBM20 in iPSC-CMs, we performed siRNA knock-down (KD) and CRISPR/Cas9-based KO of *TNPO3* (Fig 4 a, b and Supplementary Fig. 7 a, d-f). We found that *TNPO3* KD significantly decreased nuclear localization of both RBM20-WT and -P633L. This was accompanied by a decrease of *TTN* and *IMMT* alternative splicing upon *TNPO3* KD (Fig. 4c and Supplementary Fig. 7c). The degree of RBM20-WT mislocalization correlated with TNPO3 levels upon *TNPO3* KO (Supplementary Figure 7 g). Cells with decreased TNPO3 levels showed a significant shift in RBM20 localization to the cytoplasm compared to non-transfected cells (Supplementary Fig. 7 h). This confirms that TNPO3 is essential for RBM20 nuclear import in iPSC-CMs. “.

Results, page 11 lines 376-378: “Both variants were able to form two characteristic nuclear foci when relocated to the nucleus, and the degree of their nuclear relocalization correlated with the level of TNPO3 expression (Fig. 5a, Supplementary Fig. 10 a, b).”

Discussion, page 12 lines 410-412: “We identify TNPO3 as the main nuclear importer of RBM20 with the genome-wide ICS CRISPR-Cas9 screen and validate its essentiality for nuclear import of endogenous RBM20 in iPSC-CMs.”

Response Figure 8. *TNPO3* KO in iPSC-CMs (new Supplementary Figure 7 d-h). (a) A representative image of iPSC-CMs stably expressing Cas9, not transfected with sgRNAs, and not stained with anti-TNPO3 antibody. (b) A representative image of iPSC-CMs stably expressing Cas9, not transfected with sgRNAs, stained with anti-TNPO3 antibody. (c) Representative images of iPSC-CMs stably expressing Cas9, transfected with a mix of three sgRNAs targeting *TNPO3*, stained with anti-TNPO3 antibody, with close-ups of selected areas indicated. (d) Correlation between DAPI:RBM20 co-localization and TNPO3 expression in non-transfected and transfected

with sgRNAs targeting *TNPO3* iPSC-CMs. Linear regressions between the two are displayed as a dashed line, and regression standard errors are displayed in grey. Correlation coefficient = -0.09, p-value = 0.71 for non-transfected cells, and correlation coefficient = 0.69, p-value = 4.79e-06 for transfected cells. Each dot represents a single cell. (e) DAPI:RBM20 co-localization in non-transfected cells (average pixel intensity of TNPO3 staining > 1750) and in transfected cells with average pixel intensity of TNPO3 staining < 1750 (referred as “KO. **** - p < 0.0001 Student’s T-test.

Response Figure 9. TNPO3 OE in iPSC-CMs (new Supplementary Fig. 9b). Correlation between nuclear localization of RBM20 (DAPI:RBM20 correlation) and eGFP-TNPO3 fluorescence level (average pixel intensity) based on microscopy data from Supplementary Figure 9 panel (a) for RBM20-P633L and -R634Q iPSC-CMs transfected with eGFP-TNPO3. Each dot represents a single cell. Linear regression between the two is displayed as a dashed line, and regression standard error is displayed in grey. Correlation coefficients = 0.71, 0.75, p-values = 0.0002, 6.358e-06, for P633L and R634Q, respectively.

Minor comments:

1 - Dots indicating individual data points are too small in several figure panels (i.e. 1b, 1d, 1k, ...). If a reasonably sized dot plot is too crowded perhaps a violin plot would be most suited.

We now changed sizes of data points in all plots mentioned by the reviewer, as well as in other plots that we found it to be reasonable.

2 - Figure 1b: was the correlation between DAPI and RBM20 calculated on 3D reconstructions based on Z-stacks? Either way should it be feasible for the authors to provide super-resolution imaging for RBM20, as done by the other papers cited in refs 24 and 27, it would add insight to the mislocalization phenotype and how it compares with other RS mutations.

The reviewer was correct, the correlation was indeed quantified based on Z-stacks of 3D reconstructions. We mentioned this in methods section (page 22 lines 728-731).

While super-resolution imaging would provide insights into ultrastructural similarity between foci formed by different RBM20 variants, our study didn't focus on potential differences between RBM20 mutations. Ultrastructural analysis might provide indirect insights into foci structure, but without addressing potential functional differences. Importantly, even in the case of an absence of ultrastructural differences, we wouldn't be able to exclude functional differences between RBM20 foci. Although we agree that more information on the functional and ultrastructural differences between cytoplasmic foci formed by RBM20 mutants would be relevant, this was not at the center of this work. Instead, we focused on localization differences and the influence of nucleo-cytoplasmic transport on RBM20 mutant function. We also want to refer to our response to reviewer 1 comment 1, who asked if different variants affect splicing in different ways. We added the following to the discussion page 12 lines 406-408: "In addition, ultrastructural studies of cytoplasmic foci caused by different RS-domain variants could elucidate functional differences of these foci."

3 - Figure 1e: the authors indicate that LVEF in P633L patients was less severe than other P/LP variants - is this supported by statistical analyses?

We now additionally present the following summary statistics in the figure legend:

(e). Mean LVEF for P633L vs. P/LP variants in RSRSP domain: 52 ± 4.9 % vs 43.1 ± 13.9 %, $p=0.059$.

(f). There is no significant correlation between LVEF and age at presentation ($R^2=0.001$, $p=0.89$).

(g). Mean LVIDd/BSA for P633L vs. P/LP variants in RSRSP domain: 3.2 ± 0.74 % vs 2.8 ± 0.26 %, $p=0.28$. Though no significant difference is identified between P633L and other P/LP variants in the RSRSP domain, all P633L patients have a BSA-indexed LVIDd within normal range.

4 - The proteomic analyses were performed in non-CM cells: this is an understandable experimental choice but also a limitation that ought to be discussed. For instance, HeLa cells lack TTN and hence the nuclear foci observed therein are unlikely to have the same

composition of splicing factors and other proteins (besides the fact that CM-specific proteins are not expressed in HeLa)

We now mentioned this limitation in the manuscript:

- Results, page 6 line 184-185: “These results indicate that RS-domain variants exhibit mainly unaltered interactors if located to the nucleus in HeLa cells.”
- Discussion, page 12 lines 405-406: “Future studies with endogenously expressed RBM20 variants will identify mRNA targets and protein interactors of different relocalized RS-domain variants.”

5 - Figure 3d: what was the rationale for knocking out TTN in HeLa cells? It should not be expressed. Which of the other genes selected ad hoc (AKT2, SPRK1, CLK1, LMNA) are expressed? In general it would be useful if the authors could present data comparing expression in HeLa vs CMs for the genes hits from the screen (including the discarded negative regulators), and comment on any differences

Regarding the rationale for knocking out *TTN* in HeLa: We performed a knockout of *TTN* because we indeed detected *TTN* as a hit in our screen (FDR = 0.137, gene rank = 21 from positive regulators Fig. 3, Supplementary table 10). We were also surprised to see such a result, and therefore tested whether we could validate this hit with a single KO experiment. The *TTN* validation experiment revealed no observable effect (Fig. 3d). As can be seen on Response Figure 10, a single gRNA from library 6 dropped out from both input and upper fraction samples, which affected the applied hit calling.

Response Figure 10. Normalized gRNA read counts for *TTN* for each of the sub libraries (CRISPR screen Fig. 3, Supplementary Table 10).

We now performed RNA sequencing of HeLa Tet::Cas9 cells with eGFP-RBM20-WT and eGFP-RBM20-R634Q, and compared gene expression levels between them and iPSC-CMs

with RBM20-WT and -R634Q (data from Fig. 1). For all plots below, we show normalized read counts after processing raw read counts with DeSeq2⁵⁶. All but two hits (*CLDN14* and *FOXJ1*) from both screens (17 from the WT screen Fig. 3 and 21 from R634Q screen, supplementary Fig. 6) are expressed in iPSC-CMs (Response Fig. 11). We considered a gene to be expressed if normalized read counts were above 50 on average amongst all samples and replicates. The levels of gene expression differed for most of the tested genes when compared between cell types. The observed high variability in gene expression levels is to be expected due to completely different origins of these cells. The only insignificant (adjusted p-value > 0.01) differences were observed for four genes, namely *PMM2*, *DDI2*, *SNAPC4*, and *RBM39*. As expected, normalized read counts for *TTN* were below 15 in all replicates of HeLa cells (Response Fig. 11 b).

As requested, we:

- Illustrate normalized read counts for positive (Fig. 3 d) and negative (Supplementary Fig. 6 d) hits from HeLa and iPSC-CMs (Response Figure 11, see below).
- Provide normalized read counts (with DeSeq2) in HeLa and iPSC-CMs per every hit tested (Supplementary Table 12)
- provide raw read counts per gene for HeLa and iPSC-CMs in Supplementary Table 13
- add a short description of library preparation for HeLa cell samples (page 24);
- add a short summary of the results on page 9, lines 280-282: “Importantly, expression of all but two (*CLDN14* and *FOXJ1*) tested hits from both screens was detected in iPSC-CMs by RNA-sequencing, confirming the relevance of identified candidates (Supplementary Table 12).”

Response Figure 11. Normalized read counts (DeSeq2) in iPSC-CMs and HeLa. (a-c) for positive hits (from Fig. 3d), and (d) for *NUP37* and negative hits (from Supplementary Fig. 6d). Only four out of 38 genes were changed insignificantly between HeLa and iPSC-CMs (labelled ns). Only two tested hits (*CLDN14* and *FOXJ1*) were expressed at the level below 50 normalized counts in iPSC-CMs. Normalized read counts for *TTN* (b) and *LMNA* (c) are plotted separately, due to much higher level of their expression.

This new RNA-seq dataset should provide a resource to compare gene expression between the HeLa and iPSC-CM model systems exhibiting the same RBM20 genotype.

6 - Supplementary Fig. 7a highlights the intrinsically disordered nature of most of RBM20, which results in a highly unreliable structural prediction. outside of the RRM domain: this limitation should be made clearer in the text

We now shifted the critical assessment of the limitations of structural predictions from the Material and Methods section to the main text to emphasize clearly for all readers the constraints of the structural predictions (Results, page 10 lines 321-326): “All obtained predictions for RBM20’s RS-domain had low pLLDT scores, which indicated the intrinsically disordered nature of this region. The obtained Predicted Aligned Error (PAE) plots of TNPO3 in complex with the RRM-RS domain of RBM20 suggest that both proteins are presumably incorrectly located relative to each other and are not ideal for isolated interpretation of a single model (Supplementary Fig. 9a). “

7 - AAV9-TNPO3 is presented as "a promising therapeutic avenue", but what are the potential side effects? Are proteins other than RBM20 more efficiently nuclear-imported? What are the known TNPO3 targets expressed in CMs?

We thank the reviewer for their comment. In our manuscript, we identify TNPO3 as nuclear import factor that targets RBM20 into the nucleus, and that the direct interaction between them is affected by several DCM-causing mutations in *RBM20*. The demonstrated proof-of-principle experiment of TNPO3 overexpression shows that the TNPO3-RBM20 interaction serves as potential therapeutic target. However, increased nuclear transport can be induced by other means than TNPO3 overexpression, such as increasing interaction of mutated RBM20 and TNPO3 with aptamers, bifunctional antibodies, small molecules etc. We already outlined these potential avenues in the discussion (page 14, lines 449-453): “Enhancing RBM20 nuclear import could be achieved by other means, like endogenous tagging of *RBM20* with another NLS to be recognized by other importins. Alternatively, aptamers, bifunctional antibodies, or small molecule drugs that bind allosteric sites in RBM20, could be explored as therapeutic strategy that increases the affinity of mutant RBM20 to TNPO3. “

To make this point clearer and to estimate potential side effects by TNPO3 overexpression, we reanalyzed a list of TNPO3 interactors from PHAROS⁵⁰, and identified TNPO3 interactors that are expressed in iPSC-CMs (using the RNA-seq data generated above). The list comprises of 160 proteins, not including RBM20. Out of these, 130 had an average read count above 50

across six samples (three RBM20-WT and three RBM20-R634Q). This indicates that the majority of TNPO3 interaction partners are expressed in iPSC-CMs, and therefore, presumably, in heart tissue as well. Therefore, it is likely that *TNPO3* overexpression may affect the localization of other targets that are expressed in cardiomyocytes. Whether TNPO3 overexpression could lead to a more stringent nuclear localization of any of these factors, or result in other side effects is currently unknown. This information was added to the discussion: pages 13-14 lines 445-449: “Altogether, our data reveals a new therapeutic avenue for DCM patients with disease causing variants in the RS-rich region of *RBM20*. Since the majority of TNPO3 targets⁵⁰ were detectably expressed in iPSC-CMs (Supplementary Table 13), direct overexpression of *TNPO3* may affect their nuclear import resulting in potential side-effects of therapeutic *TNPO3* overexpression.”

8 - Data availability: the authors indicate that "raw data will be available upon request" but I believe that the editorial policies of the journal require deposition to open access servers. Along the same lines I could not examine the raw/processed data, though I see no obvious issues in their analyses based on the methods described

We have changed the data availability statement and made all raw data available:

Raw and processed RNA-sequencing and ICS screen data is deposited at GEO. Raw and processed proteomics data is deposited at PRIDE.

GEO: GSE220833; Reviewer access token: sbudeasoblmjxez

PRIDE: PXD038790; Reviewer account details:

Username: reviewer_pxd038790@ebi.ac.uk; Password: MZiHxYVY

All structural predictions are available upon request.

Reviewer 3:

In this manuscript, Kornienko et al. study the mechanisms underlying the mislocalization and function of pathogenic RBM20 variants. The authors found that nuclear targeting of RBM20 mutant proteins restores their splicing function. Using a recently developed image-enabled cell sorting (ICS) technology of cells expressing RBM20 variants, which are found either exclusively in the cytoplasm (R634Q) or in both the nucleus and cytoplasm (P633L), the authors uncovered TNPO3 as a main nuclear import receptor for RBM20. Overexpression of

TNPO3 in vitro and in vivo partially restored the nuclear localization and splicing defects caused by RBM20 P635L mutation.

Overall the data are clear, and important, and raise interesting questions for future research. However, there are a few points that would improve the manuscript as follows:

We thank the reviewer for their positive assessment of our work. We addressed the reviewer's concerns below.

Major points

1 - In Figure 1i, a number of differentially expressed genes are presented. While the effect of P633L-cyt and P633L-nuc as well as NLS-mediated targeting of the R634Q variant to the nucleus (Figure 2d) on differential gene expression is very convincing and striking, the heatmap in Figure 1j representing the differential splicing events is not that clear. Half of TTN differentially spliced exons do not appear to be different in P633L-cyt and P633L-nuc, also the effect on other splice targets can not be clearly seen. In Supplementary Figure 2g,h a number of mis-splicing events can be detected in P633L-nuc, which somewhat argues against the conclusion that nuclear targeting of RBM20 mutant proteins restores their splicing function. These results are in contrast to the very clear rescue effect on differentially expressed genes and bring forward the question of whether nuclear targeting of RBM20 mainly rescues other steps of post-/transcriptional control.

We thank the reviewer for raising this concern. Indeed, we observe a stronger rescue of gene expression than splicing upon nuclear relocalization (Fig. 1 i,j; Fig. 2 d,e). It has been shown^{25,26,27} that changes in gene expression are likely caused solely by cytoplasmic mislocalization of RBM20 variants. Upon full *RBM20* KO, gene expression was largely unchanged, unlike splicing. In addition, splicing was shown to be dose-dependent on the amount of RBM20-WT expressed²⁴. These results suggest that gene expression changes can presumably be rescued by only removal of RBM20 variants from the cytoplasm, while to restore splicing to the WT level, the amount of the protein in the nucleus has to be at the WT level too.

In the ICS RNA-seq experiment described in Figure 1 and Supplementary Fig. 2, P633L-RBM20 is expressed at endogenous level for both fractions, and, therefore, the results are

independent of the amount of the protein expressed. However, image-enabled cell sorting only enriches for a desired population, leaving some level of impurity. Response Fig. 12 displays histograms of DRAQ5:RBM20 correlations of HeLa cells expressing P633L-RBM20 pre- (left panel), and post-sort (middle panel). For illustration purposes, we also added a histogram for RBM20-WT cells. It is clearly seen that, although ICS does enrich for the desired populations, which is sufficient to reliably identify regulators of protein transport⁵, P633L-nuc cells do not exhibit fully nuclear localization like RBM20-WT cells do. This means that all rescue effects in P633L-nuc are presumably underestimated. On the other hand, in the RNA-sequencing experiment described in Figure 2 and Supplementary Fig. 4, we used lentiviral delivery of WT, R634Q, or NLS-R634Q to iPSC-CMs with a frame shift mutation in RBM20 leading to loss of its splicing functionality²². Here, we get a clear nuclear localization of NLS-R634Q, and, therefore, do not have the limitation described above. However, lentiviral expression of RBM20 is not endogenous. Higher levels of RBM20 present in the nucleus or cytoplasm may result in association with novel mRNAs only due to the elevated dose of the protein.

Response Figure 12. DRAQ5:RBM20 correlations measured with ICS for HeLa cells expressing RBM20- P633L pre-sort (left), post-sort into P633L-nuc and P633L-cyt fractions (middle), as well HeLa cells expressing RBM20-WT (right).

We agree with the reviewer, that based on our data and due to these limitations, it is not possible to confirm that a full rescue of RBM20 localization would fully rescue splicing. In order to provide the reader with a clearer assessment of the effect of RBM20 relocation on splicing of the core RBM20 targets¹³ (see details in methods and in response to reviewer 1 comment 1) in our experiments, we now provide:

- **the number of significant AS events** calculated with rMATS ($|\text{dPSI}| > 0.1$; $\text{fdr} < 0.1$) in the core RBM20 targets (Supplementary Fig. 2h): **we detected 71 AS events in P633Lcyt vs WT, and 62 of them (87%) were not detected as significantly changed in P633Lnuc vs WT.**

- **the number of significant AS events** ($|\text{dPSI}| > 0.1$; $\text{fdr} < 0.1$) in the core RBM20 targets (Supplementary Figure 4h) for **R634Q vs WT was 22, and 21 of them (95%) were not detected as significantly changed in NLS-R634Q vs WT.**
- absolute numbers for Fig. 1j: from 79 AS events that were significantly ($|\text{dPSI}| > 0.1$ and $\text{p-value} < 0.05$, t-test) changed in R634Q compared to WT, 72% (57 of 79) were significantly changed in P633L-cyt vs WT, and only 36% (29 of 79) in P633L-nuc compared to WT. This means that, approx. 64% of differential AS events from R634Q were restored in P633L-nuc.
- absolute numbers of AS events in the core RBM20 targets, PSI-values of which were **restored in P633L-nuc over P633L cyt: 67%** of AS events that were significantly changed ($|\text{dPSI}| > 0.1$ and $\text{p-value} < 0.05$, t-test) in P633L-cyt vs WT were insignificantly changed in P633L-nuc vs WT ($|\text{dPSI}| < 0.1$ or $\text{p-value} > 0.05$, t-test).
- absolute numbers for Fig. 2e: from 153 AS events that were significantly ($|\text{dPSI}| > 0.1$ and $\text{p-value} < 0.05$, t-test) changed in R634Q compared to WT, **80% (122 of 153) were restored** (insignificant changes) **in NLS-R634Q.**

We mentioned these numbers and further adjusted some of our conclusions the main text as follows:

- Introduction: page 3 lines 93-94: “In this study, we show that pathogenic RS-domain variants do not **disrupt** the splice regulatory activity, and that the splicing defect is **mainly** due to mislocalization of RBM20.”
- Results, page 5 lines 159-161: “In contrast, in WT and P633L-nuc cells, they were **predominantly spliced-out (67% AS events restored in P633L-nuc from those mis-spliced in P633L-cyt).**”
- Results: page 5 lines 162-163: “Altogether, our results demonstrate that **nuclear relocation of P633L at least partially rescues alternative splicing of the core RBM20 targets.**”
- Results, page 7 lines 226-228: “Moreover, splicing of the **core RBM20 targets** was restored in NLS-R634Q expressing cells to similar levels seen in WT (**80% AS events were restored from those mis-spliced in R634Q, Fig. 2e, Supplementary Table 8).**”
- Results: page 7 lines 234-236: “**However, this conclusion is based on ectopic expression of RBM20 variants, and further investigations are needed to characterize splicing restoration under physiological expression levels in cardiomyocytes.**”

- Discussion: page 12 lines 405-406: “Future studies with endogenously expressed RBM20 variants will identify mRNA targets and protein interactors of different relocalized RS-domain variants.”

2 - The mass spectrometry analysis in Supplementary Figure 3d is very important. Together with the loss of interaction of RBM20 with TNPO3, there is a large number of interactions gained in the cytoplasm. For example interactions with components of the splicing machinery, e.g. SRSF9, SRSF10, HNRNPH3, HNRNPA3, HNRNPA0, HNRNPC, etc. Could mislocalization of RBM20 in the cytoplasm sequester other components of the splicing machinery and thereby affect additional steps of mRNA processing together with the classical function of RBM20? This could explain, in part, the different splicing patterns observed in pathogenic RBM20 variants and RBM20 KOs reported before.

Indeed, from our data, there is evidence supporting that cytoplasmic RBM20 might sequester other components of the spliceosome, thereby further disrupting splicing of other targets not identified in a KO model, or disrupting additional steps of mRNA processing. We agree that this is a very interesting and important observation, and therefore, we now discussed this in main text (Results, page 6 lines 191-194): “The observed gain of interaction with spliceosome components could indicate that, RBM20 in the cytoplasm sequesters other components of the splicing machinery. This may result in disruption of mRNA processing, in addition to the splicing defects observed in RBM20 KO models¹³.”

3 - The current work appears to contradict a previous study (PMID: 34732726) showing that the RBM20 R636S mutant has a preference for 3' UTR sequences. How does this study relate to the earlier work? Is this preference affected by its nuclear targeting?

Fenix et al.²⁷ have shown using eCLIP that 3' UTRs are preferentially bound in the cytoplasm, and that those transcripts are not necessarily canonical RBM20 splicing targets. In the current study, we did not look at the mRNA binding preference of RBM20 WT or mutant using methods such as CLIP. Therefore, we cannot comment on potential changes in the mRNA binding preferences of RBM20 upon nuclear relocalization. Fenix et al. investigated a single mislocalized R636S mutant, and therefore it is unclear if differential mRNA binding is a general mechanism of action of RBM20 variants. In addition, Fenix et al. performed CLIP with the R636S variant that displays almost solely cytoplasmic localization, without testing its

binding preference in the nucleus, which makes it challenging to compare our findings with previous findings. Therefore, we do not contradict the previous study.

4 - The rationale for using iPSC-CMs carrying RBM20 S635FS mutation is not clear; is any truncated RBM20 protein expressed in these cells? The RBM20 localization in RBM20 S635FS cardiomyocytes overexpressing WT and eGFP-NLS-R634Q appears to be quite different to the endogenous localization pattern and the same experiment performed in HeLa cells.

We thank the reviewer for this comment. The RBM20 S635FS mutant generates a truncated protein that loses its splice regulatory function²². We used it as a *de facto* knockout of RBM20 which we transduced with lentiviral constructs expressing eGFP-RBM20-WT, eGFP-RBM20-R634Q, and eGFP-RBM20-NLS-R634Q. The S635S mutant was used in a similar function in Briganti et al. 2020²². The observed differences in localization of RBM20 in this experiment compared to endogenous levels are probably a result of the higher levels of RBM20 expression from the strong TetO promoter of the lentiviral construct. We discussed this limitation of protein over expression in Response to Reviewer 2 major comment 1 (page 16-17).

5 - The effect of TNPO3 OE on Ttn splicing in vivo (Figure 5i) is relatively minor. Is this due to the low infection efficiency or the low-binding affinity of P635L to the overexpressed Tnpo3? How does Ttn splicing correlate with the nuclear localization of RBM20 in cardiomyocytes isolated from Tnpo3 overexpressing mice?

We had the same hypotheses as the reviewer regarding the *Ttn* splicing effect upon TNPO3 overexpression. Generally, the observed effect on localization was not very strong (new Supplementary Figure 10 c), possibly due to not high infection efficiency and/or not perfect binding affinity of P633L-RBM20 to TNPO3 (Figure 4). Since we observed full restoration of localization for this variant in iPSC-CMs, when analyzing only positive cells (Fig. 5 a,b), we assume that, the low infection efficiency is likely the reason why the effect is more minor *in vivo*. To allow a more direct measurement of the restoration of RBM20 nuclear localization from our *in vivo* experiments, we now provide a quantification of the number of granules in CMs isolated from *Tnpo3* overexpressing or PBS-treated mice (Response Figure 13, new Supplementary Figure 10 c). We observed an insignificant decrease in the overall number of granules (p-value = 0.171), in line with the minor effects on *Ttn* splicing. We discuss this now in main text: Results page 12 lines 386-388: “*Tnpo3* overexpression *in vivo* (Fig. 5f) resulted

in partial rescue of RBM20 localization (Fig. 5g, Supplementary Fig. 10 c), *Tm* splicing (Fig. 5h, i), independently of *Rbm20* expression (Supplementary Fig. 10 d).”

This experiment serves as first evidence that, restoration of nuclear localization of mutant RBM20 may partially restore splicing *in vivo*. Importantly, and in line with the minor comment 7 of Reviewer 2, we do not consider TNPO3 overexpression as prime strategy to restore RBM20 localization, since TNPO3 is a general import factor resulting in potential side effects. Rather, we suggest that aptamers, bifunctional antibodies, or small molecule drugs that bind allosteric sites in RBM20, could be explored as therapeutic strategy that increases the affinity of mutant RBM20 to TNPO3. We added this to the main text (Discussion, pages 13-14 lines 450-454): “Enhancing RBM20 nuclear import could be achieved by other means, like endogenous tagging of *RBM20* with another NLS to be recognized by other importins. Alternatively, aptamers, bifunctional antibodies, or small molecule drugs that bind allosteric sites in RBM20, could be explored as therapeutic strategy that increases the affinity of mutant RBM20 to TNPO3.”

Response Figure 13 (new Supplementary Fig. 10c). Quantification of RBM20 granules per cell in primary cardiomyocytes isolated from mice treated with either PBS or AAV9 delivering *Tnpo3* cDNA, four weeks post injection (p value = 0.171, t-test).

REVIEWERS' COMMENTS

Reviewer #1 (Remarks to the Author):

The authors have fully addressed this reviewers concerns which in turn have substantially clarified their prior results. Further, their new ICS CRISPR screening data provide new insights into regulators of RBM20 protein localization. These data represent an important and significant resource for the cardiomyopathy research community.

Reviewer #2 (Remarks to the Author):

I commend the authors for a thorough and well-organised revision of their work. I am fully satisfied by the new experiments and by the changes in text that toned down/clarified some of the claims that are understandingly beyond the scope of the study to fully explore.

Reviewer #3 (Remarks to the Author):

I would like to thank the authors for addressing each of my (and the other two reviewers') critiques thoughtfully and clarifying a number of important points. Overall, the work is important, thorough, and convincing, and the limitations are clearly spelled out within the main text.

Specific points:

1. Given the limitations that are clearly outlined now within the main text, the statement "We demonstrate that mislocalized RBM20 variants retain their splice regulatory activity, which reveals that aberrant cellular localization drives the pathological phenotype." should be toned down.
2. In response to point 4 of my comments, the authors stated that they have used the RBM20 S635FS mutant as a knockout since it lacks splicing activity. However, the question remains as to whether these cells express a truncated protein and where it is localized. If the protein is expressed, it might exert additional effects through its interactions with other proteins (and maybe the overexpressed RBM20). Indeed, while throughout the manuscript overexpression of RBM20 results in a typical staining pattern with two characteristic foci, in this specific case, the localization pattern is very different. Is this representative? Given the high extent of rescue at core RBM20 targets, does this mean that the formation of sites for efficient alternative splicing (the two characteristic foci; PMID: 30948719) is not essential for the splicing function?
3. In response to point 4 of my comments, the authors provide quantification of the number of granules in mice overexpressing Tnpo3 versus control. Since the decrease is insignificant, presumably because of low infection efficiency, could the authors correlate the number of granules to Tnpo3 expression levels? What about the two characteristic Rbm20 foci?
4. The newly provided data in Supplementary Figure 7 regarding the effect of TNPO3 silencing on RBM20 localization in iPSC-CMs are not convincing. The correlations are based on TNPO3 staining that does not show typical localization (like in Supplementary Figure 10a).

Reviewer #3 (Remarks to the Author):

I would like to thank the authors for addressing each of my (and the other two reviewers') critiques thoughtfully and clarifying a number of important points. Overall, the work is important, thorough, and convincing, and the limitations are clearly spelled out within the main text.

We thank the reviewer for their positive feedback on our work.

Specific points:

1. Given the limitations that are clearly outlined now within the main text, the statement “We demonstrate that mislocalized RBM20 variants retain their splice regulatory activity, which reveals that aberrant cellular localization drives the pathological phenotype.” should be toned down.

We have now changed the text toning down the conclusion:

“We demonstrate that mislocalized RBM20 **RS-domain** variants retain their splice regulatory activity, which reveals that aberrant cellular localization **is the main driver** of their pathological phenotype.”

2. In response to point 4 of my comments, the authors stated that they have used the RBM20 S635FS mutant as a knockout since it lacks splicing activity. However, the question remains as to whether these cells express a truncated protein and where it is localized. If the protein is expressed, it might exert additional effects through its interactions with other proteins (and maybe the overexpressed RBM20). Indeed, while throughout the manuscript overexpression of RBM20 results in a typical staining pattern with two characteristic foci, in this specific case, the localization pattern is very different. Is this representative? Given the high extent of rescue at core RBM20 targets, does this mean that the formation of sites for efficient alternative splicing (the two characteristic foci; PMID: 30948719) is not essential for the splicing function?

Characterization of iPSC-CMs with RBM20-S635FS was done before by Briganti et al. (reference 22). In particular, figure S1 of Briganti et al. shows that the expression of *RBM20-S635FS* mRNA

is nearly zero compared to *RBM20*-WT. Given the high lentiviral overexpression levels of *RBM20* variants as described in Figure 2, we disregarded any potential background expression of *RBM20*-S635FS mRNA.

As the reviewer mentioned, *RBM20* usually forms on average two characteristic foci that are shown to be *Ttn* pre-mRNA splicing factories in the nucleus under physiological expression levels in human iPSC-CMs, which was also observed in this manuscript (Fig. 1a). However, here we analyzed eGFP-tagged *RBM20* that was exogenously expressed from a lentiviral vector, therefore resulting in higher than endogenous amounts of the protein. This overexpression explains the appearance of multiple foci in the nucleus. Our results do not contradict the essentiality of two characteristic foci for the splicing function, as the observed appearance of multiple foci does not mean that the two essential ones disappear but rather, it is our interpretation that additional foci appear due to high protein amounts.

3. In response to point 4 of my comments, the authors provide quantification of the number of granules in mice overexpressing *Tnpo3* versus control. Since the decrease is insignificant, presumably because of low infection efficiency, could the authors correlate the number of granules to *Tnpo3* expression levels? What about the two characteristic *Rbm20* foci?

We discussed the question of two characteristic foci upon *TNPO3* OE extensively in our response to Reviewer 2's major comment 1 from the first revision round. Briefly, as can be seen in Figures 5a and Supplementary Fig. 10a for iPSC-CMs, and Figure 5g for mice, endogenously-expressed *RBM20* variants are able to form the two characteristic foci if localized to the nucleus. We highlighted this in the main text previously: “Both variants were able to form two characteristic nuclear foci when relocated to the nucleus” (Results page: 11 lines 373-374).

We agree with Reviewer 3 and suppose that indeed the decrease in the number of granules is insignificant likely due to low infection efficiency. As for the precise assessment of *TNPO3* overexpression levels and / or infection efficiency and correlating them with *RBM20* localization in vivo, we are currently optimizing our staining protocols as part of another study. There, we are assessing the in vivo *TNPO3* overexpression in detail. Here, as stated in the manuscript, we added the in vivo data to provide a preliminary insight into the use of *TNPO3* as a potential candidate to treat DCM.

4. The newly provided data in Supplementary Figure 7 regarding the effect of TNPO3 silencing on RBM20 localization in iPSC-CMs are not convincing. The correlations are based on TNPO3 staining that does not show typical localization (like in Supplementary Figure 10a)

Supplementary Figure 10a shows localization of exogenously overexpressed eGFP-tagged TNPO3 delivered by a lentiviral vector, resulting in higher intensity and lower background signal, which cannot be interpreted as typical localization for the endogenous protein. In contrast, Supplementary Figure 7 shows antibody staining of endogenous TNPO3. As observed in other publications and the manufacturer's website for various antibodies, TNPO3 localization patterns differ across cell lines and even across different cells of the same line. However, the fact that in cells where we can expect a knockout of *TNPO3* (as judged by the change in RBM20 localization, as no gRNA selection was done) we observe no or little TNPO3 signal, indicates that the staining is specific.